

# Estimates of Exceedances of Critical Loads for Acidifying Deposition in Alberta and Saskatchewan

Paul A. Makar[1], Ayodeji Akingunola[1], Julian Aherne[2], Amanda S. Cole[1], Yayne-abeba Aklilu[3], Junhua Zhang[1], Isaac Wong[4], Katherine Hayden[1], Shao-Meng Li[1], Jane Kirk[5], Ken Scott[6], Michael D. Moran[1], Alain Robichaud[1], Hazel Cathcart[2], Pegah Baratzedah[1], Balbir Pabla[1], Philip Cheung[1], Qiong Zheng[1], Dean S. Jeffries[7]

[1]Air Quality Research Division, Environment and Climate Change Canada
[2]Environmental and Resource Studies, Trent University
[3]Environmental Monitoring and Science Division, Alberta Environment and Parks
[4]Watershed Hydrology and Ecology Research Division, Canada Centre for Inland Waters, Environment and Climate Change Canada
[5]Aquatic Contaminants Research Division, Environment and Climate Change Canada
[6]Technical Resources Branch, Environment Protection Division, Saskatchewan Ministry of the Environment
[7]Canada Centre for Inland Waters, Environment and Climate Change Canada

*For submission to Atmospheric Chemistry and Physics, Oil Sands Special Issue*

*Correspondence to*: Paul A. Makar (paul.makar@canada.ca)

**Abstract.** Estimates of potential harmful effects to ecosystems in the Canadian provinces of Alberta and Saskatchewan due to acidifying deposition were calculated, using a one year simulation of a high resolution implementation of the Global Environmental Multiscale – Modelling Air-quality and Chemistry (GEM-MACH) model, and estimates of aquatic and terrestrial ecosystem critical loads. The model simulation was evaluated against two different sources of deposition data; total deposition in precipitation and total deposition to snowpack in the vicinity of the Athabasca oil sands. The model captured much of the variability of observed ions in wet deposition in precipitation (observed versus model sulphur, nitrogen and base cation $R^2$ values of 0.90, 0.76 and 0.72, respectively), while being biased high for sulphur deposition, and low for nitrogen and base cations (slopes 2.2, 0.89 and 0.40, respectively). Aircraft-observation-based estimates of fugitive dust emissions, shown to be a factor of ten higher than reported values (Zhang *et al..*, 2017), were used to estimate the impact of increased levels of fugitive dust on model results. Model comparisons to open snowpack observations were shown to be biased high, but in reasonable agreement for sulphur deposition when observations were corrected to account for throughfall in needleleaf forests. The model-observation relationships for precipitation deposition data, along with the expected effects of increased (unreported) base cation emissions, were used to provide a simple observation-based correction to model deposition fields. Base cation deposition was estimated using published observations of base cation fractions in surface collected particles (Wang *et al.*, 2015).

Both original and observation-corrected model estimates of sulphur, nitrogen and base cation deposition were used in conjunction with critical load data created using the NEG-ECP (2001) and CLRTAP (2004, 2016, 2017) protocols for critical loads, using variations on the Simple Mass Balance model for forest and terrestrial ecosystems, and the Steady State



Water Chemistry and the First-order Acidity Balance models for aquatic ecosystems. Potential ecosystem damage at 2013/14 emissions and deposition levels was predicted for regions within each of the ecosystem critical load datasets examined here. The spatial extent of the regions in exceedance of critical loads varied between $1 \times 10^4$ and $3.3 \times 10^5$ km$^2$, for the more conservative observation-corrected estimates of deposition, with the variation dependant on the ecosystem and critical load protocol. The larger estimates (for aquatic ecosystems) represent a substantial fraction of the area of the provinces examined.

Base cation deposition was shown to have a neutralizing effect on acidifying deposition, and the use of the aircraft and precipitation observation-based corrections to base cation deposition resulted in reasonable agreement with snowpack data collected in the oil sands area. However, critical load exceedances calculated using both observations and observation-corrected deposition suggest that the neutralization effect is limited in spatial extent, decreasing rapidly with distance from emissions sources, due to the rapid deposition of emitted primary particles dust particles as a function of their size.

# 1 Introduction

Acidifying deposition was one of the first transboundary air pollution issues recognized as having ecological and economic consequences. In the late 1970's the UN Economic Commission for Europe (UNECE) developed a framework to assess the impacts of acidifying deposition, via the Convention on Long-Range Transboundary Air Pollution (LRTAP, or CLRTAP). The Convention described the scientific basis for the assessment of acidifying precipitation, and provided an internationally binding legal framework for mitigation and control of this and associated issues relating to regional and transboundary air pollution, and entered into force in 1983 (CLRTAP, 2016  2017). This and similar legislation elsewhere resulted in a requirement to be able to link sources of acidifying pollutants with downwind ecosystem impacts. While measurement networks were constructed to estimate acidifying deposition in sensitive ecosystems (and continue to be used for this purpose today, see Vet *et al.* (2014) for a review of current global acidifying precipitation networks and their status), the measurement sites are sparse due to their expense and the availability of the infrastructure to make observations in remote sensitive ecosystems. A further requirement thus arose: to provide estimates of acidifying pollution to sensitive ecosystems to complement the available observations.

This requirement drove the development of the first generation of chemical transport models (CTM's), which made use of inventories of the emissions of different pollutants, detailed descriptions of gas, aqueous-phase and particle chemistry, speciated gas and particle and meteorological forecast model information, to describe the downwind transformation and deposition of acidifying pollutants (cf. Eliassen *et al.*, 1982; Calvert and Stockwell, 1983; Venkatram and Karamchandani, 1988; Chang *et al.*, 1987). The models increased in sophistication over the years to include more detailed descriptions of gas and aqueous chemistry, particle chemistry and particle microphysics (cf. Binkowski and Shankar, 1995; Binkowski and Roselle, 2003; Gong *et al.*, 2006). The next generation of models was extended to merge previously separate chemistry and meteorological forecasting models into unified frameworks (Grell *et al.*, 2005; Vogel *et al.*, 2009; Moran *et al.*, 2010,



Baklanov *et al.*, 2014). The most recent versions of these models included incorporation of the impacts of model-generated aerosols into radiative transfer, and hence estimation of the impacts of feedbacks between atmospheric pollution and weather forecasting (ensemble comparisons of these fully coupled models with observations may be found in Makar *et al.*, 2015(a,b) and Im *et al.*, 2015 (a,b)).

Concurrent to the ongoing CTM development, methodologies were extended to improve the estimation of the effects of acidifying emissions on sensitive ecosystems. A key tool for this work are spatial maps of ecosystem "critical loads", where a critical load is defined (Nilsson and Grennfelt, 1988) as "A quantitative estimate of an exposure to one or more pollutants below which significant harmful effects on specified sensitive elements of the environment do not occur according to present knowledge". In the context of acidifying deposition, the critical load is the upper limit to the deposition flux of acidifying

pollutants, below which ecosystem damage due to that deposition will not occur. A critical load *exceedance* is thus defined as the *excess* deposition of acidifying pollutants *above* the critical load. Guidelines for the determination of UNECE CLRTAP critical load data were first published in 1996, with subsequent updates (CLRTAP, 2004, 2016, 2017). In North America, modified protocols were initially adopted, to provide upper limit estimates of critical loads, for cases in which more detailed data were unavailable, via an agreement between the eastern US States and eastern Canadian provinces (New

England Governors – Eastern Canadian Premiers; NEG-ECP, 2001).

Estimates of critical loads for acidifying deposition for different ecosystems are calculated using different models, but all are predicated on the concept of charge balance at steady-state; the critical load models determine the excess flux of cations available in the natural ecosystem, which could potentially balance the anions added due to acidifying deposition. The critical load calculations may thus depend on estimates of the deposition flux of both anions and cations. The anions of

interest are the total (wet plus dry) atmospheric deposited sulphur, $S_{dep}$, and total atmospheric deposited nitrogen, $N_{dep}$, where the sulphur deposition is assumed to have two negative charges (all forms of $S_{dep}$ are assumed to eventually be transformed to, and contribute to deposition as, $SO_4^{2-}$), and nitrogen is assumed to have one negative charge (all forms of $N_{dep}$ are assumed to eventually be transformed to, and contribute to deposition as, $NO_3^-$). Cations of interest include $Mg^{2+}$, $Ca^{2+}$, $K^+$, and $Na^+$, collectively referred to as base cations, and their net deposition from the atmosphere when converted to molar

charge equivalents, is referred to as $BC_{dep}$. For terrestrial ecosystems $BC_{dep}$ must be estimated from observations or CTM predictions, while for aquatic ecosystems, the total base cation concentrations within water due to atmospheric deposition and other sources are derived from direct sampling and laboratory analysis of ecosystem surface water.

We note that while an exceedance of critical loads identifies the *potential* for ecosystem damage to occur, critical loads are

based on the concept of a chemical steady-state, and depending on the buffering mechanisms available in an ecosystem, the steady-state defined by an exceedance of critical loads may not take place until some point in the future. Once exceedances of critical loads have been identified, dynamic models may be used to assess the time delay until damage occurs and/or the time required for recovery of the ecosystem subsequent to that damage (CLRTAP, 2015).



Atmospheric deposition of $S_{dep}$, $N_{dep}$ and $BC_{dep}$ may thus influence the estimation of critical load exceedances. For terrestrial ecosystems, if the value of $BC_{dep} - S_{dep} - N_{dep}$ is positive, then critical loads will not be exceeded. For aquatic ecosystems, if the value of the total charge balance of the critical load (which includes all forms of input of base cations to the system including $BC_{dep}$) is greater than the added anions, critical loads will not be exceeded. Emissions sources of base cations may

thus act to counteract the emissions sources of $S_{dep}$ and $N_{dep}$, depending on the relative emissions levels, the locations of the sources, etc. For example, some observations in the immediate environs (within 135 km) of emission sources located within the Athabasca oil sands region of Canada have shown that $BC_{dep}$ exceeds $S_{dep}$ and $N_{dep}$, implying that alkylization (rather than acidification) may be happening in this region (Watmough *et al.*, 2014). While the disturbance to the ecosystems due to the increase in pH associated with the excess base cations may cause other ecosystem effects, this finding has been used to

imply that acidifying deposition, and the consequent potential ecosystem damage due to emissions from these facilities is unlikely. This implication has been re-evaluated on a larger scale in the present work.

The provinces of Alberta and Saskatchewan are home to the majority of Canada's petrochemical extraction and refining infrastructure, in addition to other industries such as coal-fired power generation, and account for a substantial fraction of the

Canadian emissions of sulphur dioxide, nitrogen oxides, and ammonia (34, 43, and 50 %, respectively, NPRI, 2013), Within the province of Alberta , emissions originating in the Athabasca oil sands account for approximately 38.4, 3.8, and 1.0% of the Alberta total emissions of these three chemicals. These three pollutants, and their gas, particulate and aqueous-phase reaction products, are the main anthropogenic sources of $S_{dep}$ and $N_{dep}$ within this region. As we will show below, the provinces are also home to terrestrial and aquatic ecosystems which are sensitive to acidifying deposition (i.e. have relatively

low critical loads for acidifying deposition). Calculations of exceedances of critical loads within this region are therefore of interest, to assess the potential for ecosystem damage associated with these emissions, and are the focus of our work.

We use a combination of a fourth-generation CTM (the Global Environmental Multiscale – Modelling Air-quality and CHemistry; GEM-MACH), critical load estimates for aquatic and terrestrials determined using different protocols, and two

different surface observation datasets, to predict the extent to which critical loads are being exceeded, over large portions of the Canadian provinces of Alberta and Saskatchewan.

We begin with a description of the critical load data used in our evaluation, follow with a description of GEM-MACH (with a focus on its components which pertain to $S_{dep}$ and $N_{dep}$), an evaluation of the model performance, corrections to the model

predictions based on observations, and end with estimates of exceedances for terrestrial and aquatic ecosystems and our conclusions.





## 2 Methodology

### 2.1 Critical Loads of Acidic Deposition

#### 2.1.1 Critical Loads and Critical Load Exceedances – Definitions

Critical loads were estimated following methodologies set out under the UNECE Convention on Long-range Transboundary

Air Pollution (CLRTAP, 2016, 2017; de Vries, *et al.*, 2015). We define first the equations used for determining critical loads, and follow with the description of the data used to estimate critical loads of acidifying sulphur (S) and nitrogen (N) for terrestrial and aquatic ecosystems in Alberta and Saskatchewan, based on a Canada-wide implementation (Carou *et al.*, 2008), and two more recent studies focused on terrestrial ecosystems in the province of Alberta,and aquatic ecosystems in northern Alberta and Saskatchewan (Cathcart *et al.*, 2016).

For terrestrial ecosystems, critical loads of acidity were estimated using the steady-state (or simple) mass balance (SSMB) model, which links deposition to a chemical variable (the 'chemical criterion') in the soil, or soil solution, associated with ecosystem effects. The violation of a specific value (the 'critical limit') for the chemical criterion is associated with potential ecosystem damage. The most widely used soil chemical criterion is based on the ratio of base cations to aluminum (Bc:Al

where Bc is the sum of in equivalents of calcium ($Ca^{2+}$), magnesium ($Mg^{2+}$) and potassium ($K^+$)) in soil solution. The acidifying impact of S and N define a critical load function (CLF) incorporating the most important biogeochemical processes that affect long-term soil acidification (CLRTAP, 2004). The function is defined by three quantities (see Equations 1 to 3): the maximum critical load of S ($CL_{max}(S)$); minimum critical load of N ($CL_{min}(N)$); and the maximum critical load of N ($CL_{max}(N)$). The level of protection for the chosen receptor ecosystem (e.g., forests) is specified via the receptor

ecosystem's critical acid neutralizing capacity for leaching ($ANC_{le,crit}$, Equation 4). Critical loads of acidity for terrestrial ecosystems are defined in units of "equivalents" (ionic charge × moles).

$$CL_{max}(S) = BC_{dep} + BC_w - Cl_{dep} - Bc_u - ANC_{le,crit} \qquad (1)$$

$$CL_{max}(N) = CL_{min}(N) + \left(CL_{max}(S)/(1 - f_{de})\right) \qquad (2)$$

$$CL_{min}(N) = N_i + N_u \qquad (3)$$

$$ANC_{le,crit} = -Q^{\frac{2}{3}}\left[\frac{3}{2}\left(\frac{Bc_w + Bc_{dep} - Bc_u}{(Bc:Al)_{crit}K_{gibb}}\right)\right]^{\frac{1}{3}} - \frac{3}{2}\left(\frac{Bc_w + Bc_{dep} - Bc_u}{(Bc:Al)_{crit}}\right) \qquad (4)$$

The remaining terms in these equations include: $BC_{dep}$, the non-marine annual base cation deposition, $BC_w$, is the release of soil base cations owing to physical and chemical breakdown (weathering) of rock and soil minerals, $Cl_{dep}$ the non-marine chloride deposition, $Bc_u$, the average base cation removal due to the harvesting of base-cation-containing biomass from the

ecosystem (Bc = the sum in equivalents of $Ca^{2+}$, $Mg^{2+}$ and $K^+$), $f_{de}$, the denitrification fraction (loss of nitrogen to $N_2$), $N_i$, the long-term net immobilization of nitrogen in the rooting zone, and $N_u$, the average removal of nitrogen from an ecosystem




due to other forms of removal (e.g., harvesting), Q is the soil percolation or catchment runoff, $(Bc{:}Al)_{crit}$, is the critical value of the non-sodium base cation to aluminum ion ratio described above, and $K_{gibb}$ is the Gibbsite equilibrium constant.

For aquatic ecosystems, two steady-state models have been widely used for calculating critical loads (Henriksen and Posch, 2001; CLRTAP, 2004, 2016, 2017; de Vries *et al.*, 2015): the Steady-State Water Chemistry (SSWC) model and the First-order Acidity Balance (FAB) model.

The SSWC model requires volume-weighted mean annual water chemistry and runoff volume (Q) to calculate critical loads of S acidity.

$$CL(A) = Q([BC]_0^* - ANC_{limit}) \tag{5}$$

where $[BC]_0^*$ is the sea salt corrected pre-acidification concentration of base cations in the surface water, and $ANC_{limit}$ is the ANC (concentration) limit above which no damage to the specified biological indicator (e.g., fish) occurs. The sea salt correction, denoted by a superscript asterisk, assumes all chloride originates from sea salt; the current concentrations of base cations, $SO_4^{2-}$(aq) and $NO_3^-$(aq) in water along with empirical functions (see below) are used to estimate $[BC]_0^*$, following CLRTAP protocols (CLRTAP, 2016); further details regarding the sensitivity of the critical load estimates to these functions are described in Cathcart *et al.* (2016).

The FAB model (Posch *et al.*, 2012) allows the simultaneous calculation of critical loads of acidifying S and N deposition similar to the SSMB model widely used for forest soil critical loads. In addition to processes in the terrestrial catchment soils, such as uptake, immobilization and denitrification, the FAB model includes in-lake retention of N and S. The derivation of the FAB model starts from the charge balance at the outlet of a lake:

$$S_{runoff} + N_{runoff} = \sum_Y Y_{runoff} - ANC_{limit}$$
$$Y = Ca + Mg + K + Na - Cl \tag{6}$$

Steady-state mass balance equations for the runoff terms for each ion (X) are then derived as a function of the total amount of ions entering the lake $(X_{in})$ and dimensionless retention factors $(\rho_X)$:

$$X_{runoff} = (1 - \rho_X)X_{in} \tag{7}$$

The formulae for $X_{in}$ depends on the specific ion; $S_{in}$ depends on deposition alone, $N_{in}$ includes terms for net immobilization (i), growth uptake (u), and denitrification $(f_{de})$, and base cations includes terms for deposition (dep), weathering (w) and uptake (u). An equation of the following form results (the summation is over the different components within the catchment, usually simplified to be "lake" and "non-lake" (i.e. m=1 in the equation which follows), and $A_j/A$ is the relative area of the components $(A_j)$ to the total catchment area (A):





$$(1 - \rho_S) \sum_{j=0}^{m} \frac{A_j}{A} S_{dep,j} + \sum_{j=0}^{m} \frac{A_j}{A} (1 - f_{de,j}) (N_{dep,j}(1 - f_{u,j}) - N_{i,j} - N_{u,o,j})_+ =$$
$$\sum_Y \left[ (1 - \rho_Y) \sum_{j=0}^{m} \frac{A_j}{A} (Y_{dep,j} - Y_{w,j} - Y_{u,j})_+ \right] - Q \cdot ANC_{limit} \qquad (8)$$

where the "+" subscript refers to the maximum value of the term within the brackets across the catchment components j (lake and non-lake). $S_{dep}$ includes all forms of sulphur deposition (gaseous $SO_2$ dry deposition, particulate dry deposition, and wet deposition of bisulphate and sulphate ions), converted to charge x mole equivalent deposition of $SO_4^{2-}$. $N_{dep}$ includes all

forms of nitrogen deposition (gas phase dry deposition of NO, $NO_2$, $NH_3$, HONO, $HNO_3$, peroxyacetylnitrate, organic nitrates, dry deposition of particulate nitrate and ammonium, and wet deposition of ammonium and nitrate ions), converted to the charge x mole equivalent deposition of $NO_3^-$. Setting $N_{dep} = 0$ in (8) results in a formula for $CL_{max}(S)$, and setting $S_{dep} = 0$ results in a formula for $CL_{max}(N)$. The denitrification fraction was estimated as $f_{de} = 0.1 + 0.7 \cdot f_{peat}$, where $f_{peat}$ is the fraction of wetlands in the terrestrial catchment, and $CL_{min}(N)$ was taken to be $N_i + N_u$ ($N_i$ was set to the regional default value

of 35.7 eq ha$^{-1}$), and $N_u$ was based on estimates of forest biomass (Canadian Forestry Service National Forest Inventory) and literature data for the concentration of N in biomass). The net uptake of N on land was assumed to be constant ($f_{u,1}=0$), and the flux of base cations (right-hand-side of (8)) is determined using the SSWC model via equation (5). In both the SSWC and FAB models, the value of $[BC]_0^*$ is derived using an "F-factor" equation describing the change in charge balance over time from pre-industrial (time 0) to current (time t) conditions:

$$[BC]_0^* = [BC]_t^* - F \cdot ([SO_4]_t^* + [NO_3]_t - [SO_4]_0 - [NO_3]_0) \qquad (9)$$

The F-factor in (9) depends on the pre-industrial base cation concentration and (9) is solved iteratively. The in-lake retention coefficients for S and N ($\rho_S$ and $\rho_N$, respectively) are modelled by a kinetic equation (Kelly *et al.*, 1987) making them a function of runoff, the lake:catchment ratio and net mass transfer coefficients for S and N. It is assumed that the lakes and their catchments are small enough to be properly characterised by average soil and

lake-water properties; furthermore, all of the lakes examined here are treated as headwater lakes, and larger lakes are excluded from the analysis.

The risk of negative impacts owing to acidifying S and N deposition, i.e. deposition in excess of the critical load, is based on the magnitude and areal extent of exceedance. Exceedance of the critical load of S acidity for aquatic ecosystems under the

SSWC model is defined as

$$Ex(S_{dep}) = S_{dep} - CL(A) \qquad (10)$$

Where $S_{dep}$ is the sum of deposition of all forms of S, where each mole of S is treated as $SO_4^{2-}$ (i.e. two equivalents per mole of S deposited). Exceedances of acidity are defined as instances where the addition of acidity in the form of S exceeds the net buffering capacity. In contrast, under the SSMB and FAB models there is no unique amount of S and N to be reduced to

reach non-exceedance; Exceedance for a given S and N deposition pair is the sum of the S and N deposition reductions required to reach the critical load function (CLF) by the 'shortest' path (Figure 1). The computation of the exceedance




function followed the methodology described in CLRTAP (2004, 2016, 2017). In some instances, S deposition (or N) must be reduced to achieve non-exceedance.

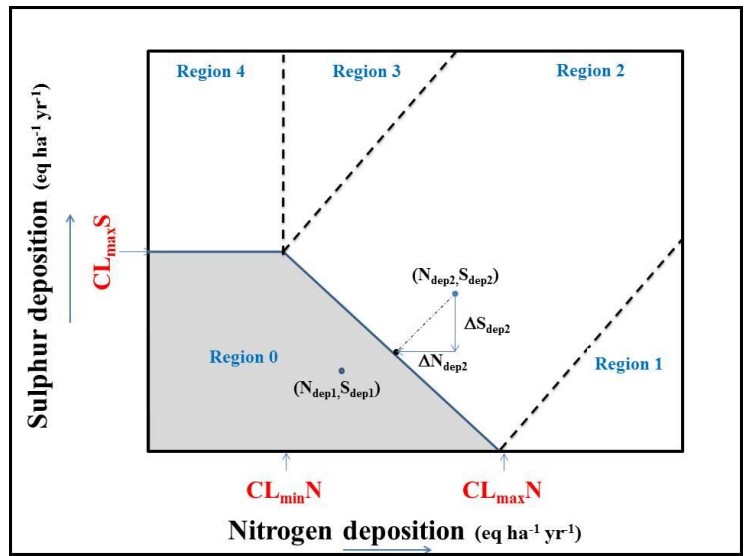

**Figure 1. Critical Load Function, showing exceedance Regions 1 through 4 and the below exceedance region 0.**

Region 0 in Figure 1 denotes ecosystems for which $S_{dep}$ and $N_{dep}$ are *below* exceedance levels (i.e. deposition does not exceed critical load).  For this region, we introduce a term $E_0$, a negative number indicating the proximity of deposition in region 0 to the nearest bordering exceedance region.  Exceedances are calculated as follows:

$$Ex\left(N_{dep}, S_{dep}\right) = \begin{cases} E_0 & \left(S_{dep}, N_{dep}\right) \in Region\ 0 \\ N_{dep} - CL_{max}(N) + S_{dep} & \left(S_{dep}, N_{dep}\right) \in Region\ 1 \\ N_{dep} - N_0 + S_{dep} - S_0 & \left(S_{dep}, N_{dep}\right) \in Region\ 2 \\ N_{dep} - CL_{min}(N) + S_{dep} - CL_{max}(S) & \left(S_{dep}, N_{dep}\right) \in Region\ 3 \\ S_{dep} - CL_{max}(S) & \left(S_{dep}, N_{dep}\right) \in Region\ 4 \end{cases} \quad (11)$$

For Region 2, the exceedance is defined with respect to the closest point between the diagonal line joining the points
$(CL_{min}(N), CL_{max}(S))$ and $(CL_{max}(N), 0))$, defined via:

$$N_0 = \frac{N_{dep} + m S_{dep} + m^2 CL_{max}(N)}{1+m^2} \quad (12)$$

$$S_0 = m(N_0 - CL_{max}(N)) \quad (13)$$

where

$$m = \frac{CL_{max}(S)}{CL_{min}(N) - CL_{max}(N)} \quad (14)$$

We define here $E_0$, a negative quantity defining the minimum distance below the exceedance lines from the $N_{dep}$, $S_{dep}$ point on Figure 1:

$$E_0 = \max\left[\left(S_{dep} - S_A\right), \left(N_{dep} - N_A\right)\right] \quad (15)$$

where



$$S_A = \begin{cases} CL_{max}(S) & N_{dep} < CL_{min}(N) \\ m\big(N_{dep} - CL_{max}(N)\big) & CL_{min}(N) \leq N_{dep} < CL_{max}(N) \end{cases}$$
$$N_A = \frac{S_{dep}}{m} + CL_{max}(N) \qquad (16)$$

Three different sources of critical load data were used in this work. We begin with Canada-wide critical loads of acidity, which employ modifications to the above UNECE methodology (CLRTAP, 2004), used in eastern North America (NEG-ECP, 2001, Ouimet, 2005), and then expanded across Canada (Carou *et al.*, 2008; Jeffries *et al.*, 2010; Aherne, 2013). We

follow with more recent estimated determined using the UNECE protocols, for terrestrial ecosystems for the province of Alberta (Aklilu, 201x), and aquatic ecosystems in northern Alberta and Saskatchewan (Cathcart *et al.*, 2016).

**2.1.2 Canada-wide Critical Loads of Acidity: Lakes and Forest Soils**

The earliest critical load data used in the current work are for forest and lake ecosystems, and resulted from updates to Environment and Climate Change Canada databases, subsequent to the publication of the Canadian Acid Deposition Science

Assessment (ECCC, 2004; Jeffries and Ouimet, 2005).

Lake chemistry surveys were conducted in Canada in order to obtain data for critical load estimates (Jeffries *et al.*, 2010). Critical loads of acidity for each sampled lake were estimated using the SSWC model (Henriksen and Posch, 2001). In addition to the lake survey data, other inputs to the SSWC include ecosystem-specific characteristics that were estimated

using a mixture of methods, including broad mineralogical, geological, hydrological and biological surveys, and other related information. At the time these aquatic critical load data were collected, acidic deposition estimates at ECCC were conducted using A Unified Regional Air-quality Modelling System (AURAMS; Gong *et al.*, 2006). The critical load values for lakes were therefore gridded to the map of Canada used by the AURAMS model, with a grid-cell resolution of 45 km × 45 km. The SSWC critical load values for each surveyed lake contained within each AURAMS grid-cell were compared –

when data from multiple lakes within the same grid cell were available, the lowest fifth percentile of the resulting critical load values was assigned to that grid cell (for grid cells containing smaller numbers of lakes, the critical load for the most sensitive lake was used).. The lake critical load data thus represent the most sensitive lake ecosystems within the given 45 km$^2$ grid cell. The data were subsequently re-mapped to the higher resolution GEM-MACH grid used here; the centroids of those 2.5 km GEM MACH grid-cells falling within the AURAMS lake critical load polygons were assigned the

corresponding AURAMS grid critical load values. The resulting critical loads are shown in Figure 2 (a), with red values indicating the most sensitive ecosystems and blue values indicating the least sensitive ecosystems. AURAMS cells for which no lake information was available were assigned "null" values (shown in grey). These critical load data identified lake ecosystems in north-eastern Alberta, northern Saskatchewan, and north-western Manitoba as particularly sensitive to acidifying precipitation.





The forest ecosystem critical loads used here began with provincial and regional surveys that were combined to form a unified Canada-wide critical load dataset (Carou *et al.*, 2008). Critical load and exceedance of S and N were estimated for forest soils following the methodology and guidelines established by the NEG-ECP (NEG-ECP 2001, Ouimet 2005), which largely follow the UNECE methodology (CLRTAP, 2004, 2016, 2017). The long-term critical load was estimated using

SSMB model; the key spatial data-sets (or base maps) required as inputs are atmospheric deposition, base cation weathering rate and critical alkalinity leaching. Average annual total (wet plus dry) atmospheric base cation deposition data during the period 1994–1998 was estimated using observed wet deposition, observed air concentrations, and modelled meteorological data along with land-use specific dry deposition velocities, and mapped on the Global Environmental Multiscale (GEM) grid at a resolution of 35 km × 35 km (see Section 2.2 for details on GEM and its companion on-line chemistry module, GEM-

MACH). Under the NEG-ECP protocol, weathering rates were estimated using a soil type–texture approximation method (Ouimet, 2005). The approach estimates weathering rate from texture (clay content) and parent material class. This method was used in conjunction with the Soil Landscapes of Canada (SLC, version 2.1) to estimate base cation weathering rates across western Canada. Under the NEG-ECP (2001) protocol, several simplifying assumptions were applied to equations (1) through (4): (a) a critical Bc:Al ratio of 10, and a $K_{gibb}$ of 3000.0 were used, (b) harvesting removals were not considered;

therefore, long-term net uptake $N_u$ and $Bc_u$ were set to zero, (c) denitrification and net N immobilization was assumed to occur, but at an invariant low level appropriate for well-drained upland forest ecosystems (both $N_i$ and the equivalent of the $\left(CL_{max}(S)/(1 - f_{de})\right)$ term of equation (2) were set to 35.7 eq ha$^{-1}$ yr$^{-1}$), (d) the deposition of Cl ions was assumed to be zero, and (e) the weathering release of soil base cations ($BC_w$) was assumed to be dependent on temperature. The net critical load functions for the forest ecosystems with these simplifications becomes:

$$CL(S + N) = BC_{dep} + BC_w(T) + Q^{\frac{2}{3}}\left[\frac{3}{2}\left(\frac{Bc_w(T)+Bc_{dep}}{(Bc:Al)_{crit}K_{gibb}}\right)\right]^{\frac{1}{3}} - \frac{3}{2}\left(\frac{Bc_w(T)+Bc_{dep}}{(Bc:Al)_{crit}}\right) + 2\,(35.7) \tag{17}$$

With

$$BC_w(T) = BC_w e^{\left[3600\left(\frac{1}{281}-\frac{1}{274+T}\right)\right]} \tag{18}$$

Where T is the temperature in degrees C, and $Bc_w = 0.75\ BC_w$ (NEG-ECP, 2001, Nasr *et al.*, 2010, Whitfield *et al.*, 2010, Aherne, 2011).

The resulting critical load values were referenced to the corresponding GIS polygons under the SLC containing that soil type, resulting in a Canada-wide map of forest soil critical loads for acidity. These polygons were superimposed on the map of GEM-MACH 2.5 km$^2$ resolution grid cells. Similar to the approach for lake critical loads described above, the lowest 5[th] percentile value from the forest critical load polygons existing within each GEM-MACH grid cell was assigned to that grid cell. The forest soil critical load values on the resulting GEM-MACH grid cell thus represent the most sensitive forest

ecosystems within that grid cell. Polygons for which forest soils were not present were assigned "null" values. Under the NEG-ECP protocol (NEG-ECP, 2001) critical loads were simplified for acidic deposition ($S_{dep} + N_{dep}$), as such exceedance was defined for combined deposition:



$$Ex\left(N_{dep}, S_{dep}\right) = S_{dep} + N_{dep} - CL(S + N) \tag{19}$$

The resulting critical load map for forest soils is shown in Figure 2(b), with the same colour scale as Figure 2(a). The lake
ecosystems can be seen to be more sensitive to acidic deposition compared to forest soil ecosystems (lower critical load
5    values, red shades in Figure 2).

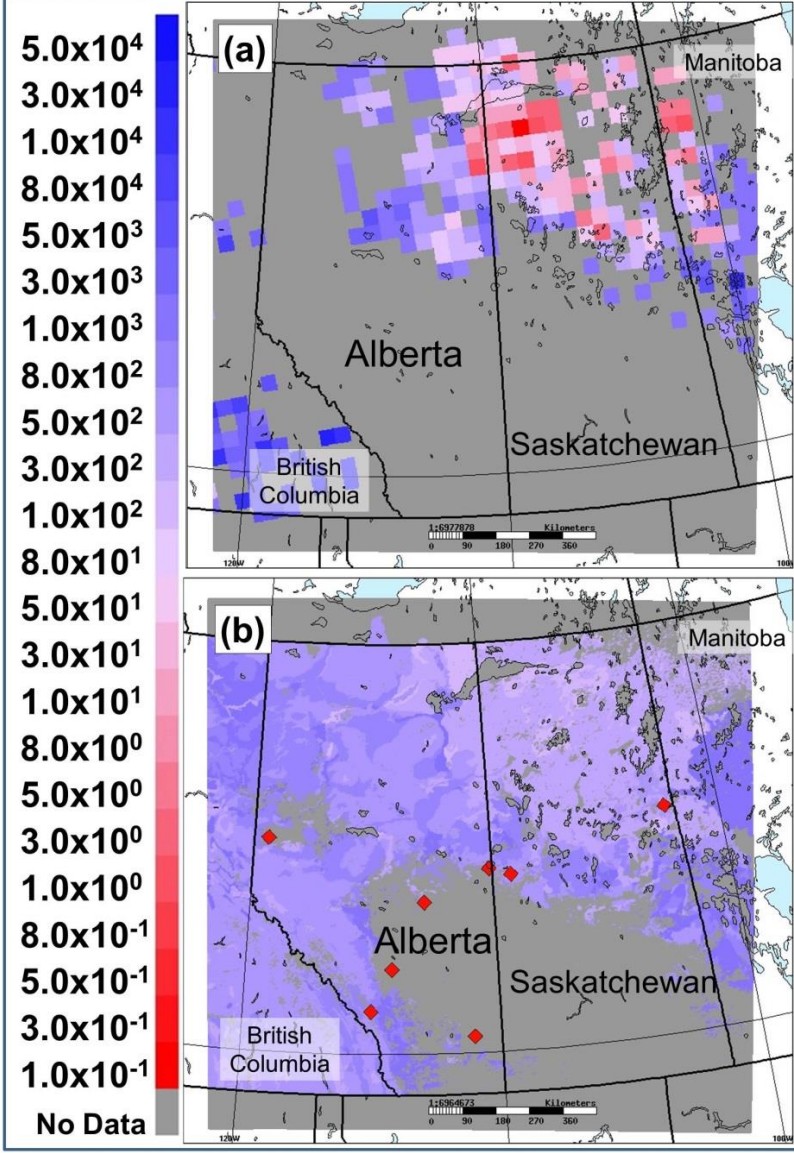

**Figure 2. Critical loads of acidity on the 2.5km GEM-MACH domain, based on a Canada-wide implementation: (a) Lake ($S_{dep}$)
and (b) Forest ($S_{dep}+N_{dep}$) ecosystems (eq ha$^{-1}$ yr$^{-1}$). Forest values were calculated using 1994-1998 interpolated/extrapolated $BC_{dep}$
observations (diamonds show the location of those Canada-wide stations used to estimate $BC_{dep}$, which reside within the 2.5km
10    resolution model domain). Red regions (low numbers) on the scale have the lowest critical loads, hence are the most sensitive to
deposition.**



Later in this work, we discuss the effect of permutations on the assumed level of atmospheric base cation deposition in the above calculations towards the resulting estimates of critical load and critical load exceedances.

### 2.1.3 Province of Alberta: Critical Loads of Acidity for Terrestrial Ecosystems

The SSMB model was used to estimate $CL_{max}(S)$, $CL_{max}(N)$, and $CL_{min}(N)$, for terrestrial ecosystems in the province of
Alberta following methods recommended under the Convention on Long-Range Transboundary Air Pollution (CLRTAP, 2016, 2017). Critical loads were not derived for areas comprising cultivated or agricultural land, rock, and exposed or developed soil. Our initial estimate of non-marine annual base cation deposition ($BC_{dep}$) was the interpolated/extrapolated 1994-1998 base cation database described above. The release of base cations as a result of chemical dissolution from the soil mineral matrix ($BC_w$) followed the soil texture approximation method (equation 18), with soil information vertically
weighted to a rooting depth of 50 cm to create a homogeneous soil layer for calculations. Soil information for this calculation was obtained from the Soils Landscape Canada version 3.2 database (AAFC, 2010). The average base cation removal in harvested biomass ($Bc_u$) was calculated using the Alberta Vegetation Index dominant forest cover database to determine type and distribution of forests (ABMI, 2010), harvest information (AAF, 2015), and information on nutrient uptake by forest type (Paré *et al.*., 2012). For unmanaged ecosystems (i.e. not harvested) $Bc_u$ was set to zero, and the
removal of biomass due to grazing in grasslands was set to 8 eq ha$^{-1}$ yr$^{-1}$. The acid neutralization capacity leaching ($ANC_{le,crit}$) was determined using critical Bc:Al ratios applied by vegetation type (a $(BC:Al)_{crit}$ ratio of 6 was used for Mixed Forest, Shrubland and Broadleaf Forest, while Coniferous Forest and Grassland made use of ratios of 2 and 40, respectively). The denitrification fraction ($f_{de}$) was assigned using a seven level scale (AAFC, 2010; CLRTAP, 2016, 2017). $f_{de}$ values for "very rapid", "well", "moderately well", "imperfectly", "poorly", and "very poorly" drained soils were respectively 0.0, 0.1,
0.2, 0.4, 0.7, and 0.8. The long-term net immobilization of N in the rooting zone was assumed to be 0.5 kg N ha$^{-1}$ yr$^{-1}$ (35.7 eq ha$^{-1}$ yr$^{-1}$) (CLRTAP, 2016, 2017). The average removal of N from an ecosystem ($N_u$) followed Pregitzer *et al.* (1990), using Alberta Vegetation Index dominant forest cover data to identify the type and distribution of forests (Alberta, 2016), and nutrient information from the Canadian Tree Nutrient Database (Paré *et al.*, 2012). For grasslands the value of $N_u$ was set to 43 eq ha$^{-1}$ yr$^{-1}$ to account for nitrogen removal due to grazing.

The resulting maps for $CL_{max}(S)$, $CL_{max}(N)$, and $CL_{min}(N)$ for Alberta Terrestrial Ecosystems are shown in Figure 3 (using the same colour scale as Figure 2). $CL_{max}(S)$ values (Figure 3(a)) are lower than the forest critical load values created under the NEG-ECP (2001) protocol (Figure 2(b)), reflecting the more detailed treatment of the acid neutralizing capacity term, and the impacts of harvesting on estimated critical loads. NEG-ECP (2001) protocol critical load estimates were intended as
"upper limits", that is, they were expected to underestimate ecosystem sensitivity, relative to the more detailed calculation used in the creation of Figure 3.




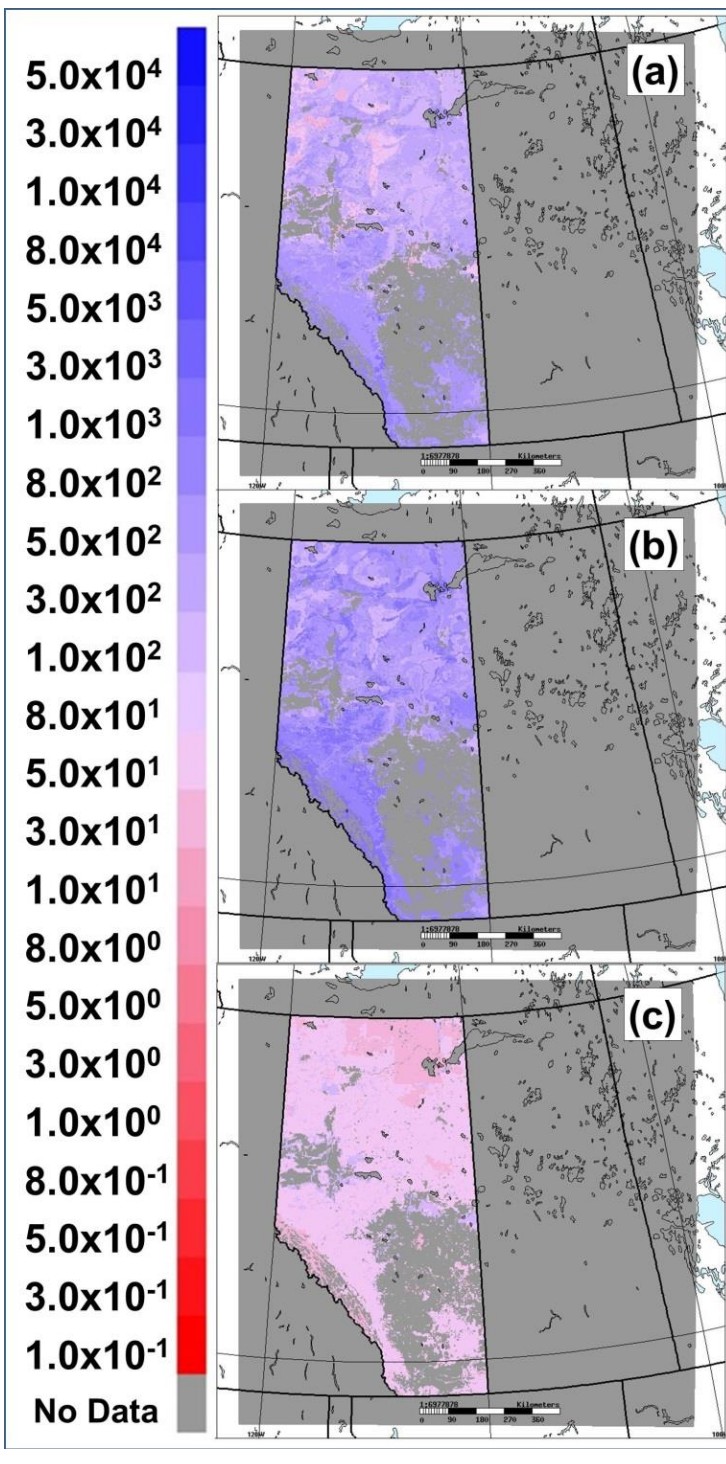

**Figure 3. Critical loads of acidity with respect to sulphur and nitrogen for terrestrial ecosystems, Province of Alberta implementation (eq ha$^{-1}$ yr$^{-1}$), using $BC_{dep}$ from interpolated/extrapolated 1994-1998 observations. (a) Maximum critical load for sulphur. (b) Maximum critical load for nitrogen. (c) Minimum critical load for nitrogen.**





### 2.1.4 Northern Alberta and Saskatchewan: Critical Loads of Acidity for Aquatic Ecosystems

The critical load data for lake ecosystems described in Section 2.1.2 were updated for aquatic ecosystems in northern Alberta and Saskatchewan,as part of an ongoing project to update previous Canada-wide critical load data, following the full UNECE protocols (equations 1 through 16), resulting in new spatially georeferenced critical load maps for acidity with respect to $S_{dep}$ and $S_{dep} + N_{dep}$ (equations (1-9,17,18) respectively).

Water chemistry from 2409 observations of 1344 lakes were used to produce predictive maps of lake concentrations for four target variables across northern Alberta and Saskatchewan for the determination of critical loads: base cations (BC), dissolved organic carbon (DOC) and sulphate ($SO_4^{2-}$). A regression kriging approach was used to generate (predict) water chemistry for 137,587 lake catchments following Cathcart *et al.* (2016). Regression kriging is a spatial interpolation method wherein a regression of the target variable on covariate variables (e.g., landscape characteristics such as soil, climates, vegetation; a total of 185 covariates were included) is combined with kriging on the regression residuals (Hengl *et al.*, 2007; Hengl, 2009). The water chemistry (target variable) data were obtained from Environment and Climate Change Canada's Level 1 and 2 monitoring networks in Saskatchewan, Alberta and the Northwest Territories (Jeffries *et al.*, 2010), lake surveys undertaken by the Government of Saskatchewan (Scott *et al.*, 2010), the RAMP monitoring network in Alberta (RAMP, 2016), and from the Alberta Environment and Parks surface water quality data portal (Alberta Environment and Parks, 2016). Critical loads of acidity for lakes were calculated from the predictive maps of lake concentration using the SSWC and the FAB models. As previously noted, the FAB model extends the SSWC model to consider terrestrial and aquatic sources and sinks of S and N, similar to the SSMB (Henriksen and Posch, 2001; Posch *et al.*, 2012). A variable $ANC_{limit}$ was used, adjusted for the strong acid anion contribution from organic acids (DOC) after Lydersen *et al.* (2004). Long-term normals for catchment runoff (Q) were estimated from meteorological data and soil properties using a model similar to MetHyd (a meteo-hydrological model; Slootweg *et al.*,2010). Long-term (1961–90) average monthly temperature, precipitation and cloudiness were derived from a $0.5° \times 0.5°$ global database (Mitchell *et al.*, 2004). Default net mass transfer coefficients for N (6.5 m $a^{-1}$) and S (0.5 m $a^{-1}$) were applied to all lakes (Kaste and Dillon, 2003; Baker and Brezonik, 1988). Nitrogen immobilization in catchment soils was set at 0.5 kg N $ha^{-1}$ $yr^{-1}$ (35.7 eq $ha^{-1}$ $yr^{-1}$) following the Mapping Manual (CLRTAP 2004). The denitrification fraction in the catchment soils was estimated as $f_{de} = 0.1 + 0.7 \cdot f_{peat}$, where $f_{peat}$ is the fraction of wetlands in the terrestrial catchment; landcover fractions of peat were obtained from the 2010 USGS North American Landcover database (USGS, 2013). Nitrogen removal in harvested biomass was estimated using biomass and species composition obtained from the National Forest Inventory (Beaudoin *et al.*, 2014) in combination with nutrient concentrations from the Canadian Tree Nutrient Database (Paré *et al.*, 2012) and the Tree Chemistry Database (Pardo *et al.*, 2005).

The resulting *CL(A), CL$_{max}$(S),CL$_{max}$(N) and CL$_{min}$(N)* maps created using the above data (Figure 4) cover much of the same region as depicted in Figure 2(a). Figures 2, 3 and 4 have matching colour scales, showing the relative sensitivity of the different ecosystems estimated using the critical load calculation protocols employed in each data set. The lakes and



aquatic ecosystem data, shown in Figure 2(a) and Figure 4, are in general more sensitive to acidifying deposition than the forest (Figure 2(b)) and terrestrial ecosystem data (Figure 3), a theme which recurs in our subsequent calculation of critical load exceedances (Section 3.6).

**Figure 4.** **Critical loads of acidity with respect to $S_{dep}$ ($CL(A)$), and $S_{dep}+N_{dep}$, for aquatic ecosystems (eq ha$^{-1}$ yr$^{-1}$). (a) $CL(A)$ (b) $CL_{max}(S)$. (c) $CL_{max}(N)$. (d) $CL_{min}(N)$.**

## 2.2 Global Environmental Multiscale – Modelling Air-quality and CHemistry (GEM-MACH) , Version 2

### 2.2.1 GEM-MACH v2 Overview

GEM-MACH is Environment and Climate Change Canada's comprehensive chemical reaction transport model. The model follows the on-line paradigm (in that atmospheric chemistry modules have been incorporated directly into a weather forecast model (GEM) (Moran *et al.*, 2010; Makar *et al.*, 2015 (a,b)). The parameterizations include gas-phase chemistry (42



species, ADOM-II mechanism, Stockwell *et al.*, 1989), aerosol microphysics (Gong *et al.*, 2003(a,b)) and cloud processing of gases and aerosols including uptake and wet deposition (Gong *et al.*, 2006; Gong *et al.*, 2015). The model's aerosol size distribution makes use of the sectional (bin) approach, with two possible configurations: (1) a processing-time efficient two-bin configuration used for operational forecasting and longer scenario simulations (fine and coarse particle sizes are

subdivided within certain aerosol microphysics processes in order to preserve solution accuracy while minimizing advective transport time) and (2) a more detailed 12 bin size distribution used to more accurately simulate aerosol microphysics and the size spectrum of particles. The aerosols in GEM-MACH are also speciated chemically into particle sulphate, nitrate, ammonium, primary organic aerosol, secondary organic aerosol, elemental (aka "black") carbon, sea-salt and crustal material, within each size bin. The crustal material component includes all particulate matter not speciated under the other

components, and hence includes base cations as a fraction of its total mass. As will be discussed below, the observations of Wang *et al.* (2015) were used to approximate the base cation fraction of GEM-MACH's crustal material, and hence estimate the mass of base cation deposition predicted by the model.

A comparison of GEM-MACH version 1.5.1 against other peer on-line models appears elsewhere (Makar *et al.*, 2015(a,b)), as does a description of the main updates associated with version 2 of the model (Makar *et al.*, 2017). Comparisons of the

operational 2-bin version of the model against observations have also appeared in the literature (Pavlovic *et al.*, 2016; Munoz-Alpizar *et al.*, 2017). Our description below will focus on the model's modules for gas-phase dry deposition, particle phase dry deposition, cloud processing and aqueous phase chemistry (wet deposition)..

### 2.2.2 Gas-phase dry deposition in GEM-MACH

A detailed description of the gas-phase dry deposition module of GEM-MACH (with an emphasis on the chemical species

which contribute to $S_{dep}$ and $N_{dep}$) appears in the Supplementary Information; here we provide an overview. Gas-phase deposition is handled using the commonly used "resistance" approach, where the deposition velocity is the inverse of the sum of aerodynamic, quasi-laminar sublayer and net canopy resistances. The aerodynamic resistance is the same for all gases, the quasi-laminar sublayer resistance depends on gas diffusivity, but these terms are relatively minor compared to the net canopy resistance, which tends to control the deposition velocity for many of the gases (notable exceptions being $HNO_3$

and $NH_3$ which have a relatively low canopy resistance). The net canopy resistance follows the approach of Wesely (1989) and Jarvis (1976), with terms for the resistance contributions associated with deposition to plant stomata, mesophyll, and cuticles, the resistance of gases to buoyant convection, the resistance associated with leaves, twigs, bark and other exposed surfaces in the vegetated canopy, the resistance associated with the height and density of the vegetated canopy, and the resistance associated with soil, leaf litter, etc., at the surface. The net canopy resistance includes a term to account for the

impact of precipitation and high humidity on stomatal and mesophyll resistances, and a temperature-dependent correction term for snow-covered surfaces.

Soil resistances are calculated following Wesely (1989) with a parameterization based on the values for $SO_2$ and $O_3$, with a seasonal dependence (Midsummer, Autumn, Late Autumn, Winter and Transitional spring). Canopy resistances are based



on Zhang *et al.* (2003), with the same seasonality as above. The resistance for the lower canopy follows Wesely (1989) using a function of the effective Henry's law constant and terms for $SO_2$ and $O_3$ resistances. The mesophyll and cuticle resistances follow Wesely *et al.* (1989), with seasonal variations as above and vegetation-dependent leaf area index values. The resistance of gases to buoyant convection follows Wesely *et al.* (1989), and is a function of the visible solar radiation.

The stomatal resistance follows a similar approach to Jarvis (1976), Zhang *et al.* (2002, 2003), Baldocchi *et al.* (1987), and ValMartin *et al.* (2014), and results from several terms describing its dependence on light ($k_s(Q_p)$), water vapour pressure deficit ($k_s(\delta e)$), temperature ($k_{st}$), $CO_2$ concentration ($k_{sca}$), the leaf area index (*LAI*) , and the ratio of the molecular diffusivities of water to the gas being deposited ($\frac{D_{H2O}}{D_{gas}}$). The approach taken for the dependence on light provides stomatal resistance values similar to those of Baldocchi *et al.* (1987), but are lower than those of Zhang *et al.* (2002) for the same

vegetation types, decreasing stomatal resistances and thus increasing the stomatal resistance contribution to deposition velocities, relative to Zhang *et al.* (2002). The other terms in the stomatal resistance employed curve fitting where possible across different sources of deposition data, due to the wide variation noted in the underlying measurement literature.

Deposition velocities are calculated for the $S_{dep}$ and $N_{dep}$ contributing gases $SO_2$, $H_2SO_4$, NO, $NO_2$, $HNO_3$, PAN, HONO, $NH_3$, organic nitrates, as well as several other transported gases of the ADOM-II gas phase mechanism. We note that the

rapid conversion of gaseous sulphuric acid ($H_2SO_4$) to particulate sulphate due to its low vapour pressure ensures that the direct contribution of $H_2SO_4$ deposition to $S_{dep}$ is relatively minor. Further details on the deposition velocity formulation, and tabulated coefficients for the species contributing to $S_{dep}$ and $N_{dep}$, appear in the Supplementary Information..

Gas-phase dry deposition velocities are incorporated as a flux lower boundary condition in the solution of the vertical diffusion equation within GEM-MACH.

### 20   2.2.3 Particle phase dry deposition in GEM-MACH

Particle dry deposition in GEM-MACH makes use of the size-segregated formulation of Zhang *et al.* (2001), which in turn follows Slinn (1982). The gravitational settling velocity (a function of the particle density, wet diameter, air viscosity, and the temperature and air pressure) is calculated for each particle size at each model level. At the lowest level, the settling velocity is added to the inverse of the sum of the aerodynamic resistance above the canopy and the surface resistance. The

aerodynamic resistance is a function of atmospheric stability, surface roughness, and the friction velocity, while the surface resistance is the inverse of the sum of collection efficiencies for Brownian diffusion, impaction and interception, multiplied by correction factors to account for the fraction of particles which stick to the surface. The Brownian diffusion is a function of the Schmidt number of the particle (ratio of the kinematic viscosity of the air to the particle's Brownian diffusivity). The impaction term is dependent on the Stokes number (itself a function of the gravitational settling velocity) and the land-use

type, and the interception term is taken to be a simple function of the particle diameter and a land-use and seasonal dependent characteristic radius.



The resulting deposition velocities have the characteristic strong dependence on particle size noted in observations, with minimum deposition values occurring at particle diameters of about 1 μm, with an increase in deposition velocities of up to two orders of magnitude with decreasing or increasing particle size. As will be discussed later in this work, one of the consequences of the size-dependence of particle deposition velocity is that particles which are larger (or smaller) than 1 μm

diameter settle more rapidly than the latter particles, and hence have shorter transport distances than 1 μm diameter particles. This phenomena is responsible for the rapid decrease in surface deposition with increasing distance from sources of base cations.

Particle gravitational settling and deposition velocities are handled in this version of GEM-MACH using a semi-Lagrangian

advection approach in the vertical for each column; vertical backtrajectories are calculated from the settling and deposition velocities, and mass-conservative interpolation is used to determine the new concentration profile and the flux to the surface. The particle deposition component of $S_{dep}$ and $N_{dep}$ (via the deposition of particle sulphate, particle nitrate, and particle ammonium) is typically very small compared to the gaseous dry deposition of primary emitted gases ($SO_2$, $NO_2$, $NH_3$), secondary gases ($HNO_3$), and wet deposition of ions ($HSO_3^-$, $SO_4^{2-}$, $NO_3^-$, $NH_4^+$).

**2.2.4 Cloud processing of gases and aerosols, and inorganic particle chemistry in GEM-MACH**

The cloud chemistry and aqueous processing of gases and aerosols in GEM-MACH makes use of the methodologies used in GEM-MACH's precursor model, A Unified Regional Air-quality Modelling System (AURAMS), and are described in detail in Gong *et al.* (2006). Aqueous chemistry includes the transfer of gaseous $SO_2$, $O_3$, $H_2O_2$, ROOH, $HNO_3$, $NH_3$ and $CO_2$ to cloud droplets, along with the oxidation of S(IV) to S(VI) within the cloud droplets by several pathways. The stiff system of

equations described by the aqueous chemistry is solved using a bulk approach and a computationally efficient predictor-corrector algorithm. Aerosol sulphate, nitrate and ammonium may be taken up into cloud droplets following activation, and may be returned to the aerosol phase following aqueous chemistry via particle evaporation. Rebinning of mass transferred back to the particle phase is accomplished through a mass-conservative rebinning algorithm similar to that described in Jacobson (2003).

Wet deposition processes (tracer transfer from cloud droplets to raindrops, scavenging of aerosols and soluble gases by falling hydrometers, downward transport by precipitation, and evaporation of raindrops and potential loss of mass prior to deposition) are explicitly included in GEM-MACH. Cloud droplet to raindrop tracer transfer is handled using a bulk autoconversion rate obtained from the meteorological model. Impact scavenging of size-resolved aerosols is parameterized using a scavenging rate based on the precipitation rate and the mean collision efficiency. Irreversible cavenging of soluble

gases makes use of the Sherwood number and diffusivity of the gas, the precipitation rate, the Reynolds and Schmidt numbers, and the raindrop diameter, while reversible scavenging makes use of equilibrium partitioning.

The cloud fields provided to the aqueous phase chemistry module depend on the model resolution – for the high resolution simulations carried out here, the hydrometeors are explicitly simulated and transported using the 2-moment scheme of



Milbrandt and Yao (2005 (a,b)). A full description of the cloud processing model and the formulation of its components appears in Gong *et al.* (2006).

Inorganic particle chemistry makes use of the HETV system of equations for sulphate, nitrate and ammonium described in detail in Makar *et al.* (2003), based on the ISORROPIA algorithms of Nenes *et al.* (1999). The concentrations of particle

sulphate, nitrate, ammonium, and gaseous $NH_3$ and $HNO_3$ are solved in bulk for non-ideal high concentration solutions via first determining the chemical subspace in which the total nitrate, sulphate, ammonium and relative humidity resides (breaking the problem into twelve subspaces for the different combinations of gases, salts, and aqueous ions which may exist under those conditions), then solving a double iteration including the full system of equations incorporating activity coefficient calculations and vectorization across the subspaces for computational efficiency. Following the bulk

calculations, the resulting aerosol mass of sulphate, nitrate and ammonium are rebinned using an approach similar to that of Gong *et al.* (2006).

### 2.2.5 Emissions and Simulation Setup

The emissions used in the simulations carried out here are described in detail in Zhang *et al.* (2017, this special issue).

All simulations used a nested model setup, feeding into the meteorological and chemical boundary conditions for a 2.5km

resolution Alberta and Saskatchewan simulation. The latter domain is depicted in Figures 2 and 3 (the 2.5km resolution domain is the entire coloured and grey shaded region). Archived GEM 10km forecast simulations were driven by data assimilation analysis fields, and were used to in turn drive successive overlapping 30 hour forecasts of both a Canadian domain 2.5km resolution meteorological forecast, and a 10km GEM-MACH forecast. The final 24 hours of these simulations provided the meteorological and chemical boundary conditions respectively for a series of 24-hour simulations

of GEM-MACH on the domain shown in Figures 2 and 3. This nesting approach was selected to provide the best possible meteorological and chemical inputs for the 2.5km high resolution domain. The output from the 24-hour simulations were then brought together to create the continuous time record of concentrations and deposition on the high resolution model grid Three simulations were carried out with this setup. The first of these is made use of the two aerosol bin configuration of GEM-MACH, for an entire year of simulated chemistry and meteorology (August 1, 2013 to July 31, 2014), in order to

obtain a year of model output, required for critical load calculations. The outer 10km North American domain of the simulation made use of the operational GEM-MACH forecast emissions inventories for the years 2010 (Canada), 2011 (USA) and 1999 (Mexico), while the inner nest made use of 2013 (Canada) and 2011 (USA) inventories (see Zhang *et al.*, 2017). The predicted deposition thus represents the model predictions using emissions reported under current Canadian regulatory requirements. Two additional simulations were then carried out, for the period August 13[th] to September 10[th],

making use of the 12-bin version of the model: a base case and a primary particulate scenario. The primary particulate scenario made use of aircraft-observation-based estimates of primary particulate emissions from six oil sands facilities, and both making use of continuous emissions monitoring data for Alberta for $SO_2$ and $NO_x$ emissions from large stack sources (see Zhang *et al.,* 2017, this issue, for the full description of these emissions). This second pair of simulations was carried out to investigate the potential impact of possible under-reporting of primary particulate emissions on model critical load



exceedance predictions. About 96% of these primary particulate emissions by mass are associated with fugitive dust emissions sources, and over 68% of this mass is in the coarse mode (diameters greater than 2.5 μm) (Zhang *et al.*, 2017). The potential impact of these sources of base cations on acidifying deposition will be discussed in Sections 3.3, 3.5 and 3.6.

## 2.3 Deposition Observations

### 2.3.1 Deposition of ions in precipitation

Wet-only precipitation measurements were collected at six sites in Alberta (AB) by Alberta Environment and Parks and two sites in Saskatchewan (SK) by the Canadian Air and Precipitation Monitoring Network (CAPMoN) (Figure 2(b)). In wet-only samples, a heated precipitation sensor opens the collector lid when precipitation is detected, and closes the lid when precipitation ends. For the SK samples, the collector bucket was lined with a polyethylene bag which was removed, weighed,
sealed, refrigerated, and shipped to the laboratory for major ion analysis. For the AB samples, the samples were transferred from the clean collection bucket to a smaller sample bottle, capped, refrigerated if stored on site, and shipped to the laboratory for analysis. Collection occurred approximately daily at the SK sites and approximately weekly at the AB sites. Quality control was performed by the collecting networks.

Annual precipitation-weighted mean concentrations of $SO_4^{2-}$, $NO_3^-$ and $NH_4^+$ were calculated from the daily or
weekly samples using recommended methods and completeness criteria (WMO/GAW, 2004, 2015) and the resulting deposition fluxes were compared with model values. Where there were measurement gaps of > 3 weeks (two sites), or where there was only partial coverage of the 12 months (one site), fluxes were compared over shorter measurement periods. The collector buckets described above tend to underestimate the total precipitation, so the flux of ions derived from their records must be corrected using independent observations of total precipitation. At the SK sites, separate on-site rain and snow
gauges were used to manually record the daily precipitation amount. At the AB sites, precipitation gauges for independent quantification of total precipitation were not available, hence weekly deposition fluxes were calculated using daily precipitation depth data from the nearest meteorological station, or combination of meteorological stations, with the most complete coverage (ECCC, 2017, AAF, 2017).

Total precipitation depth collected in the AB wet deposition collectors, summed over all collection periods at the
sites, was 51-96% of the estimated precipitation depth at meteorological stations. Our deposition flux calculations implicitly assume that the ion concentrations measured in the sample are representative of all the precipitation during the period. However, the mechanism of precipitation loss (undercatch due to wind, evaporative loss, delay in lid opening) may lead to unrepresentative concentration values. Additional uncertainty is introduced by the use of precipitation depth from collectors that are not co-located, particularly at Kananaskis. Therefore, wet deposition fluxes from the AB sites have higher
uncertainty than the fluxes at the two SK sites, where 105% and 78% of the standard gauge precipitation was captured by the collector.



### 2.3.2 Deposition of S and N compounds to snowpack

Observations of total deposition of sulphur, nitrogen, and base cations to snow-covered open surfaces were collected in two separate studies. Samples were collected in the immediate vicinity of the oil sands by Environment and Climate Change Canada, and snowpack samples in northern Saskatchewan were collected by Saskatchewan Environment (snowpack station locations are discussed in Section 3.4). Both sets of data were collected in open clearings and thus deposition is to snow-covered *open* surfaces. They thus provide *minimum* estimates of deposition, particularly for gases. One method of accounting for deposition to forests and related vegetation is via collection of precipitation samples below foliage, which assumes that deposited materials leaves the vegetation via precipitation and/or melting of snow, to reach the collector ("throughfall"). Watmough *et al.* (2014) compared winter throughfall versus open deposition in the oil sands region, and showed maximum throughfall values to be about 1.9 times their open deposition counterparts. However, throughfall observations do not account for the portion of the deposited material which remains on or within the vegetated surfaces, and hence must also be considered a conservative estimate of total deposition. Using the algorithms of GEM-MACH's gas-phase deposition module, typical ratios of dry deposition velocity between a needle leaf forest and an open snow covered surface for $SO_2$ and $NH_3$ respectively are 2.63 and 1.97 (temperature -5C, $u* = 0.1$ m s$^{-1}$, solar radiation = 100 W m-2, z0 = 0.1m, Monin-Obukhov length = 50). However, the ratios for dry deposition of particles with diameters of 2.5 and 10 μm are 0.76 and 0.82, respectively (Zhang *et al.*, 2001), indicating that the open snowpack observations may slightly overestimate $BC_{dep}$ and $Bc_{dep}$ (in contrast to the Watmough *et al.* 2015 observations), but significantly underestimate $S_{dep}$ and $N_{dep}$.

### 2.3.2.1 Environment and Climate Change Canada oil sands snowpack sampling

*Snowpack sampling:* To assess winter-time atmospheric deposition of acidifying emissions to the oil sands region, snowpack samples were collected at varying distances from the major oil sands development area in early spring 2014 (n=130), as well as at 9-12 sites in the Peace-Athabasca River Delta (PAD) located ~200 km north of the major oil sands developments (See Figure 9(a)). Based on historical snow accumulation data for the Fort McMurray region (Environment-Canada. National Climate Data and Information Archive), the onset of permanent freezing began on November 6th of 2013, and all samples were collected within an 8 day period ending March 6th, to ensure maximum snowpack depth and minimize snowpack alterations over the course of sampling. Sampling sites were accessed by helicopter, and snow samples were collected at 50-100 m upwind of landing sites, to reduce potential contamination by helicopter exhaust. Stainless steel tools used snow collection were acid-washed prior to use in the field and a standard two-person protocol was used to minimize potential contamination. Snow pits were dug to the bottom of the snowpack using a stainless steel shovel and 10 cm diameter custom-made stainless steel corers and stainless steel spatulas were used for snow collection, which ensured an even sampling of the complete snowpack profile. Snow sampling equipment was cleaned with snow at each site prior to collection. Snow for water chemistry and multi-element analyses was collected into 13 L pre-cleaned high density polypropylene pails. The weight and depth of ~10 cores was recorded at each site for further determination of snow water equivalence (SWE). After





collection, snow was kept frozen until processing at the Canada Centre for Inland Waters (CCIW), in Burlington, Ontario, Canada.

As part of QA/QC protocol, field blanks and duplicate (5% of sites) and triplicate samples (1% sites) were also collected at random sites.

*Water Chemistry analysis***:** Snow was analyzed for water chemistry parameters, including pH, major cations, anions, and sulphate, and 45 elements including crustal elements, following standard procedures at the National Laboratory for Environmental Testing (NLET) in Burlington, Ontario, Canada. NLET is a certified member of the Canadian Association for Environmental Analytical Laboratories (CAEAL) and undergoes regular external reviews to maintain this accreditation.

*Snow-Water Equivalent and Loadings***:** Average SWE were determined at each site and used to calculate loadings as described previously (Kelly *et al.* (2009), Kelly *et al.* (2010), Kirk *et al.* (2014), Manzaono *et al.* (2016)). Briefly, equations [16], [17] and [18] were used to calculate SWE and Aerial Water Volume (AWV), which was used to obtain sulphate, nitrate, ammonium and base cations loadings, depending on the parameter used in equation (18):

$$SWE \left(\frac{kg}{m^2}\right) = \frac{snowpack\ weight\ (kg)}{\pi[corer\ radius\ (m)]^2} \tag{16}$$

$$AWV \left(\frac{L}{m^2}\right) = \frac{SWE\left(\frac{kg}{m^2}\right)}{\rho_{water}\left(\frac{kg}{m^3}\right) \times \frac{1}{10^3}\left(\frac{m^3}{L}\right)} \tag{17}$$

$$\sum Ion\ loading \left(\frac{\mu g}{m^2}\right) = AWV \left(\frac{L}{m^2}\right) \times \sum Ion\ concentration \left(\frac{\mu g}{L}\right) \tag{18}$$

20 **2.3.2.2 Saskatchewan Environment Snow Sampling**

*Sampling Design:* Snowpack surveys were conducted by Saskatchewan Ministry of Environment at 18 sites during Feb 16[th] - 22[nd] (n=13) and Apr 1[st] - 2[nd] (n=5), of 2014. Snow cores were collected at the centre of frozen lakes to minimize the influence of trees and topography on deposition and chemistry. Distances from the approximate centre of emissions in the Athabasca Oil Sands operations ranged from 106-291 km. Site selection in 2014 was based on criteria used to select lakes 25 for a sediment coring study, described by Laird *et al.*, (2017).

*Sample collection*

Sampling equipment was cleaned with snow at each site prior to collection. Multiples (10-20) of intact snow cores were collected using an acrylic 7.62 cm diameter tube and a stainless steel spatula and composited into Teflon[TM] bags. Snow water equivalents (SWE) and snowpack loadings were determined as described for the ECCC snowpack samples, above.
30 Samples were delivered frozen to the Biogeochemistry Analytical Laboratory at the University of Alberta, Edmonton.





*Water Chemistry analysis:* Samples were melted in a temperature controlled clean room and stirred prior to filtration (0.7 μm). Total and dissolved base cations were measured by ICP-MS. Other ions (ammonium, nitrate, sulphate, chloride), dissolved organic carbon, conductivity, pH, and Gran alkalinity were measured according to standard methods. Acceptance criterion of ±15% was applied to analytical charge balance.

## 3. Results
### 3.1 GEM-MACH estimates of annual $S_{dep}$ and $N_{dep}$

The model estimates of total $S_{dep}$ and $N_{dep}$ (eq ha$^{-1}$ yr$^{-1}$), along with the percentage contribution of the different resolved

10 components of sulphur and nitrogen deposition, are shown in Figures 5 and 6, respectively. The bulk of the total $S_{dep}$ close to the sources of emissions is due to dry deposition of $SO_2(g)$ and wet deposition of $HSO_3^-$, while the wet deposition of $SO_4^{(2-)}$ dominates in downwind regions. $N_{dep}$ near sources is dominated by dry deposition of $NO_2(g)$ and $NH_3(g)$ near sources and dry deposition of $HNO_3(g)$ and $NH_4^{(+)}$ further downwind. Figure 5 (b-e) and Figure 6 (b-i) show that for sites downwind of the source regions (hot-spots in panel (a) of these figures), wet deposition dominates. As we discuss below,

15 the model predictions correlate well with wet deposition observations at precipitation-monitoring stations located downwind of emissions sources, and these relationships allow for an approximate correction of model $S_{dep}$ and $N_{dep}$ estimates using observations.

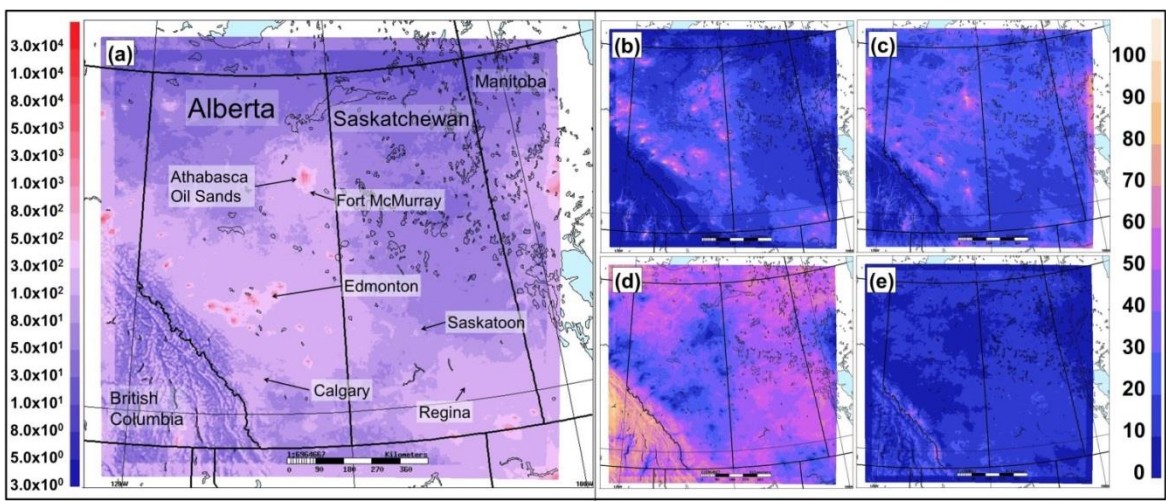

**Figure 5. GEM-MACH predictions of total Sulphur deposition and its speciation. (a) Total S deposition (eq ha$^{-1}$ yr$^{-1}$), and percentages of total S deposition due to: (b) SO$_2$ (dry), (c) HSO$_3^-$(aq) (wet), (d) SO$_4^{2-}$(aq) (wet), and (e) particulate sulphate (dry).**



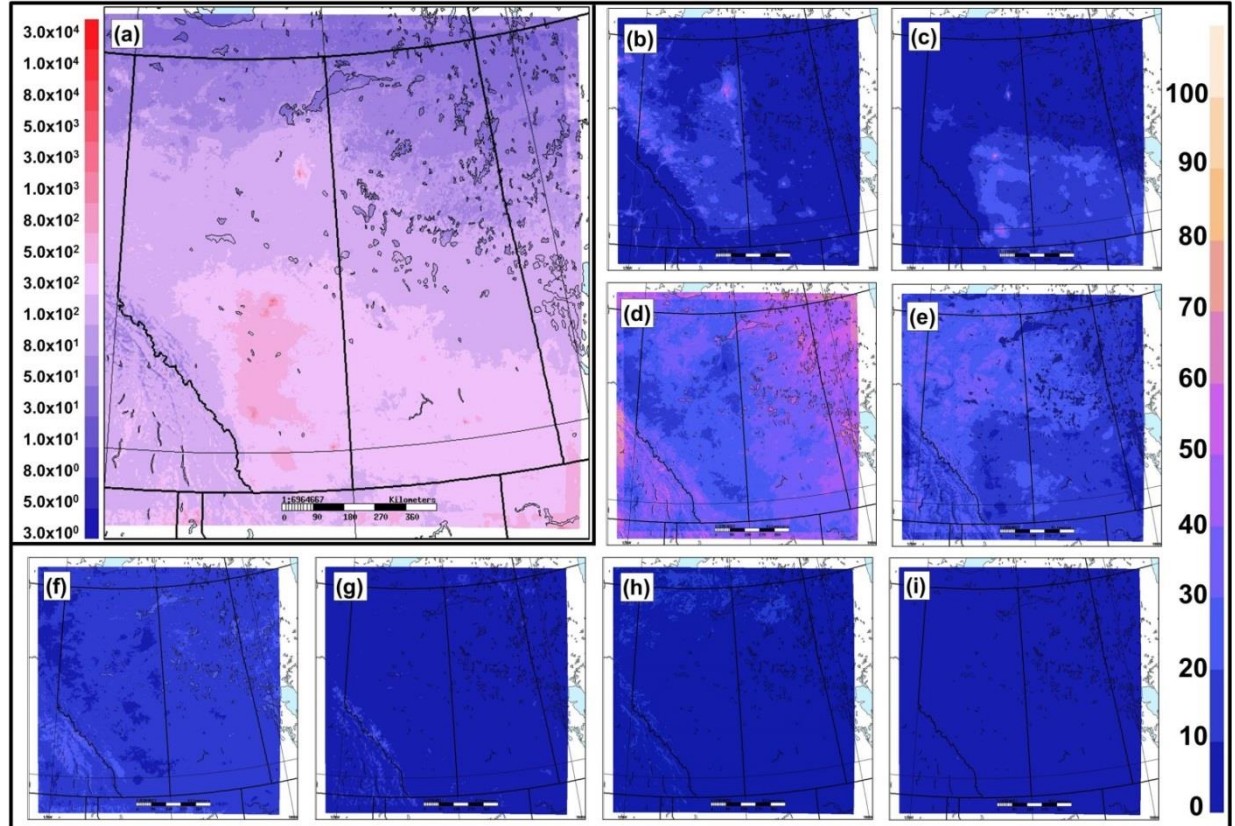

**Figure 6. GEM-MACH predictions of total Nitrogen deposition and its speciation.** (a) Total N deposition (eq ha$^{-1}$ yr$^{-1}$), percentages of total N deposition due to: (b) NO$_2$ (dry), (c) NH$_3$ (dry), (d) NH$_4^+$(aq) (wet), (e) HNO$_3$ (dry), (f) NO$_3^-$(aq) (wet), (g) particulate ammonium (dry), (h) peroxyactylnitrate (dry). (i) each of particulate nitrate (dry), gaseous organic nitrate (dry), NO (dry) and HONO (dry) (each contribute less than 10% to N$_{dep}$).

## 3.2 Model evaluation: wet deposition

The observed wet deposition of deposited sulphur (as SO$_4^{2-}$), nitrogen (NH$_4^+$ + NO$_3^-$), and base cations (sum of Ca$^{2+}$, Mg$^{2+}$, Na$^+$, and K$^+$) are compared to model estimates in Figure 7 (b), (c), and (d), respectively (station locations are shown in Figure 7(a)). Note that GEM-MACH's particle speciation includes a "crustal material" component, of which base cations are a component. The model wet and dry estimates of crustal material deposition were combined, and the fraction of crustal material which is composed of base cations was estimated from the observations of surface dust collected by Wang *et al.* (2015), in the vicinity of the oil sands, in order to estimate base cation deposition. Model estimates of deposited sulphur in precipitation were biased high, with a slope of 2.2, but the model accounts for most of the observed variation with a Pearson correlation coefficient (R$^2$) of 0.90. Model estimates of deposited nitrogen were biased slightly low (slope = 0.89, R$^2$ = 0.76), and the model estimate of base cations were biased low (slope = 0.40, R$^2$ = 0.72).

The positive bias in simulated sulphur deposition may reflect an underestimation of the SO$_2$ deposition flux closer to the sources (the precipitation sites are located far from SO$_2$ emissions sources; a model underestimate of upwind SO$_2$





deposition flux may thus result excess sulphur being transported downwind, increasing simulated wet deposition of sulphur at these downwind precipitation sites). The negative bias in simulated base cations may result from an underestimate in the model's input emissions for the "crustal material" component of primary particulate matter from either reported anthropogenic or natural sources (and/or in the base cation fraction of these emissions). We discuss the potential impact of

under-reporting to the National Pollutant Release Inventory (NPRI), below.    The deposition velocity of particulate matter is a strong function of particulate size, with submicron and supermicron particles having the highest and micrometer-sized particles having the lowest deposition velocities, respectively. The size distribution of particles thus determines their residence time prior to deposition, and hence errors in the spatial pattern of estimated $BC_{dep}$ may also reflect errors in the assumptions on the size distribution of emitted particles. Both of these possibilities are discussed further in Section 3.5.

The relatively high correlation for all three deposited quantities suggests that the linear relationships between model estimates and observed ions in precipitation may be used as a means of providing observation-corrected estimates of the $S_{dep}$, $N_{dep}$ and $BC_{dep}$ required for the critical load and critical load exceedance calculations described in Section 2. Therefore, critical load exceedances were calculated using the original model deposition for sulphur, nitrogen and base cations, and also using model deposition corrected using the model-observation linear relationships shown in Figure 7. We note that the

resulting corrected values may underestimate exceedances near the sources of $S_{dep}$ and $N_{dep}$ precursor species emissions. For example, if the positive bias in wet sulphur deposition of Figure 7(b) results from a model underestimate of dry deposition of $SO_2$ near its sources, an overall downwind correction of $SO_2$ as per Figure 7(b) may underestimate sulphur deposition from $SO_2$ near the sources. The resulting corrected values should thus be considered a lower bound for exceedances in the near-source environment.





**Figure 7. (a) Ions-in-total-precipitation sample collection sites. Scatterplots compare model and observed wet deposited (a) S, (b) N, and (c) base cations in precipitation (eq ha$^{-1}$).**

### 3.3 Estimates of primary particulate emissions and resulting $BC_{dep}$ from aircraft observations near the oil sands

An airborne measurement campaign was undertaken in August and September of 2013 in the Athabasca oil sands region as part of a broader measurement plan (the Joint Oil Sands Monitoring program) to characterize emitted air pollutants, determine the extent of subsequent atmospheric transport and chemical transformation, and support the improvement of air quality models and satellite column retrieval algorithms. "Enclosure" (box) flights were carried out around individual emitting facilities, in order to characterize their emissions fluxes. As part of that work, a mass balance model was developed (the Top-down Emission Rate Retrieval Algorithm, TERRA, Gordon *et al.*, 2015). TERRA makes use of aircraft flux data





and mass conservation equations to estimate emissions from facilities, and was shown to produce $SO_2$ emissions estimates which were within 5% of direct within-stack estimates from Continuous Emissions Monitoring. The algorithm has more recently been used to estimate the emissions fluxes of intermediate volatility organic compounds (Liggio *et al.*, 2016), volatile organic compounds (Li *et al.*, 2017) and the primary emissions of gaseous organic acids from these facilities (Liggio

*et al.*, 2017).

The TERRA algorithm, aircraft observations of total particulate matter number concentration and size close to the sources, and the fugitive dust speciation reported in Wang *et al.* (2015) were used to estimate fugitive dust emissions for six oil sands facilities, for the 12 particle bin version of the GEM-MACH model (Zhang *et al.*, 2017). We refer to these emissions and corrections to deposition based on them hereafter as "aircraft-observation-based". As shown in Zhang *et al.* (2017), the

aircraft-observation-based primary particulate emissions estimates are much higher (on average, by a factor of ten) than the values reported to the National Pollutant Release Inventory by the facilities, with 96% of the primary particulate emissions being associated with fugitive dust, and 68% of this mass being at particle sizes greater than 2.5 μm diameter. This in turn suggests that the primary particles may rapidly deposit with increasing distance from the emissions sources. The mean Wang *et al.* (2015) base cation fractions of primary particulate matter in the 0 to 2.5 μm particle diameter size range and the

2.5 μm to 10 μm particle size ranges were found to be quite similar; we have used the former here, to describe the mass fraction of the aircraft primary particulate emissions assumed to be composed of base cations. While we have used the reported emissions inventory in annual acid deposition modelling, this comparison between the inventory and the aircraft emissions estimates suggests that former may significantly underestimate the $BC_{dep}$ and $Bc_{dep}$ terms used in critical load and critical load exceedance estimates.

The potential impact of higher-than-reported primary particulate emissions on the estimation of base cation deposition was investigated here via two 29 day simulations of the 12-bin version of GEM-MACH, employing the reported emissions versus the aircraft-observation-based estimates. The ratio of gridded net model wet and dry deposition of "crustal material" between the two simulations was calculated. Figure 8 shows the average value of this ratio, derived from sampling the resulting gridded field at 10 km distance and 20 degree angles about a reference point within the oil sands emissions area,

out to 600km distance. As noted above, most of the primary particulate matter in the aircraft-observation-based emissions resides in the coarse mode (particle sizes greater than 2.5 μm). These larger particles have higher deposition velocities and consequently undergo rapid deposition close to the sources. The use of the aircraft-observation-based emissions thus results in enhancements in crustal material deposition relative to the reported emissions simulation, of a factor of 11 close to the sources. The ratio drops exponentially with distance from the sources, and shows the impact of the size fractionation

observed from the aircraft data. A combination of exponential decay functions (see Figure 8) was found to fit the average ratio to a very high correlation ($r^2$=0.998). Zhang *et al.* (2017) used the observations of Wang *et al.* (2015) to show that 93% of the primary particulate matter emissions were composed of crustal material. Wang *et al.* (2015) also includes the relative fraction of base cations within these particles. The exponential decay function thus describes the average relative




enhancement of crustal material (and hence base cation) deposition, associated with the use of the aircraft-observation-based

primary particulate emissions, relative to the reported values.

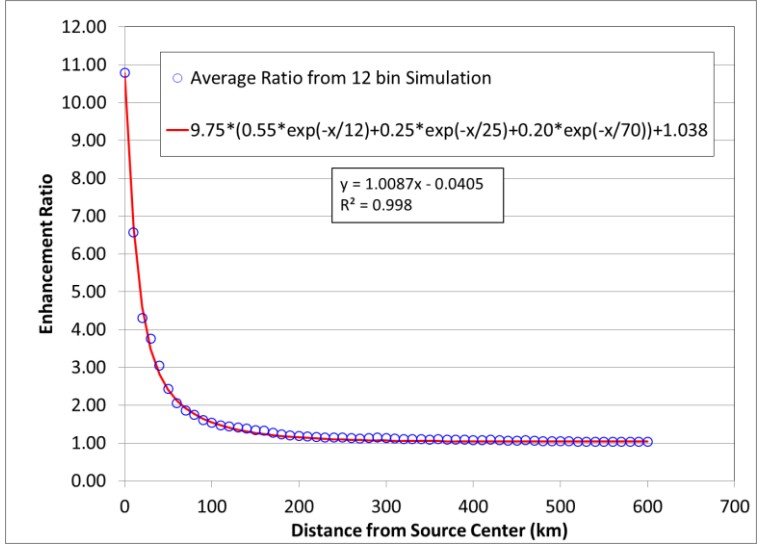

**Figure 8. Temporal and spatial average ratio of total deposited crustal material as a function of distance from a reference point**
**within the oil sands emissions region: ratio of deposition from the model simulation using aircraft-observation-based primary**
**particulate emissions to the model simulation using reported fugitive dust emissions.**

Figure 8 shows that the additional fugitive dust emissions result in a substantial enhancement in crustal material (hence base

cation) deposition close to the sources, but this enhancement approaches only 3.8% further downwind due to size-segregated

particle deposition en-route, with the more rapid deposition of super-micron sized particles. This result was expected, given

that the aircraft observations showed that 93% of the emitted primary $PM_{10}$ mass resides in particle sizes greater than 2 μm

diameter. Particle deposition velocities have a well-established size-dependence (cf. Wesely *et al.*, 1985; Zhang *et al.*,

2001), with a rapid increase in deposition velocities occurring for particles with diameters between 1 and 10 um (a factor of

28.6 between these two particle diameters, for particle deposition to Needle-leaf trees and a wind speed of 2 m s$^{-1}$, Zhang *et*

*al.*, 2001).

While the reported fugitive dust emissions in the reported inventory were used in the 2-bin annual GEM-MACH simulations

carried out here, the aircraft-observation-based emission estimates and the shorter duration model simulation described

above suggest that the primary particulate emissions in the reported inventory may greatly underestimate the base cation

deposition. The scaling function shown in Figure 8 along with the correction to downwind base cation observations from the

precipitation data shown in Figure 5(d) were therefore used to create a combined corrected estimate of base cation

deposition. We note that this combined estimate would increase base cation deposition by a factor of ~25 in the immediate

vicinity of the oil sands operations, and drop off to a factor of 2.5 further downwind. However, as is shown in the next




section, the use of this correction on the original model estimates improves both the correlation and the slope of the model-derived estimates of $S_{dep}$, $N_{dep}$ and $BC_{dep}$ relative to observations of winter deposition to snow.

## 3.4 Comparison of model and observed snowpack deposition

The observed snowpack-derived deposition fluxes are compared to the modelled values for total sulphur, nitrogen and base cations in Figure 9 (b), (c), and (d), respectively (site locations are shown in Figure 9(a)). The uncorrected model and observation pairs for each site are shown in blue for each of these figures. The model slopes for sulphur and nitrogen are relatively high and correlations relatively low in comparison to the total deposition in precipitation comparisons carried out at stations further downwind (compares Figures 7 and 9). The model values however represent the total deposition to all

surface types within each model grid-square, while the snowpack observations correspond to values in forested clearings; thus, as noted in Section 2.3.2, the snowpack observations may underestimate the total deposition by a factor of 2.6 ($S_{dep}$) and 2.0 ($N_{dep}$).

The nitrogen deposition (Figure 9(b)) is dominated by deposition of ammonium, and other work (Whaley *et al.*, 2017), has

found that model overestimates of surface concentrations of ammonia in the immediate vicinity of oil sands emissions sources likely result from incomplete stack information for the relevant facilities' ammonia emissions records (missing volume flow rates, temperatures). In the absence of this information, default EPA values for stacks are used in the emissions processing, which likely underestimates the vertical dispersion of emitted ammonia; see Whaley *et al.* (2017), Zhang *et al.* (2017)).

The model estimates of base cation deposition to snowpack have a strong negative bias (slope = 0.05, R$^2$ = 0.22). This bias is considerably stronger than noted for the precipitation sites further downwind (compare Figure 9(d) to Figure 7(d)). The additional bias is likely due to the under-reporting of primary particulate emissions in the emissions inventories.

Purple lines and symbols on Figures 9(b), and 9(c) depict the relationships between modelled and open snowpack $S_{dep}$ and $N_{dep}$ loads, when the latter are corrected to approximate throughfall values using the model-derived $SO_2$ and $NH_3$ deposition velocity ratios for needle-leaf forest to open snow-covered surfaces. These corrections result in a considerable improvement to the slope between model-derived and snowpack $S_{dep}$, fluxes, and the apparent $N_{dep}$ overestimate is halved.

Green lines and symbols on Figure 9(d) compare model values of $BC_{dep}$ corrected by the combination of precipitation and aircraft-observation-based scaling factors described earlier, to the observations, which are also corrected using the expected ratio of needle-leaf forest to open snow covered particle deposition velocities. Red symbols and lines indicate the fit occurring when only the model values are corrected. The correction of modelled values improves both the slope and correlation coefficient of the best fit line, while correction of observations for the expected influence of snowfall versus





snowpack further improves the slope.    We note that the combination of precipitation and aircraft-observation-based correction factors on the model's original estimates of base cation deposition increase that estimate by a factor of 25, yet result in a substantial improvement to the fit and slope relative to observations.    These results suggest that the primary particulate emissions resulting from aircraft observations is an underestimate, and/or that the base cation mass fraction

5   derived from collection of deposited surface dust (Wang *et al.*, 2015) is biased low relative to fugitive dust in the atmosphere in this region.  Further observation flights are planned for the spring and summer of 2018 to sample both base cation mass fractionation and particulate size distribution in order to further improve estimates of base cation emissions from oil sands operations and other sources.

10                                                                          .



**Figure 9** **(a) Snowpack sample collection sites (purple: Environment and Climate Change Canada sampling sites; orange: Saskatchewan Environment sampling sites). (b), (c), (d): Relationships between modelled and snowpack-derived $S_{dep}$, $N_{dep}$ and $BC_{dep}$, fluxes, respectively. Blue lines: uncorrected model estimates compared to uncorrected snowpack observations. Red lines: Model estimates corrected using downwind precipitation observations (b,c,d) and aircraft-obervation-based fugitive dust emissions estimates (d). Purple lines: original model values compared to snowpack-derived loads corrected by the expected ratios of throughfall to open surface collection for $S_{dep}$ (b) and $N_{dep}$ (c). Green line (d): model $BC_{dep}$ estimates scaled using precipitation and aircraft observations paired with observations corrected by the expected ratio of throughfall to open surface collection for $PM_{2.5}$. Units are eq ha$^{-1}$ for the snowpack sampling periods; model values are the sum of hourly values during snowpack sampling times.**

## 3.5 Comparison of base cation fluxes

Given the dependence of critical loads on base cation levels in both terrestrial and aquatic ecosystems, we compare the observation-based base cation catchment export from aquatic ecosystems (Figure 10(a)) to three different estimates of base cation fluxes used in the subsequent critical load exceedance calculations. Figure 10(a) is equivalent to the sum of atmospheric deposition, soil weathering, soil cation exchange and groundwater contributions within catchment water, and consequently has larger values than the remaining three estimates, which depict different estimates of the atmospheric component ($BC_{dep}$). Figure 10(b) shows the $BC_{dep}$ values estimated via interpolation and extrapolation of Canada-wide observation station data collected between 1994 through 1998, with the observation stations within GEM-MACH's 2.5km domain shown as diamond symbols. Figure 10(c) shows the original GEM-MACH-derived base cation deposition (using the reported fugitive dust emissions, the model's summed wet and dry crustal material deposition, and the Wang *et al.* (2015) base cation fractionation reported above). Figure 10(d) shows the base cation deposition fields resulting from correcting the model values of Figure 10(c) with the precipitation-observation-based and aircraft-observation-based emission scaling factors of Figures 7(c) and 8, and represent an observation-corrected estimate of base cation deposition. We note that the observation stations of Figure 10(c) measure only the wet component of base cation deposition. However, model calculations show that the dry particulate matter flux of base cations drops off rapidly with distance from the sources. The precipitation sites are intended as background sites, located far from sources, and the bulk of base cation deposition at these locations is expected to be via wet deposition.

Three important features should be noted from Figure 10.

First, the net base cation flux exported from aquatic ecosystem catchments (Figure 10(a), data described in Section 2.1.4) is usually much larger than any of the three estimates of $BC_{dep}$ in the remaining panels of the Figure. This implies that the aquatic ecosystem base cation load is usually dominated by soil weathering, soil cation exchange and groundwater inputs. The area of lowest cation flux exported from aquatic systems is observed in north-west Saskatchewan,



Second, the observation-derived estimates of $BC_{dep}$ derived from sparse measurement station data, at station locations designed to be relatively remote from sources (Figure 10(b)) are relatively spatially homogeneous compared to the two remaining $BC_{dep}$ estimates, which are derived from model estimates of crustal material emissions. However, the model results suggest that these station locations may consequently miss much of the base cation deposition associated with large

5  sources of fugitive dust emissions, which is highly localized. The largest values in the model estimates are in close proximity to the anthropogenic sources (Figure 10(c,d)). The latter show a rapid drop-off of estimated base cations with distance from the sources, as was expected from Figure 8. Within these anthropogenic emission "hot-spots" of Figure 10(c,d), $BC_{dep}$ estimates reach as high as $3 \times 10^4$ eq ha$^{-1}$yr$^{-1}$, compared to background levels in the 10's of eq ha$^{-1}$ yr$^{-1}$ (note that the colour scale on Figure 10 is logarithmic).

Third, the corrections applied to Figure 10(c), to create the combined aircraft-observation-based and precipitation-observation based corrected field of Figure 10(d), are in reasonable agreement with the 1994-1998 observation station values at the remote-from-sources observation station locations (diamond symbols, Figure 10(b)), and also reflect the increases of base cation deposition expected from the aircraft-observation-based fugitive dust emissions estimates in the immediate

15  vicinity of the oil sands. As noted in the previous section, these final estimates of $BC_{dep}$ also have a greater degree of agreement with snowpack observations of base cations in the immediate vicinity of the oil sands (Figure 9 (d)).



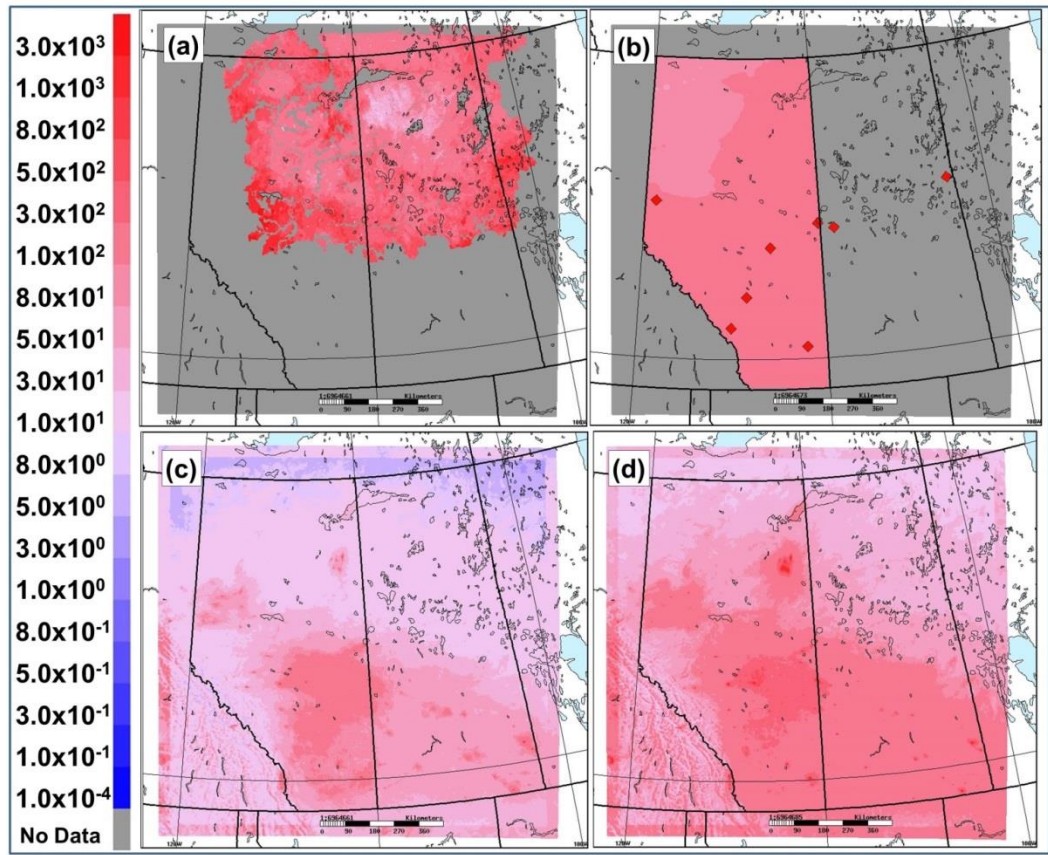

**Figure 10. Base cation fluxes (eq ha$^{-1}$ yr$^{-1}$). (a) Total export flux of base cations from aquatic ecosystem catchments. (b) Atmospheric deposition flux of base cations from surface data collected between 1994 through 1998, monitoring station locations shown as red diamonds, (c) base cation deposition from GEM-MACH, making use of Wang *et al.* (2015) speciation, (d) GEM-MACH $BC_{dep}$ corrected using and precipitation measurements and aircraft observations of fugitive dust.**

Watmough *et al.* (2014) presented observations within 135 km of the oil sands which compared $S_{dep}+N_{dep}$ to $BC_{dep}$. The base cations were found to be in excess of the $S_{dep}$ and $N_{dep}$, and hence one of their conclusions was that "despite extremely low soil base cation weathering rates in the region, the risk of soil acidification is mitigated to a large extent by high base cation deposition". However, the rapid decrease in base cation deposition with distance from the sources in Figures 10(c,d) and Figure 8 suggest that this neutralization effect may be limited with increasing distance from the sources of base cation emissions. We re-examined the summer throughfall data presented in Watmough *et al.* (2014), and show the excess in base cation deposition ($BC_{dep} - N_{dep} - S_{dep}$) as a function of distance from the oil sands emissions region in Figure 11. The data show a rapid decrease in neutralization with distance from sources in the oil sands region, with a linear best fit crossing the intercept, from neutralizing to non-neutralizing conditions, at a distance of 142 km. The data also show a wide variation within the 30 km central region, suggesting neutralization is not uniform. Both these observations (Figure 11) and the model estimates of $BC_{dep}$ (Figure 10(c,d)) thus suggest the neutralization impact of base cation deposition from oil sands




sources will be limited in spatial extent. A circle with radius 140 km around the oil sands emissions region appears on the maps of critical load exceedance in Section 3.6, to serve as a visual guideline of this observation-based cross-over distance between base cation neutralization and acidification.

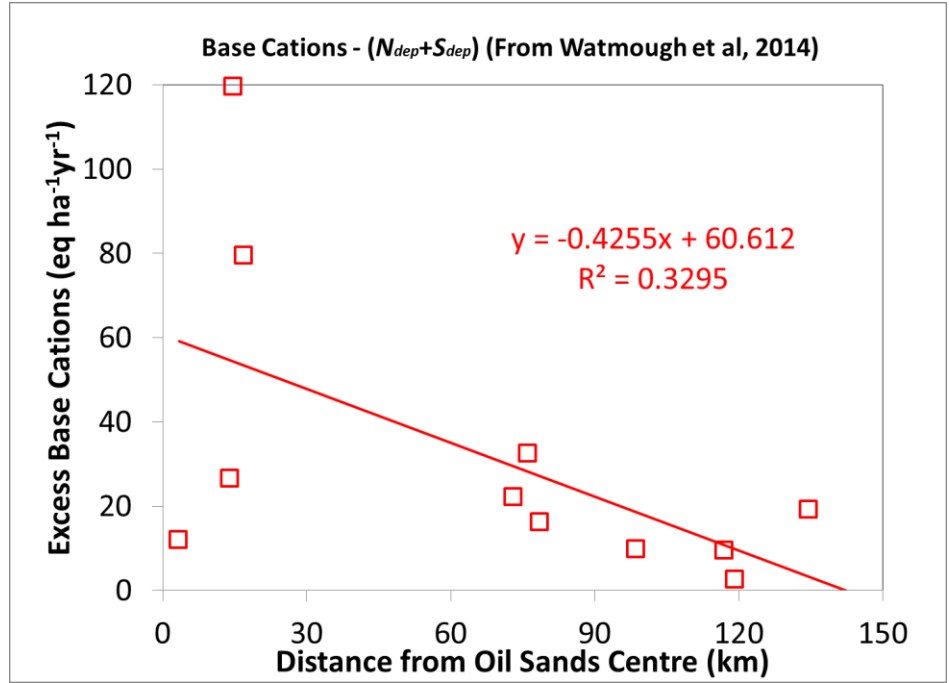

**Figure 11.** $BC_{dep} - N_{dep} - S_{dep}$, **using the data of Watmough *et al.* (2014).**

The estimates of $BC_{dep}$ from Figure 10 (b) and (d) are shown as ratios to the base cation catchment export flux (Figure 10(a)) in Figure 12 (a,b) respectively. The ratios are usually less than unity (blue shades) indicating that contributions aside from $BC_{dep}$ control the base cation budget, while regions where $BC_{dep}$ is greater than the observation-based total base cation export in catchment water (red shades) occur in the center of the oil sands region and in part of northern Saskatchewan. The latter indicate regions where atmospheric base cation deposition is expected to exceed catchment export in surface water, and hence where accumulation of base cations may occur over time, resulting in neutralization. The measured in-situ concentrations in surface water (cf. Cathcart *et al.*, 2016), combined with our model estimates of $S_{dep}$ and $N_{dep}$, indicate that at the current time this potential accumulation is insufficient to counteract much of the exceedances of critical loads (see following sections). However, we note that these regions may warrant further future water sampling for changes in base cation concentrations, due to their potential for future neutralization.





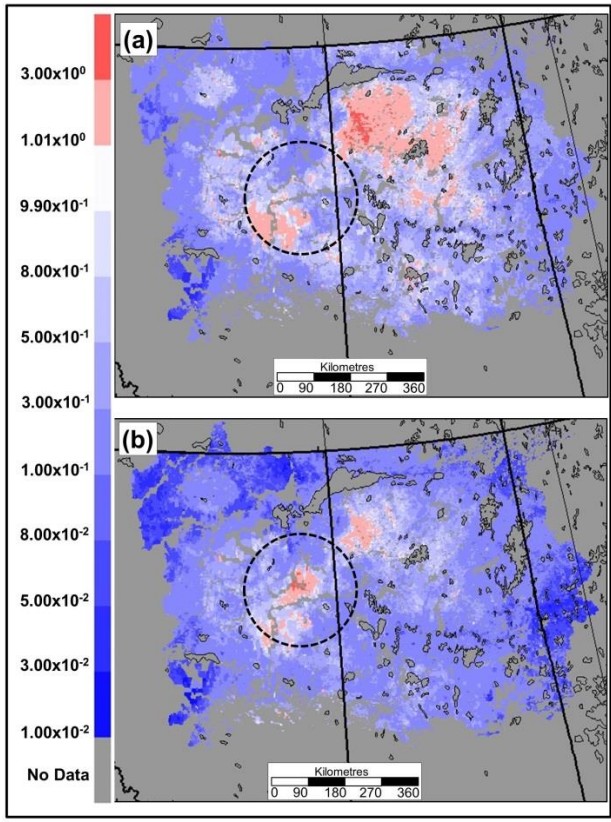

**Figure 12.** **Ratio of estimates of base cation deposition to base cation fluxes exiting aquatic ecosystems.** **(a) Ratio of interpolated/extrapolated base cation flux from 1994 to 1998 observations to aquatic base cation flux.** **(b) Ratio of model-generated and precipitation and aircraft-observation corrected base cation flux to aquatic base cation flux.** **Circled region: 140 km radius diameter circle around the Athabasca oil sands.**

## 3.6 Estimates of Critical Load Exceedances

We now estimate critical load exceedances, using both uncorrected and observation-corrected model estimates of $S_{dep}$, $N_{dep}$ and $BC_{dep}$, along with the different sources of critical load data and methodologies described above. The data of Wang *et al.* (2015) showed that the equivalent units sodium fraction of $BC_{dep}$ was 4.3%, so we assumed $Bc_{dep} = 0.957 \, BC_{dep}$, in the work which follows.

### 3.6.1 Exceedances with respect to Forest and Terrestrial Ecosystem Critical Loads

The forest critical load exceedances for $S_{dep}+N_{dep}$ calculated using the upper limit NGE-ECP (2001) critical load estimates (Canada-wide data, equations (16) and (17)), and the full CLRTAP (2004, 2016, 2017) calculation protocol (Alberta data), are shown in Figures 13 and 14 respectively. All critical load exceedances in this section are depicted using the same logarithmic colour scale for easy cross-comparison: red regions represent exceedances, and blue regions are below



exceedance. Lighter coloured shades are closer to net neutral conditions. Each critical load exceedance figure includes the total area in exceedance, and its percentage compared to the area of available critical load data. The portions of the model domain which do not coincide with the given dataset are depicted as "no data", in gray.

Figure 13 shows the predicted levels of exceedance using different $S_{dep}$, $N_{dep}$ and $BC_{dep}$ estimates. Figure 13(a) shows the predicted exceedances when the 1994-1998 $BC_{dep}$ values inferred from Canada-wide station observations are used (those stations within the 2.5km model domain appear as yellow diamonds). Figure 13(b) shows the predicted exceedances using the model's uncorrected values of $BC_{dep}$, $S_{dep}$ and $N_{dep}$. Figure 13(c) shows the predicted exceedances using precipitation and aircraft-observation corrected deposition fluxes. The different deposition estimates result in a factor of 7 variation in the

predicted area of exceedance, with the observation-corrected values having the smallest area at $1.15 \times 10^4$ km$^2$ in exceedance, or about 1% of the total (coloured) area of available critical load data. The strong impact of the model's spatially distributed base cation field and the precipitation-observation reduction in $S_{dep}$ may be seen by comparing Figures 13(b) and (c). The 140 km radius circule is around the Athabasca oil sands– acidification is predicted by the original model fields constructed using the reported emissions (Figure 13(b)), while most of this region is neutralized with the scaling of model values to

match observations (Figure 13(c)). Many of the other exceedance regions of Figure 13(b) are greatly reduced in size with the scaled information (Figure 13(c)). Nevertheless, the size of the total region in exceedance of critical loads for forest ecosystems across the entire domain using the NGE-ECP (2001) protocol, designed to create a lower estimate of critical load exceedances, is still considerable, about the size of Columbia ($1.14 \times 10^4$ km$^2$).



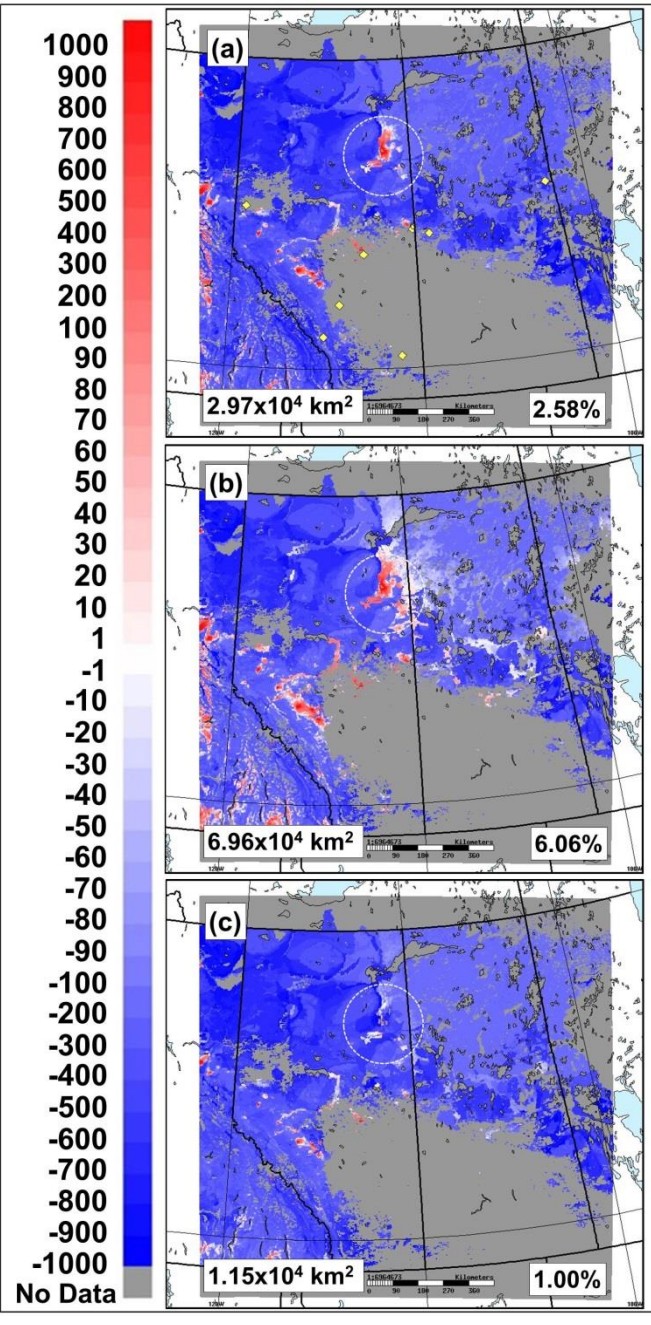

**Figure 13. Predicted forest ecosystem critical load exceedances with respect to acidity (S + N deposition), using NEG-ECP (2001) protocols (eq ha$^{-1}$ yr$^{-1}$). White to red regions: exceedance, white to blue regions: below exceedance. (a) GEM-MACH S+N deposition, interpolated and extrapolated base cation deposition from 1994-1998 observations. Station locations for base cation observations are shown as yellow diamonds. (b) GEM-MACH S+N deposition, model base cations from reported emissions of crustal material and Wang *et al* (2015) cation fractionation. (c) GEM-MACH S+N deposition scaled according to precipitation observations, base cations *scaled* using precipitation and aircraft data   Lower left of each panel: total area in exceedance, in km$^2$. Lower right of each panel: percentage of the entire critical load data area which is in exceedance. Circled region:  140 km radius diameter circle around the Athabasca oil sands.**





The terrestrial ecosystem critical load exceedances for the same estimates of $BC_{dep}$, $N_{dep}$, and $S_{dep}$, for the Alberta data using the full CLRTAP (2004, 2016, 2017) protocol appear in Figure 14 (a, b, c). While the critical load data in this case are only available for the province of Alberta itself, the regions of exceedance within that province have increased in size relative to the estimates of the NGE-ECP(2001) protocol. The influence of the precipitation observation and aircraft-observation

corrections on model-estimated deposition are evident, comparing Figure 14(c) to Figure 14 (a,b), particularly within 140 km distance of the oil sands. The increases in $BC_{dep}$ and decreases in $S_{dep}$ result in exceedances falling below zero in the central part of the circled region within the province of Alberta, and being reduced in magnitude elsewhere. However, it is important to note that despite these corrections, predicted exceeded areas nevertheless have a significant spatial extent, within some parts of the 140km radius, and remain spatially significant outside of that zone (Figure 14(c)). The total within-

Alberta area in exceedance for terrestrial ecosystem critical loads using the corrected fields is $7 \times 10^4$ km$^2$ (roughly equivalent in spatial extent to Ireland, and accounting for about 10% of the area of the province of Alberta).

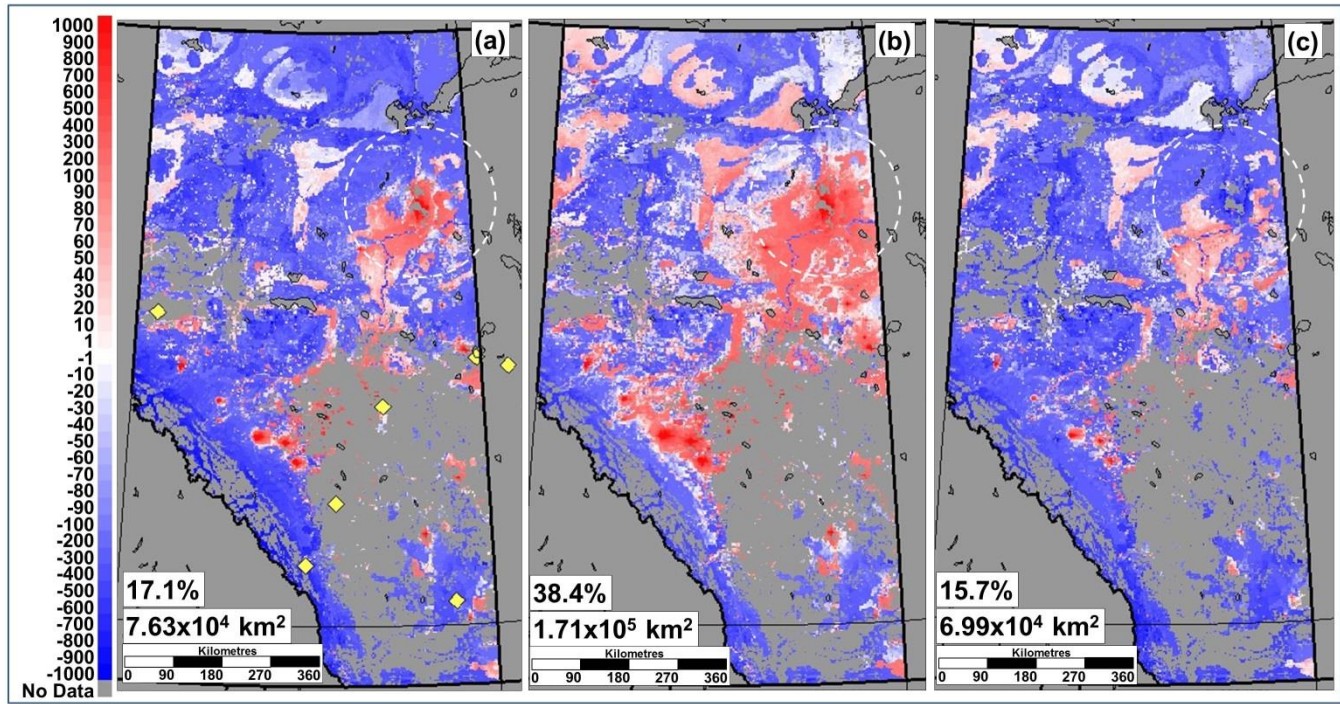

**Figure 14. Predicted terrestrial ecosystem critical load exceedances with respect to sulphur and nitrogen (eq ha$^{-1}$ yr$^{-1}$), Alberta Environment and Parks data. (a) GEM-MACH S and N deposition, 1994-1998 observed base cation deposition. Observation**
**stations shown as yellow diamonds. (b) GEM-MACH S and N deposition, NPRI/Wang *et al.* base cation deposition. (c) GEM-MACH S and N deposition, base cation deposition scaled according to aircraft and precipitation-based corrections. Circled region: 140 km radius diameter circle around the Athabasca oil sands.**

The total area of exceedance falling within each of the four regions described in Section 2.1.1 and Figure 1, along with the percentage of the total area in exceedance, are shown in the boxed portion of each panel of Figure 15. Exceedances predominantly occur in Region 2 in all cases, suggesting that both $S_{dep}$ and $N_{dep}$ are contributing most frequently to the total





exceedance. The $BC_{dep}$ field in Figure 15(b) is in general lower than for Figure 15(a), resulting in lower values of $CL_{max}(N)$, and a greater proportion of Region 1 exceedances in Figure 15(b) compared to Figure 15(a). In Figure 15(c), both $BC_{dep}$ and $N_{dep}$ have increased; while the total region in exceedance has decreased, the relative proportion within Region 1 between Figure 15 (b) and (c) therefore remains almost unchanged. The proportion of the terrestrial ecosystems where exceedances

5    are with respect to $S_{dep}$ alone (Region 4) is the smallest for the exceedance estimate using observation-based corrections of $S_{dep}$, $N_{dep}$, and $BC_{dep}$ (Figure 15(c)).

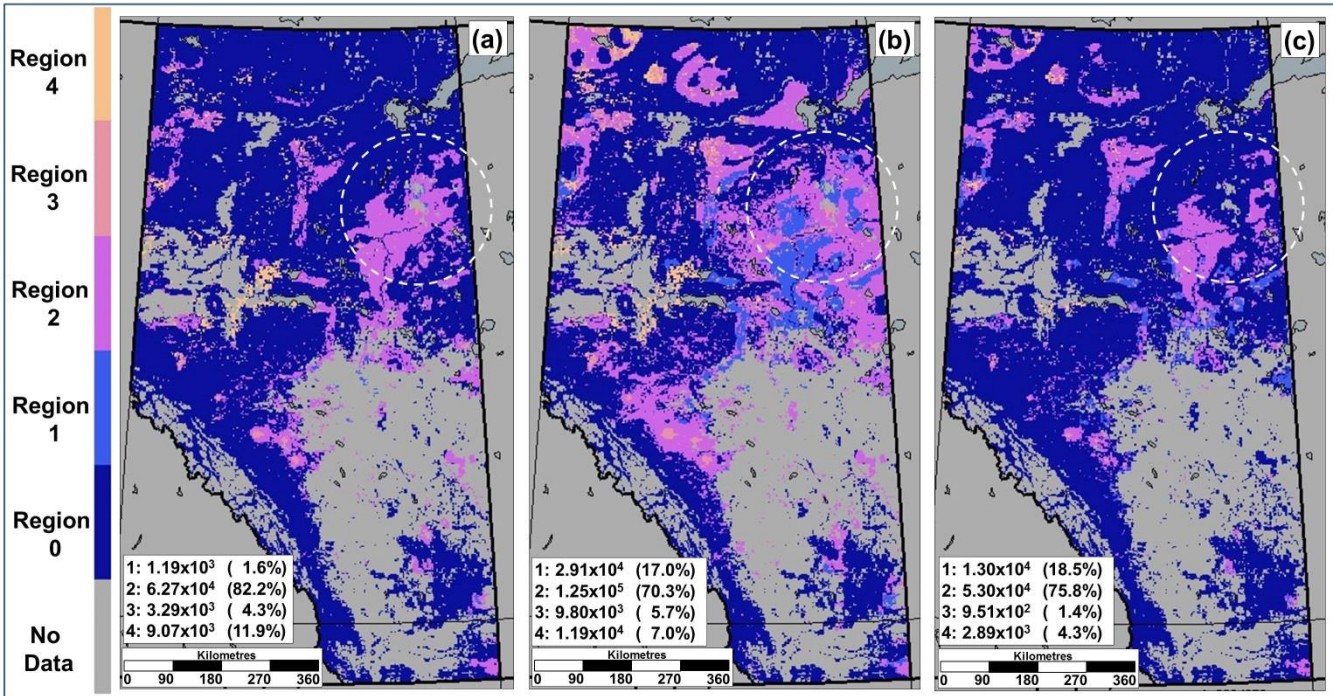

**Figure 15. Predicted regions of terrestrial ecosystem critical load exceedance, panels arranged as in Figure 14. Inset information shows the area within S + N exceedance regions 1, 2, 3, and 4 (km$^2$) and the corresponding percentage of the total area of exceedance. Circled region: 140 km radius diameter circle around the Athabasca oil sands.**



### 3.6.2 Exceedances with respect to Aquatic Ecosystem Critical Loads

#### 3.6.2.1 SSWC model:  Canada-wide versus Alberta and Saskatchewan critical load datasets.

As noted earlier, aquatic ecosystems tend to be more sensitive to acidifying precipitation (i.e. have lower critical loads) than the forest / terrestrial ecosystems.  Exceedances with respect to $S_{dep}$, calculated using equation (7), for the Canada-Wide and the Alberta and Saskatchewan critical load data, are shown in Figures 16 and 17, respectively.  Unlike the forest and terrestrial ecosystem critical loads, the base cations of the SSWC model are derived from observations of surface water, hence only the observation-based corrections to $S_{dep}$ are applied to these figures (Figure 16 and 17 (a) are using the uncorrected model $S_{dep}$, while (b) of each figure uses the precipitation-observation-based $S_{dep}$ correction discussed earlier.

The Canada-Wide data (Figure 16) covers a smaller spatial extent, and utilizes a coarse 45km resolution superimposed in the 2.5km resolution of GEM-MACH; spatial variation of the exceedances within the 45km squares are thus the result of variations in the 2.5km $S_{dep}$ values.    The $S_{dep}$ correction reduces the critical load exceedance percentage area in both cases (from 24.9% to 15.9% for Figure 16, and from 79.6% to 47.1% in Figure 17).  Aquatic ecosystems in the more recent of the two datasets (Figure 17) are clearly more sensitive than the older data (Figure 16); the use of more recent water sampling observations, and georeferenced soil and other data, have resulted in critical load estimates being somewhat lower than the earlier data (compare also Figure 2(a) and Figure 4(b)).  The georeferenced data (Figure 17) also gives a more complete spatial coverage for the region, allowing greater local detail, but also showing that portions of the region for which no data were previously available (e.g. grey areas in Northern Saskatchewan, Figure 16) are likely to be in exceedance for $S_{dep}$ (corresponding regions in Figure 17).

The lower estimates of the net area of the exceedance region in these two figures is $7.8 \times 10^4$ km$^2$ for the older critical load data, and $3.3 \times 10^5$ km$^2$ for the new georeferenced critical load data.  The former area is roughly equivalent to that of the Czech Republic ($7.9 \times 10^4$ km$^2$), the latter that of Germany ($3.6 \times 10^5$ km$^2$).

It is worth noting here that the extent of neutralization implied by comparing the atmospheric deposition of base cations ($BC_{dep}$) to $S_{dep}$ and $N_{dep}$ does not seem to be reflected in the lake water samples used to create the critical loads used in Figures 16 and 17, although some effects due to oil sands fugitive dust deposition may be seen in the observation-corrected exceedance estimates for the areas on the northern side of the oil sands (blue regions Figures 16(b) and 17(b), northern end of the circled region on each Figure).  The estimated export of base cations from catchments is usually higher than the $BC_{dep}$ values (see Figures 10 and 11 and related discussion), implying a net loss of deposited base cations.  However, some areas within the domain have higher predicted base cation deposition than observed export in surface waters, indicating the potential for an accumulation of base cations over time.  This implies a potential lag time between atmospheric deposition and surface water response.



**Figure 16. Predicted lake ecosystem critical load exceedances with respect to $S_{dep}$, NEG-ECP (2001) protocol (eq ha$^{-1}$ yr$^{-1}$). (a) Exceedances calculated using original GEM-MACH $S_{dep}$. (b) Predicted exceedances calculated using GEM-MACH $S_{dep}$ scaled using precipitation deposition observations. Circled region: 140 km radius diameter circle around the Athabasca oil sands.**





**Figure 17. Predicted aquatic ecosystem critical load exceedances with respect to $S_{dep}$, CLRTAP (2016, 2017) protocol (eq ha$^{-1}$ yr$^{-1}$). (a) Exceedances with uncorrected model $S_{dep}$. (b) Predicted exceedances with model $S_{dep}$ corrected to match precipitation observations. Circled region: 140 km radius diameter circle around the Athabasca oil sands.**





### 3.6.2.2 FAB model: exceedances with respect to $N_{dep}+S_{dep}$ for Aquatic Ecosystem Critical Loads

The exceedances for aquatic ecosystems with respect to both $N_{dep}$ and $S_{dep}$ are shown in Figure 18, using the original (Figure

18(a)) and precipitation-observation-corrected (Figure 18(b)) model fields for $N_{dep}$ and $S_{dep}$. The total area of exceedance again decreases with use of the observation-corrected fields (though not to the same degree as Figure 17, probably due to the increases resulting with the $N_{dep}$ correction offsetting some of the decreases associated with the $S_{dep}$ correction). The total area in exceedance is similar to the SSWC results (decreasing slightly for the original model $S_{dep}$ and $N_{dep}$, increasing slightly for the corrected fields: compare panels (a) and (b) between Figures 17 and 18). The FAB model critical loads suggest

deposition significantly below exceedance takes place in specific lakes (dark blue, Figure 18), while the SSWC model (Figure 17) suggests a more smoothly distributed variation between exceedance and non-exceedance regions.

Both the SSWC and FAB exceedance estimates show the oil sands region as a prominent "hot-spot" of aquatic critical load exceedance, with an influence extending far beyond the 140 km circle shown on Figures 17 and 18. Exceedances to aquatic ecosystem critical loads are predicted as far east as northern Manitoba, and into the North-West Territories on the northern

end of the data region. The exceedances using the uncorrected model deposition estimates are roughly equivalent in size to Spain ($5.0 \times 10^5$ km$^2$), while the exceedances using the observation-corrected model deposition are closer to the size of Germany ($3.6 \times 10^5$ km$^2$). By comparison, Alberta and Saskatchewan have areas of $6.6 \times 10^5$ and $6.5 \times 10^5$ km$^2$, respectively: the predicted area in exceedance of aquatic ecosystem critical loads is a significant fraction of the spatial extent of these provinces.

Figure 19 shows the shift in exceedance regions resulting between the original (a) and observation corrected (b) $N_{dep}$ and $S_{dep}$ model fields; with the reduced $S_{dep}$ and increased $N_{dep}$ resulting from the corrections to observations, a greater proportion of the exceedance regions fall within Region 1, and the totals within Region 1 increase. The totals within the remaining regions decrease, reflecting the drop in $S_{dep}$ with the precipitation-observation-based corrections.



**Figure 18.** Predicted aquatic ecosystem critical load exceedances with respect to sulphur and nitrogen, (eq ha$^{-1}$ yr$^{-1}$). Boxed numbers are the area in exceedance and the percent of the total area for which critical loads are available which is in exceedance. (a) Calculated using original model sulphur and nitrogen deposition. (b) Calculated using model sulphur and nitrogen deposition corrected to match precipitation observations. Circled region: 140 km radius diameter circle around the Athabasca oil sands.







**Figure 19.  Predicted regions with respect to deposition of sulphur and nitrogen, aquatic ecosystems.  Boxed numbers give the area in exceedance within each region and the corresponding percentage of the total area in exceedance.  (a) Calculated using original model sulphur and nitrogen deposition estimates.   (b) Calculated using model $S_{dep}$ and $N_{dep}$ estimates corrected to match precipitation observations.  Circled region:  140 km radius diameter circle around the Athabasca oil sands.**



## 4 Discussion

The critical load exceedance calculations described in the previous section were carried out with the best currently available datasets and modelling tools. However, the work has also identified limitations of those sources of information, which, if improved, would lead to improved critical load exceedance predictions. In addition, while the calculations identify the potential for ecosystem damage to be taking place now or at some point in the future, additional analysis would be required to estimate the time span to the occurrence of that damage, or to subsequent recovery. We discuss these issues, and make specific recommendations for future work, below.

(1) Clearly, better estimates of the emissions of primary particulate matter and their base cation fractionation are needed, as well as additional ambient concentration and deposition observations of the species contributing to $S_{dep}$, $N_{dep}$ and $BC_{dep}$ in sensitive regions. We have attempted to correct model results using the available data: comparisons between modelled and observed deposition, and the impact of aircraft-observation-based estimates of base cation emissions on deposition. Combined, these corrections greatly improve the bias and correlation fit between observed and estimated base cation deposition to snowpack in the vicinity of the oil sands in winter. Observation-corrected model $BC_{dep}$ values are therefore recommended for future critical load exceedance work. However, in the region examined here, this combined correction amounts to a twenty-five fold increase in base cation emissions relative to the reported values for oil sands sources. We note that the increase may represent underestimates of primary particulate matter emissions by mass, and/or a higher base cation fractionation of that mass than was observed in surface dust collected by Wang *et al.* (2015). Additional measurement-based estimates of speciated primary particulate emissions and ambient concentrations are required to carry out exceedance calculations with improved model performance.

(2) Other work (Whaley *et al.*, 2017) has suggested that bidirectional fluxes of ammonia in the boreal forest region may be taking place, and would account for GEM-MACH underestimates in the column ammonia concentration relative to satellite and aircraft observations. Further research is needed to improve bi-directional flux parameterizations (the parameterization used in the given case improved ammonia performance for the boreal forest, but decreased it for agricultural regions). However, we also note that the bidirectional flux system will result in increased "natural" ammonia fluxes from land, but will not result in upward fluxes of ammonia over water. We have carried out tests which suggest that bidirectional fluxes of ammonia will increase the net flux of ammonia to water-covered surfaces, and hence the net $N_{dep}$ to aquatic ecosystems calculated in the current work should be considered a lower estimate.

(3) As noted earlier, exceedances to critical loads indicate the *potential* for ecosystem damage, but not the timeline over which damage may be expected to occur, or the time to ecosystem recovery, if acidifying deposition is reduced. These time estimates may be obtained with the use of dynamic models (CLRTAP, 2015), and their use is recommended for targeted studies in the areas we have predicted to be in exceedance of critical loads. These dynamical modelling studies should be accompanied by measurements in the same specific exceedance areas. In





past observational studies of lakes in the environs of the Athabasca oil sands (Hazewinkel *et al.*, 2008; Curtis *et al.*, 2010; Laird *et al.*, 2013), two out of twenty lakes were found to show signs of acidification. These observation locations are depicted in Figure 20, overlaid on the map of exceedances for aquatic ecosystems with respect to $S_{dep}$ of Figure 17(b). Lake sediments from four locations (white symbols, Figure 20) were found to have increasing

levels of acidity, but within natural variability (Hazewinkel *et al.*, 2008), two lakes (red symbols, Figure 20) were found to have undergone recent acidification (Curtis *et al.*, 2010; Laird *et al.*, 2013), and the remaining locations (blue symbols, Figure 20) were not found to be acidifying. However, the sediment core stratigraphy within the region was found to be "broadly consistent with increased anthropogenic pressures in the region" (Hazewinkel *et al.*, 2008), and an examination of fifty years of six lake sediment cores found evidence for a factor of 2.5 to 23

increase in the flux of polycyclic aromatic hydrocarbons since the 1960s (Kurek *et al.*, 2013) . One of the acidifying lakes was noted to be relatively shallow and in peaty soil, with the implication that similar lakes may show the effects of acidification first (Curtis *et al.*, 2010). Twelve lake sediment cores showed that the signs of ecological changes such as sediment enrichment have been increasing over the last three decades, and increased phosphorus concentrations in several lakes were attributed to the dry deposition of $NO_x$ (=NO + $NO_2$) and other

forms of $N_{dep}$ (Curtis *et al.*, 2010). However, a study of sediment cores from 15 non-acid-sensitive lakes in northern Saskatchewan did not show evidence of lake enrichment by $N_{dep}$, based on analysis of algal communities (Laird *et al*, 2017, Mushet *et al*, 2017).Dynamical modelling (CLRTAP, 2015) would further aid in prioritizing locations for further studies to quantify acidifying effects.





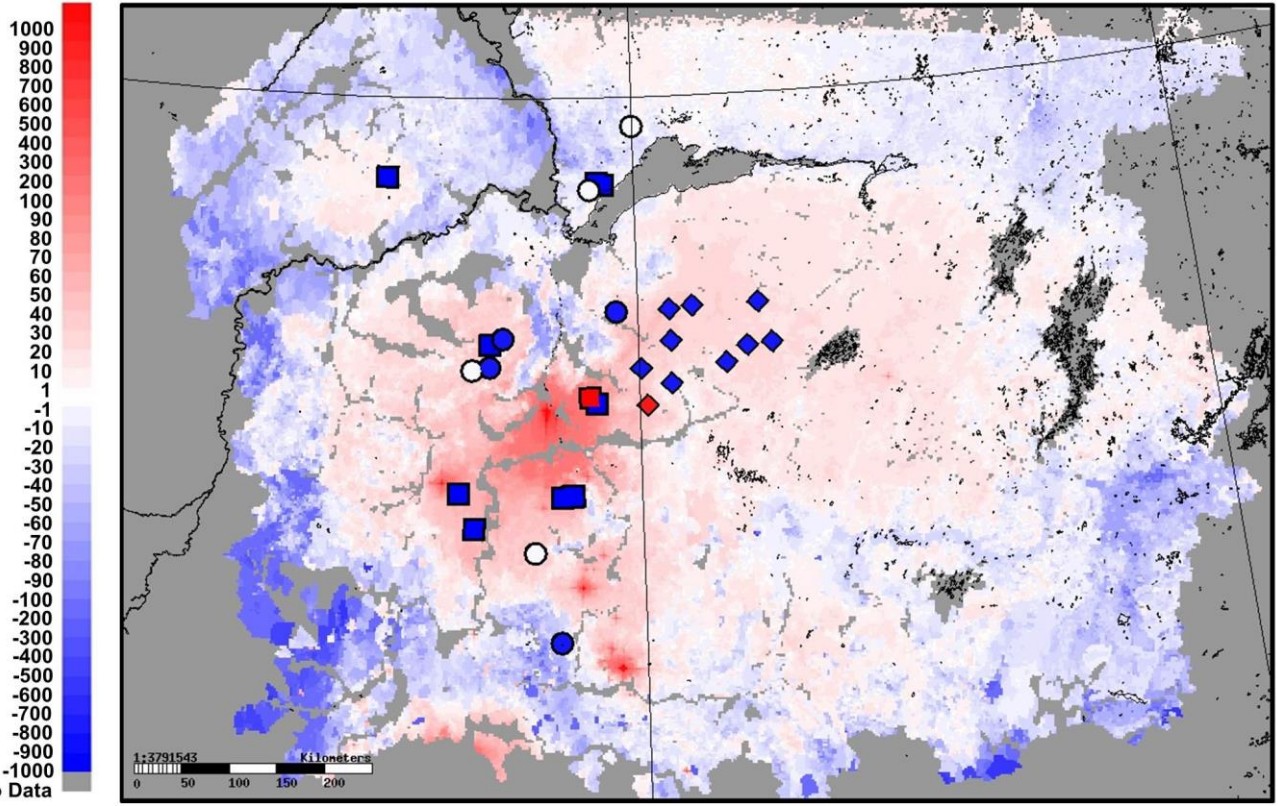

**Figure 20. Comparison of predicted exceedances with model S$_{dep}$ corrected to match precipitation observations (Figure 18b, units eq ha$^{-1}$ yr$^{-1}$) with lake observation data of Hazewinkel *et al.*, 2008 (circles), Curtis *et al.*, 2010 (squares) and Laird *et al.*, 2013 (diamonds). Blue symbols: sample locations showing no acidification at the current time, white symbols: locations with decreasing pH, but within natural variability; red symbols: locations where signs of acidification were detected. Note that the colour of the symbols, which are for illustration purposes only, does not correspond to numerical exceedance values on the colour scale, which refers only**

(4) Future GEM-MACH simulations should include the full twelve-bin particle size distribution rather than the more computationally efficient operational forecast two-bin particle size distribution used here for the annual simulation, in order to better capture the variation in base cation particle deposition with distance as a function of particle size.

(5) New measurement studies are needed in order to acquire the data to improve the current parameterizations used for estimating deposition velocities, particularly for gas-phase dry deposition. For example, most current parameterizations are based on direct observations of SO$_2$ and O$_3$, with deposition parameters for other gases being inferred by indirect means, and the temperature dependence of deposition to snowpack has been measured directly for only two species, SO$_2$ and HNO$_3$ (see Supplementary Information). Future work to characterize gas deposition, particularly under cold conditions, is therefore recommended.




## 5 Summary and Conclusions

Our work has predicted that critical loads for acidifying deposition are being exceeded in the provinces of Alberta and Saskatchewan, for both terrestrial and aquatic ecosystems. Model predictions indicate that that total deposition downwind of sources is dominated by the wet component. Model comparisons of sulphur, nitrogen and base cation deposition with observations indicate that the model has some skill in accounting for the observed variability in wet deposition ($R^2$ of 0.90, 0.76, and 0.72, respectively). We therefore used the model versus observation linear relationships from wet deposition to provide a correction to model values for total deposition of sulphur, nitrogen and base cations. Aircraft-observation-based estimates of primary particulate matter emissions were shown to result in a factor of 10 increase in atmospheric base cation deposition close to the oil sands emissions regions, and corrections for base cation deposition based on these estimates were also incorporated into our investigation of exceedances. Making use of both the original model predictions and the corrected fields, exceedances of critical loads were calculated using simplified methodologies designed to provide lower limit estimates of exceedances (NEG-ECP, 2001), and more rigorous methodologies to take into account additional factors such as ecosystem buffering capacity (CLRTAP, 2004, 2016, 2017). While atmospheric base cation deposition was shown to have a significant neutralizing impact for terrestrial ecosystems close to the sources of fugitive dust emissions, this effect was shown in both observations and model results to drop off rapidly with distance, in accord with well-known physics controlling the deposition velocities of atmospheric particles as a function of their size. Exceedances were predicted further downwind, despite these corrections to the original model estimates (which include an assumed factor of twenty-five increase in primary particulate matter emissions from oil sands sources, relative to reported emissions). Aquatic ecosystem critical load data suggest that the base cation loading within catchment waters is insufficient to counteract much of the atmospheric deposition of sulphur and nitrogen. The results thus indicate that potential ecosystem damage may be taking place, due to acidifying deposition in the provinces of Alberta and Saskatchewan. The use of dynamic models to determine the timelines until damage occurs and/or recovery may take place, and observational studies for the presence of ecosystem damage, are recommended for future work, with a focus on the highest exceedance regions predicted here. Further observations of deposition of sulphur, nitrogen and size-resolved base cations are also recommended, at distances greater than 140 km from the sources, to further evaluate and improve on our findings.

Specific results of our work include:

(1) The spatial extent of predicted exceedances of forest and terrestrial ecosystem critical loads range from $1 \times 10^4$ km$^2$ to $6.69 \times 10^4$ km$^2$ (10% of the area of the province of Alberta), with the latter estimate based on the more comprehensive critical load calculation protocol.

(2) The spatial extent of predicted exceedances of aquatic ecosystem critical loads in the region studied is larger than that of forest and terrestrial ecosystem critical loads. Estimates using both earlier lake observation data and more recent georeferenced data indicate that a significant fraction of northern Alberta and Saskatchewan lakes are



predicted to be in exceedance. Some neutralization due to base cation levels in water observations may be occurring immediately to the north of the oil sands, but overall, exceedances are predicted over much of the north of the two provinces, and extend eastwards into Manitoba, for all three of the critical load datasets and protocols employed here.

(3) Our work suggests that other sources of base cations, aside from atmospheric deposition, usually controls the surface water base cation concentration. Our model results and our re-examination of the throughfall data of Watmough *et al.* (2014) suggests that the neutralization associated with base cation deposition from sources of fugitive dust in the oil sands area will be limited in spatial extent. Despite this near-source neutralizing effect, potential ecosystem damage associated with acidifying precipitation may take place further downwind.

Nevertheless, our work demonstrates that both natural and anthropogenic base cation emissions may have a significant impact on, and should be included in, critical load exceedance calculations.

    (4) We predict that in some portions of the study region, base cation deposition from the atmosphere may exceed the estimated removal of base cations from catchments in water. While the observations of surface water ion content and estimates of the export of water from catchments used to create the critical loads employed here indicate that

the base cation level in surface water is insufficient to counteract acidification, there exists the potential for this to change over time. Repeat measurements of catchment water in these regions of potential base cation buildup, and follow-up work to improve and evaluate catchment water export rates, are therefore recommended. Strategies to measure deposition to very acid-sensitive regions (e.g., exceedance (red) regions in Figure 14(c), Figure 17(b), and Figure 18(b)), which are distant from existing conventional deposition monitoring sites, should be considered.



## 6    Author Contribution

P.A.M: Study concept and design, analysis of model output and critical load exceedances, writing of manuscript and modifications of same; A.A.: GEM-MACH simulations, assistance with model evaluation and analysis; J.A.: critical load data section of text, provision of aquatic ecosystem critical loads; A.S.C: preparation of precipitation deposition data, assistance with new and historical aquatic critical load data sections of manuscript; Y.A.: AEP terrestrial ecosystem critical load data and description and AEP precipitation data; J.Z: creation of emissions files used in the model,; I. Wong: provision of ECCC Lakes and Forest critical load data; K.H. and S-M.L.: provision of aircraft data; J.K.: provision of ECCC snowpack data and text on same; K.S.: provision of Saskatchewan Environment snowpack data and text on same; M.D.M: assistance with emissions data for model; A.R.: assistance with Supplement 1 text; H.C: creation and provision of aquatic critical load data; P.B.: assistance with generation of emissions data; B.P. and P.C.: assistance with GEM-MACH setup, model simulations; Q. Zheng: assistance with emissions generation. In addition, the first author would like to thank all co-authors for extensive comments on different versions of the manuscript.

## 7    Acknowledgements

This project was jointly supported by the Climate Change and Air Quality Program of Environment and Climate Change Canada, Alberta Environment and Parks, and the Joint Oil Sands Monitoring program.

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
