# Peer review of "Estimates of Exceedances of Critical Loads for Acidifying Deposition in Alberta and Saskatchewan"

_Atmospheric Chemistry and Physics, 2017_

## Referee Comment (RC1) · Anonymous Referee #2 · 31 Mar 2018

Review of MS acp-2017-1094:

**"Estimates of Exceedances of Critical Loads for Acidifying Deposition in Alberta and Saskatchewan"** by P.A. Makar *et al.*

submitted for publication in *Atmospheric Chemistry and Physics*.

General remarks:

This rather extensive paper reports on detailed deposition calculations for the two Canadian provinces as well as their use in exceedance calculations for different sets of critical loads (CLs) for terrestrial and aquatic ecosystems. As the larger part -- and more of the novel material – is concerns atmospheric depositions, I suggest to change the title to "Estimates of Acidifying Deposition and Critical Load Exceedances in Alberta and Saskatchewan" (or similar), and also to restructure the paper accordingly, i.e. first depositions, and then CLs and their exceedances. With respect to depositions I suggest to move some of the material to the 'Supporting Information', since it's mostly material taken from existing literature. The paper (and the reviewer) would have benefitted if the authors had carefully read the paper before submission: there are close to 30 references that are there and not cited or cited and not in the reference list (see below); also, the equation numbering is quite faulty in some parts. Furthermore, the definition of critical load exceedances (in case of non-exceedance) requires some attention (see below). Apart from this, I consider the material of paper suitable for publication, after the authors have also taken into consideration the (often minor) remarks/corrections listed below.

Detailed remarks:
*Note:* '*X* → *Y*' means: replace '*X*' by '*Y*' (in the text).

**Title:**
See 'General remarks' above.

**Abstract**:
P[age]1, L[ine]33: Suggest to change 'protocols' to 'methods' (throughout the paper!), as 'protocol' has its own meaning in the context of CLRTAP!
P1, L34: Delete 'forest and': forests are terrestrial ecosystems!
P2, L2: Delete 'emissions and'.
P2, L7: 'was shown to have' → 'has' (otherwise it sounds the authors have shown that in this paper).
P2 L11: 'primary particle dust particles' → 'primary dust particles'.

**Introduction:**
P2, L17: Delete 'regional and'.
P2, L18: The reference to the CLRTAP Manual sh/could be simplified (throughout the whole paper!), as it's always the same source. Just call it always CLRTAP (2017) -- with the text in the References as is now under 'CLRTAP, 2004' -- since 2017 is the (last) time you accessed it.
P3, L 16: 'Estimates of critical loads' → 'Critical loads'.
P4, L1-2: No! If $BC_{dep}$ is greater than $S_{dep}+N_{dep}$, a large part of that $BC_{dep}$ could be taken up by forests and harvested (i.e. taken away) and thus not be available for neutralising the S and N deposition; and a case could also be made for the converse -- If the statement were true, then the CL would be equal to $BC_{dep}$!
P4, L5: 'emissions levels' → 'emission levels'.
P4, L7: What is 'alkylization'? (I guess the authors mean 'alkalinisation'?).
P4, L24: 'aquatic and terrestrials' → 'aquatic and terrestrial ecosystems'.
P4, L25: Insert 'deposition' after 'surface'.

**Methodology**:

P5, L15: 'sum of in equivalents of' → 'sum of'; the criteria is generally reported as 'molar Bc:Al (or Al:Bc) ratio! That's way the factor 3/2 appears in eq.4 to convert it to equivalents!

P5, L19: There is no 'level of protection' defined for CLs.

P5, L19-20: It's the chemical criterion (here Bc:Al ratio) that *defines* the critical ANC leaching – the user does not *specify* the critical ANC leaching (in the case described here), he just computes it!

P5, eq.2: '($CL_{max}(S)/(1-f_{de})$)' → $CL_{max}(S)/(1-f_{de})$; i.e. drop the superfluous parentheses.

P5, eq.4: Delete the superfluous parentheses (twice); only the square brackets are needed.

P5, L30: Bc is already explained above (line 15).

P5, L31: Insert 'annual' after 'long-term'.

P6, L1: 'due to other forms of removal (e.g., harvesting)' → 'due to, e.g., harvesting'.
… and all variables in the text should be in *italics* if they are so in the equations (also further below)!

P6, L8: '$Q$' is already defined on line 1.

P6, eq.6: $Y$ does not stand for Ca+Mg+K+Na-Cl, but $\Sigma_Y$ stands for the sum of base cations minus chloride (Ca+Mg+K+Na–Cl).

P6, L28: '(i)' → '(subscript $i$)', etc.

P7, L4: What is 'charge x mole equivalent'? moles? moles of charge? …

P7, L22: There is a change in font size from that line onwards – Any reason?

P8, L1: 'In some instances, S deposition (or N) must be reduced to achieve non-exceedance' What do the authors want to say? As it stands, it's trivial/obvious.

P8, Figure 1: (a) Why is the slope of the Critical Load Function (CLF) shown as 45°? This is a special case only for $f_{de}$=0 (see eq.2); (b) The point ($N_0,S_0$), computed in eqs.12,13 should be shown on the Figure; (c) It should be indicated in the Figure how the quantity $E_0$ is derived, i.e. where $N_A$ and $S_A$ are located on the CLF.

P8, L5: 'denotes ecosystem': No, it does not denote ecosystems, it denotes 'the case for which'.

P8, L15: $E_0$, as a negative quantity, cannot be a distance, only a positive quantity can; e,g, $|E_0|$ = $-E_0$ (thus it would be better to define $E_0$ as a positive quantity and make it $-E_0$ in eq.11)!

P8, L15: There are no 'exceedance lines' – what you mean is the critical load function.

P8, eq.15: In fact, $N_A$ is the $N_{dep}$-value on the CLF for a given $S_{dep}$, and $S_A$ the $S_{dep}$-value on the CLF for a given $N_{dep}$. It can be easily shown that $S_{dep}$–$S_A$ is always greater than (or equal to) $N_{dep}$–$N_A$, or, to express it in positive terms (i.e. distances): $S_A$–$S_{dep} \leq N_A$–$N_{dep}$. Thus eq.15 simplifies to:

$$(15) \quad E_0 = \begin{cases} S_{dep} - CL_{max}(S) & \text{for } N_{dep} \leq CL_{min}(N) \\ m(N_{dep} - CL_{max}(N)) & \text{for } CL_{min(N)} < N_{dep} < CL_{max}(N) \end{cases}$$

… and eq.16 becomes superfluous.

P9, L5: Insert 'critical loads' after 'estimated'.

**Note: For computing $E_0$ is a different distance measure is used than for computing positive exceedances. This is not really faulty, but peculiar, and should at least be mentioned (and maybe 'justified'). More generally, the authors should give reasons why they map negative exceedances, as policy makers might not be so much interested in them; generally, they are 'happy' when there is non-exceedance (however small) ... But it makes 'nice' maps; and maybe there is another reason as well …**

P9, L11: '… in order to obtain data for critical load estimates': Only for that purpose?

P9, L15/16: '… and other related information': What else c/would that be?

P9, L16/17: 'estimates … were conducted': Do you really conduct estimates?

P9, L20: Delete 'lowest'.

P9, L21: 'for grid cells containing *smaller* number of lakes, the … most sensitive lake was used': Smaller than what? (20? …). And, by the way, percentiles can be computed for any number …

P9, L22/23: 45 km$^2$ grid cell – What's that? If it's a 45 km × 45 km grid cell its size is 2025 km$^2$; if it's area is 45 km$^2$, what are the lengths of the sides? – It's no problem to call it a 45 km grid cell in the former case …

P10, L5: Only BC deposition is needed for CL alculations.

P10, L6: Critical alkalinity leaching is not an input, but calculated from, e.g. a critical Bc:Al ratio.

P10, L14: '(a) a critical Bc:Al ratio of 10 and a $K_{gibb}$ of 3000.0 were used': This is not a *simplifying* assumption! Furthermore, provide units for this numbers!

P10, L16: Replace 'invariant' by 'spatially uniform'.

P10, L16/17: 'the equivalent of the ($CL_{max}(S)/(1-fde)$) term …': Although I can infer what you want to say, it's confusing for the non-expert. What you are doing is modelling denitrification as a constant flux $N_{de}$, (i.e. $CL_{min}(N) = N_i+N_u+N_{de}$) and not with the fraction $f_{de}$, as in eq.2 – but this has to be explained (better)!

P10, L17: Why the somewhat 'awkward' number 35.7? Why not 36, or 40? Tell the reader that it comes from a nice round number in a different unit and an assumed soil depth of 0.5 m (this is a simplifying assumption that should be mentioned!)

P10, L18: 'weathering … was assumed to be dependent on temperature': This is not really a simplifying assumption to me.

P10, eq.17: (a) the minus sign should be a plus sign! (b) Why do you insert the numbers (2 times 35.7) for $N_i$ and $N_{de}$, but not for $(Bc:Al)_{crit}$ and $K_{gibb}$? (c) Some reason should be given why this is now called CL(S+N), and not $CL_{max}(N)$ as in eq.2.

P10, eq.18: The square brackets are superfluous.

P10, L23: '$Bc_{we} = 0.75 BC_{we}$': this is another simplifying assumption, that should be mentioned above.

P10, L27: (a) I guess it's 2.5 km × 2.5 km (or 2.5 km for short) grid cells. (b) Delete 'lowest'.

P10, L31: Parentheses around $S_{dep}+N_{dep}$ superfluous.

P11, eq.19: This is a 'dangerous' equation! What if $N_{dep} < N_i+N_{de}$? The remaining N-sink can **not** compensate any S deposition! Maybe it does never happen in AL and SK (?), but it has to be said that the (potentially) remaining N sink is not used to compensate $S_{dep}$, as the equation does as it stands!

P12, L1: 'permutations' seems a strange expression in this context!

P12, L11: 'Soils' → 'Soil'.

P12, L20: How deep was the rooting zone?

P12, L27: Comparing $CL_{max}(S)$ with CL(S+N) does not make much sense, as the latter includes N-terms; thus CL(S+N) could be compared to $CL_{max}(N)$.

P14, L4: '(equations 1 through 16)': This is misleading; e.g. eqs.1-4 describe the CLs for terrestrial ecosystems. Improve the citation of equation numbers in the whole paragraph! By the way: the exceedance calculations given in eqs.11-16 are not fully correct for FAB CL functions, as also the first segment of the CLF is tilted.

P14, L6: 'predictive maps': wouldn't 'interpolated maps' be more appropriate?

P14, L6/7: 'four target variables': In the next 2 lines only 3 variables are mentioned (BC, DOC, $SO_4^{2-}$ ); what's the 4th one?

P14, L16: 'wetlands' → 'peat'.

P19, L1: 'Yao' → 'Yau'.

P21, L20 – P22, L4: This paragraph goes into very much detail … Maybe to Supp Info?

P22, L7: Isn't sulphate an anion?

P22, L12: 'Manzaono' → 'Manzano'.

P22, L13 and eqs: The equation numbers are wrong: On page 11 yu had already eq.19!!

P22, L14: 'used in equation (18)': Which eq.18?

P22, L24: Delete 'in 2014' (?)

**Results:**

P25, L1: Insert 'in' after 'result'.
P25, L3: 'input emissions'➔ 'emission inputs'.
P26, L4: Sub-header: 'oil sands' → 'Oil Sands region' (?)
P27, L28: 'simulation, of' → 'simulations by'.
P31, L30/31: merge lines.
P33, L13: Delete parentheses around '$BC_{dep}$–$S_{dep}$–$N_{dep}$'.
P34, L16: 'sampling for' → 'sampling to monitor'.
P35, L9: 'critical load exceedances' → critical loads and their exceedances'.
P35, L13: Sub-header: 'Exceedances with respect to' → 'Exceedances of'.
P36, L18: 'Columbia': **NO**! Columbia is 100 times larger (about $1.14 \times 10^6$ km$^2$)!
P38, L3: ''have increased in size relative to' → 'are larger than'.
P40, L1: Sub-header: 'Exceedances with respect to' → 'Exceedances of'.
P40, L6: 'equation (7)': No, it's equ.(5), I presume.
P40, L6: 'Canada-Wide' → 'Canada-wide'.
P40, L11: 'superimposed in' → superimposed om'.
P40, L20: 'to be in exceedance' → 'to be exceeded'.

**Discussion:**
P46, L12: 'improve the bias and correlation' → 'reduce the bias and improve the correlation'.
P46, L30: 'expected to occur' → 'expected to occur or has occurred'.
P47, L4: 'of Figure 17(b)': Or 18(b)? As said in the caption of Fig.20

**Author Contribution:**
P51, L7: 'Lakes and Forest' → 'lakes and forest'.

**References:**
The following **references** are cited in the text, but **missing** here:
- NPRI, 2013
- Aherne, 2013
- Aklilu, 201**x**
- ECCC, 2014
- Nasr et al., 2010
- Whitfield et al., 2010
- Pregitzer et al., 1990
- Stockwell et al., 1989
- Gong et al., 2003a
- Gong et al., 2003b
- Gong et al., 2015
- Makar et al., 2017
- Wesely et al., 1989 [or is this the same as Wesely, 1989?]
- Slinn, 1982
- Jacobson, 2003 [or should it be Jacobson, 1999, which is given, but not cited?]
- Watmough et al., 2015 [or should it be 2014?]
- Whaley et al., 2017

The following **references** are **superfluous**, as they are not cited in the text:
- Brook et al., 1999
- Dasch and Cadle, 1986
- Ellsworth and Reich, 1993
- Environment and Climate Change Canada, 2017 [ I guess that's ECCC, 2017, cited in the text!?]
- Henriksen, 1984
- Hicks et al., 1987
- Hosker and Lindberg, 1982
- Jacobson, 1999 [maybe 2003? -- see above]
- Meyers et al., 1998

- Sverdrup and De Vries, 1994
- Voldner et al., 1986
- Wesely and Hicks, 2000

**Figures:**
Figure 1: Improve as suggested above.

Figure 2: Caption: 'Lake ($S_{dep}$)' is a somewhat starnge notation. Why not give the equation number used to calculate the CL. Same for 'Forest ($S_{dep}+N_{dep}$)'.

Figure 4: Caption: Replace '$S_{dep}+N_{dep}$' by '$S_{dep}$ and $N_{dep}$' [and add '(FAB model)'after it].

Figure 5: Caption: 'Sulphur' → 'sulphur' or 'S'.

Figure 6: Caption: 'Nitrogen' → 'nitrogen' or'N'.
- Plate (i): Isn't it the **sum of** particulate nitrate (dry), gaseous organic nitrate (dry), etc? And not **each of**!?

Figure 13: Caption: Add that Alberta and Saskatchewan are shown in the maps (?) [also in other Figures!]

Figure 14: Caption: Add that Alberta is shown in the maps (?) [also in other Figures!]
- Year after Wang *et al.* is missing.

Figure 15: Caption: Add '(see Figure 1)' to explain the regions 1,2,3,4! The term 'region' in this context is a bit confusing – e.g., 'cases' would be clearer, to distinguish from geographical regions.

Figure 19: Caption: Add '(see Figure 1)' to explain regions 1,2,3,4!

Figure 20: Caption incomplete!

---

## Referee Comment (RC2) · Anonymous Referee #1 · 31 May 2018

General Comments:

This manuscript by Makar et al. entitled "Estimates of Exceedances of Critical Loads for Acidifying Deposition in Alberta and Saskatchewan," is a comprehensive assessment of atmospheric deposition of nitrogen (Ndep), sulfur (Sdep), and base cations (BCdep) for Alberta and Saskatchewan, and how that relates to critical loads of acidification for terrestrial and aquatic systems. They explore many different improvements to base datalayers using climatic adjustment, and improved source emissions from aircraft estimates. These improvements move the field forward in our understanding of this environmental stressor, specifically in Canada, but potentially for any temperate

industrialized country with active mining operations.

I only had two substantial comments, neither of which negate the quality of the manuscript (though #2 may), and addressing each I think would improve an already strong submission. First, little attempt is made to extrapolate and infer whether the improvements to deposition (i.e. climatic adjustment for Ndep and Sdep and emissions adjustments for BCdep), which appear very important for Alberta and Saskatchewan, may or may not be advised in other industrialized parts of the world. My suspicion is that in vast areas of the US, Europe, and China, similar adjustments may be warranted. Second, the use of the 5th percentile for lakes (or the minimum) may be flawed. This is elaborated more below in the specific comments, but the accuracy of the 5th percentile as truly representing the 5th depends on the nderlying sample. If there are many lakes in the grid cell (e.g. >20), it will be accurate, if there are not, (e.g. <10), it will not. This is exacerbated when the authors decided to select the minimum when the sample is very small. In these cases, the minimum of the small sample is unlikely to be anywhere near the true minimum, and is probably closer to the mean. Some discussion of this is needed or edits to the methods for the aquatic. This is the same problem that Clark et al. (2018) fell into, and needs to be mentioned (https://doi.org/10.1002/eap.1703). These two issues raised are important, but neither is a "deal breaker" in terms of acceptance for publication, as there are many important strengths and insights in the manuscript. Other, specific comments below.

Specific Comments:

Abstract:

"Aircraft-observation-based estimates of fugitive dust" is awkward, how about "Aircraft-based estimates of fugitive dust" or "Aircraft observations of fugitive dust"

"Aircraft-observation-based estimates of fugitive dust emissions, shown to be a factor of ten higher than reported values". Clarify "reported", is this from ground observations, reported in some governmental permit, modeled?

Introduction:

Pg 4 lines 13-17: Hard to follow multiple embedded proportions, just proportionalize everything to Canada.

Pg 4 line 17-18: If NOx and NHx emissions from the oil sands are only 3.8 and 1%, respectively, of the total emissions of NOx and NHx in Alberta, how are they the main anthropogenic sources? Everything else is natural?

Methods:

18 pages of methods is ridiculous, but if this is ok with the journal, it's ok with me. I'd prefer to see this in an Appendix and have a brief methods section in the main text.

Pg 9 line 19-23: The subsampling is likely flawed. Using the 5th percentile is only appropriate if there are many lakes in a grid cell (i.e. > 20). If there are very few lakes in a grid cell, assuming a normal distribution, the lakes will more approximate the mean than anything else. Thus, selecting the "minimum" when there are only a few lakes is the minimum in name only, as these lakes more likely represent the mean. We ran into the same problem in Clark et al. (2018) Ecological Applications and discussed its implications (https://doi.org/10.1002/eap.1703). Please consider revising the methodology (i.e. only use grid cells with > 20 lakes) or discuss this issue in the paper.

Parse out in the figures and captions of Figs 2-4 the "No Data" category. There are many possible reasons for no data, and knowing and communicating that is helpful. What that "No Data" because it was not the right land cover type (e.g. non-forested) or because there was no estimate of data (e.g. BC dep), or some other reason. These are very different. Please separate the reason for no data into at least two categories (i.e. "No appropriate cover" and "No data").

Results:

Figure 5: Interesting that most of the N dep is from NH4+ wet, which did not seem to

be mentioned much in the introduction. Maybe introduce this a bit more so it's not a surprise.

Pg. 24-25: Please add a bit more discussion on the sources of error that could be attributable to the model and could be attributable to the observed estimates. Both are potential sources of error and I found this a bit too brief.

Pg. 25: Very nice that you used uncorrected and corrected deposition estimates in calculations of exceedance.

As mentioned earlier "Aircraft-observation-based" is a mouthful and not necessary. Use "aircraft-based." It would be silly to take a computer up in the aircraft to make this an "aircraft-simulation-based" estimate.

Pg 27, line 12-13. I don't follow the logic of why this following from the preceding sentences: "This in turn suggests that the primary particles may rapidly deposit with increasing distance from the emissions sources." If the emission inventories are too low, why does that mean they deposit faster? Couldn't this explain the bias in the BCdep (i.e. that the actual emissions are higher, and thus the model estimates BCdep as too low)?

Pg 27 line 18-19. It shouldn't be "aircraft emissions estimates" (i.e. the emissions of the aircraft) it should be "aircraft-based emissions estimates"

Pg. 27-28. Would be nice to have a "take home message" in terms of how far from a large point source do you need to be before one needs to worry. Seems like from Figure 8 you need to be >100 km, which would be a useful rule of thumb to include. Are these underestimates for Canada likely occurring elsewhere in the world as well?...Later I see that the 142 km threshold (pg. 33) is presented, which is slightly different but related and useful.

Pg 29 line 1-2. Unclear why the improvement in the BC emissions using the aircraft observations also improves the Sdep and Ndep, please clarify.

Section 3.4. It's not really clear to me what value the comparison with snowpack adds except to say that the snowpack isn't a very useful comparison. If that is the case, I'd move all this to the supplement, and just have a short statement to that effect. This would shorten an already very long paper.

Figure 12 a, what is the red hotspot to the NE of the Athabasca oil sands where BCdep again dominates? Is that another source or evidence of long range transport?

Figure 14 b-c: Unclear whether the S and N dep are precipitation adjusted, which I think they should be in one of either b or c? The description in Figure 13 is more clear and complete (i.e. "GEM-MACH S+N deposition scaled according to precipitation observations, base cations scaled using precipitation and aircraft data".

Not really clear what Figures 15 and 19 on the Regions adds, please clarify.

Discussion:

This is fantastic work for Canada, but some discussion of whether the results and lessons would hold for other parts of the world would be nice if briefly discussed (e.g. Europe and the US?).

Pg 46 line 29-30: In addition to exceedances not helping with the timeline or recovery, they also don't really comment on the magnitude of effect, also an important note.

Pg 47 and Figure 20: This is a really nice addition, but it seems to me that comparing exceedances with actual impact is a much bigger effort than this would suggest. If it's just included as a preview, it's fine, otherwise, maybe caveat that much more comparison with observed effects is needed (e.g. forest tree growth??).

Please add a non-spatial plot to simplify the information in Figure 20. The take home point appears to be that exceedance does not translate to effect.

Pg 49 line 15: I would not characterize a radius of 142 km as "rapid". I'd say 10 km would be rapid, but that circle in the figures is pretty large!

---

## Author Comment (AC1) · 20 Jun 2018

**Responses to Reviewers comments, "Estimates of Exceedances of Critical Loads for Acidifying Deposition in Alberta and Saskatchewan"**

Original reviewer comments are in normal font text, and the responses are in *italics*.

**Anonymous Referee #1**

General Comments:

This manuscript by Makar et al. entitled "Estimates of Exceedances of Critical Loads for Acidifying Deposition in Alberta and Saskatchewan," is a comprehensive assessment of atmospheric deposition of nitrogen (Ndep), sulfur (Sdep), and base cations (BCdep) for Alberta and Saskatchewan, and how that relates to critical loads of acidification for terrestrial and aquatic systems. They explore many different improvements to base datalayers using climatic adjustment, and improved source emissions from aircraft estimates. These improvements move the field forward in our understanding of this environmental stressor, specifically in Canada, but potentially for any temperate industrialized country with active mining operations.

*We thank the reviewer for the positive review of our work!*

I only had two substantial comments, neither of which negate the quality of the manuscript (though #2 may), and addressing each I think would improve an already strong submission.

First, little attempt is made to extrapolate and infer whether the improvements to deposition (i.e. climatic adjustment for Ndep and Sdep and emissions adjustments for BCdep), which appear very important for Alberta and Saskatchewan, may or may not be advised in other industrialized parts of the world. My suspicion is that in vast areas of the US, Europe, and China, similar adjustments may be warranted.

*The key issue here is that air-quality model estimates of $N_{dep}$, $S_{dep}$ and $BC_{dep}$ are subject to a host of possible errors, ranging from errors in the emissions inputs to the level of the model's ability to accurately simulate the chemical and physical processes leading to deposition. In that respect, correcting the model output using comparisons to the measurements in some fashion is something we can definitely recommend. However, these corrections must be done on a case-by-case basis. For example, while we show that our model estimates for the component of $S_{dep}$ in precipitation are highly correlated with observations, they were almost a factor of 2 too high relative to observations. In a different jurisdiction, with different emissions (and emissions accuracy, and/or a different air-quality model), a completely different bias might result. So yes, we would advise that some form of model-measurement fusion, whether as simple as what we have carried out here, or using a more complex methodology as pioneered by one of us (Robichaud), should be carried out for future critical load exceedance estimates using air-quality models, in order to improve on their accuracy. We have added a sentence to that effect at the end of the last paragraph in the revised Abstract, viz, "We strongly recommend the use of observation-based correction of model-simulated deposition in estimating critical load exceedances, in future work," and have also added a fifth point to our Summary and*

*Conclusions section: "(5) We have found that corrections of model estimates of $S_{dep}$, $N_{dep}$ and $BC_{dep}$ using observations, and using direct observation-based emissions data for base cations, have a significant impact on model estimates of critical load exceedances. Here, relatively simple corrections using model-observation relationships were employed. We note that other means of model-measurement fusion for acidifying pollutants are under investigation, and show great promise for creating observation-corrected air-quality model deposition fields (e.g. Robichaud et al., 2018)."*

Second, the use of the 5th percentile for lakes (or the minimum) may be flawed. This is elaborated more below in the specific comments, but the accuracy of the 5th percentile as truly representing the 5th depends on the underlying sample. If there are many lakes in the grid cell (e.g. >20), it will be accurate, if there are not, (e.g. <10), it will not. This is exacerbated when the authors decided to select the minimum when the sample is very small. In these cases, the minimum of the small sample is unlikely to be anywhere near the true minimum, and is probably closer to the mean. Some discussion of this is needed or edits to the methods for the aquatic. This is the same problem that Clark et al. (2018) fell into, and needs to be mentioned (https://doi.org/10.1002/eap.1703).

*The original lines of text were poorly worded, and this has been corrected first of all to clarify the procedure used in the Jeffries et al (2010) work to collect the CL data mentioned, and an additional several lines of discussion has been added. We note in the latter that the Jeffries et al (2010) data is an historical dataset included here due to its availability and to allow us to compare to more recent data and methodology, described in section 2.1.4: "The SSWC critical load values for each surveyed lake contained within each AURAMS grid-cell were compared – when data from multiple lakes within the same grid cell were available, the fifth percentile of the resulting critical load values was assigned to that grid cell (for grid cells containing less than 20 lakes, the critical load for the most sensitive lake was used). The lake critical load data thus represent the most sensitive lake ecosystems within the given grid cell based on the available data. We note, however, that this procedure used in the creation of this dataset (Jeffries et al., 2010) becomes less accurate as the number of lakes per grid cell becomes small, with either over- or under-estimates of local ecosystem sensitivity. This was one of the factors leading to more recent updates in aquatic critical load maps for Canada, discussed in more detail in section 2.1.4."*

These two issues raised are important, but neither is a "deal breaker" in terms of acceptance for publication, as there are many important strengths and insights in the manuscript. Other, specific comments below.

Specific Comments:

Abstract:

"Aircraft-observation-based estimates of fugitive dust" is awkward, how about "Aircraft-based estimates of fugitive dust" or "Aircraft observations of fugitive dust"

*Changed to "aircraft-based".*

"Aircraft-observation-based estimates of fugitive dust emissions, shown to be a factor of ten higher than reported values". Clarify "reported", is this from ground observations, reported in some governmental permit, modeled?
*Changed to "higher than reported to national emissions inventories".*

Introduction:

Pg 4 lines 13-17: Hard to follow multiple embedded proportions, just proportionalize everything to Canada.
*Done.*

Pg 4 line 17-18: If NOx and NHx emissions from the oil sands are only 3.8 and 1%, respectively, of the total emissions of NOx and NHx in Alberta, how are they the main anthropogenic sources? Everything else is natural?
*The first two sentences of this paragraph have been changed to "The provinces of Alberta and Saskatchewan are home to the majority of Canada's petrochemical extraction and refining infrastructure, in addition to other industries such as coal-fired power generation, and account for a substantial fraction of the Canadian anthropogenic emissions of sulphur dioxide (34%), nitrogen oxides (43%), and ammonia (50 %), see Zhang et al., 2018). Emissions originating within the Athabasca oil sands region account for approximately 6.5, 1.3, and 0.3% of the Canadian anthropogenic emissions of these three chemicals, based on inventories used in Zhang et al. (2018). ". The original sentence referred to the total Alberta anthropogenic emissions, and implies that the sum of other sources in Alberta contribute to the Alberta totals to a greater degree than the Athabasca oil sands.*

Methods:
18 pages of methods is ridiculous, but if this is ok with the journal, it's ok with me. I'd prefer to see this in an Appendix and have a brief methods section in the main text.

*We have moved the sections dealing with the snowpack observation protocol to the Supplemental Information to reduce the Menthods section; the SI also includes a detailed description of the gas-phase deposition, and is now 14 pages long. We are reluctant to move more of the methods to the SI for two reasons. First, this is the first time that the deposition algorithms used in GEM-MACH have appeared in the literature, and the details of algorithms (and inputs) selected for deposition calculations can have a substantial impact on a model's estimates of deposition. That is, they form a substantial portion of the "novel" part of the work, being unreported in the past, and have a significant impact on the overall results, as we demonstrate in later figures. Second, the critical load estimates, while largely a review of past work, are often difficult to find in the peer-reviewed literature, and yet are also critical to the results of the paper. For example, some of the gridded Canadian estimates we include for historical comparison purposes (section 2.3.2) only appear in the "grey" internal government literature, and a description of how these are ultimately derived from the same source of information as the more recent CLRTAP protocols is also difficult for researchers to access. The descriptions of the CL data employed thus shows the*

*historical record of CL development in Canada, and present information on their construction which up until now have not been easily available to the general scientific public. To clarify our intent for the latter section, we have changed the title of the CL section to "Estimates of Critical Loads of Acidic Deposition in Canada– A Review of Recent Work", and added the text, "In this section, we review recent work on the estimation of critical loads in Canada, starting from the UNECE definitions, in order to provide a complete description of the critical load datasets used in our subsequent estimates of exceedances.".*

Pg 9 line 19-23: The subsampling is likely flawed. Using the 5th percentile is only appropriate if there are many lakes in a grid cell (i.e. > 20). If there are very few lakes in a grid cell, assuming a normal distribution, the lakes will more approximate the mean than anything else. Thus, selecting the "minimum" when there are only a few lakes is the minimum in name only, as these lakes more likely represent the mean. We ran into the same problem in Clark et al. (2018) Ecological Applications and discussed its implications (https://doi.org/10.1002/eap.1703). Please consider revising the methodology (i.e. only use grid cells with > 20 lakes) or discuss this issue in the paper.

*The Jeffries et al (2010) dataset was only available to us in its final form (which is why the resolution is much lower than our 2.5km resolution); the section describes how that data was constructed in the original reference. We agree that this is an issue, however, and have modified the text accordingly: "The SSWC critical load values for each surveyed lake contained within each AURAMS grid-cell were compared – when data from multiple lakes within the same grid cell were available, the fifth percentile of the resulting critical load values was assigned to that grid cell (for grid cells containing less than 20 lakes, the critical load for the most sensitive lake was used). The lake critical load data thus represent the most sensitive lake ecosystems within the given grid cell based on the available data. We note, however, that this procedure used in the creation of this dataset (Jeffries et al., 2010) becomes less accurate as the number of lakes per grid cell becomes small, with either over- or under-estimates of local ecosystem sensitivity. This was one of the factors leading to more recent updates in aquatic critical load maps for Canada, discussed in more detail in section 2.1.4."*

Parse out in the figures and captions of Figs 2-4 the "No Data" category. There are many possible reasons for no data, and knowing and communicating that is helpful. What that "No Data" because it was not the right land cover type (e.g. non-forested) or because there was no estimate of data (e.g. BC dep), or some other reason. These are very different. Please separate the reason for no data into at least two categories (i.e. "No appropriate cover" and "No data").

*This has been indicated in the revised figure captions, with the addition of the following sentences to the captions:*
*Figure 2: No Data: (a) No lake observations were available in the given 45 x 45 km grid cell; (b) No forest data were available and/or the "No Data" regions were not forested".*
*Figure 3: No Data: data was only collected within the province of Alberta (outside of Alberta, no data reflects the limitation of data collection); within Alberta, data was only collected for natural terrestrial ecosystems (no data within Alberta thus refers to landscapes modified by human activities such as agriculture).*

*Figure 4:  No Data:  data were not collected for the largest lakes and river systems within the coloured region; the boundaries of the coloured region represent the limit of the catchment basins for which data were collected.*

Results:

Figure 5: Interesting that most of the N dep is from NH4+ wet, which did not seem to be mentioned much in the introduction. Maybe introduce this a bit more so it's not a surprise.

*Given the material discussed, we assume that the reviewer is referring to Figure 6 (total and percent components of $N_{dep}$); Figure 5 is the corresponding $S_{dep}$ figure.  We note that "most of" is in the sense of a relative local contribution.  One has to be a bit careful here in that the relative contributions in Figures 5 and 6 are percent contributions at each location, not total masses.  That is, $NH_4^+$ is the dominating relative contribution to total N production for downwind regions far from major anthropogenic emissions sources.  The absolute values of N deposition (and hence $NH_4^+$ in those regions (such as northern Saskatchewan and Manitoba)) are low. The manuscript has been modified to mention this, within the discussion of Figure 6: "The bulk of the relative fraction of total $S_{dep}$ close to the sources of emissions is due to dry deposition of $SO_2(g)$ and wet deposition of $HSO_3^-$, while the wet deposition of $SO_4^{(2-)}$ dominates in downwind regions.  The relative fraction of $N_{dep}$ near the sources is dominated by dry deposition of $NO_2(g)$ and $NH_3(g)$ near sources and dry deposition of $HNO_3(g)$ and $NH_4^{(+)}$ further downwind.  Figure 5 (b-e) and Figure 6 (b-i) show that for sites downwind of the source regions (hot-spots in panel (a) of these figures), wet deposition dominates.   We note that the mass of $S_{dep}$ and $N_{dep}$ deposited decreases with distance from the sources; for example, $NH_4^{(+)}$ dominates the relative fraction of $N_{dep}$ in locations more distant from the sources, where total $N_{dep}$ is relatively low.  ".*

Pg. 24-25: Please add a bit more discussion on the sources of error that could be attributable to the model and could be attributable to the observed estimates. Both are potential sources of error and I found this a bit too brief.
*We have modified and added to the last few sentences of the last paragraph of page 24 to read, "Air-quality models such as GEM-MACH are quite complex, with many possible sources of model error; some possibilities include but are not limited to errors in the input emissions data (as we examine below for base cation emissions and deposition), errors in the plume rise algorithms leading to potential errors in the relative distribution of deposition near versus far from the sources (Gordon et al., 2018, Akingunola et al., 2018), potential errors in the magnitude of $N_{dep}$ associated with the absence of bi-directional fluxes of $NH_3$ (Whaley et al., 2018) in the simulations carried out here, and biases within the meteorological forecast components of the model.  As we discuss below, the model predictions nevertheless correlate well with wet deposition observations at precipitation-monitoring stations located downwind of*

*emissions sources, and these relationships allow for an approximate correction of model $S_{dep}$ and $N_{dep}$ estimates using observations. This allows us to reduce the potential impact of sources of model error on estimates of critical load exceedances.".*

Pg. 25: Very nice that you used uncorrected and corrected deposition estimates in calculations of exceedance.

*Thank you. As noted above, we have stressed the importance of these corrections and have recommended in the revised manuscript that some form of model-measurement fusion be employed wherever possible for estimation of total deposition, to reduce model errors.*

As mentioned earlier "Aircraft-observation-based" is a mouthful and not necessary. Use "aircraft-based." It would be silly to take a computer up in the aircraft to make this an "aircraft-simulation-based" estimate.

*Done.*

Pg 27, line 12-13. I don't follow the logic of why this following from the preceding sentences: "This in turn suggests that the primary particles may rapidly deposit with increasing distance from the emissions sources." If the emission inventories are too low, why does that mean they deposit faster? Couldn't this explain the bias in the BCdep (i.e. that the actual emissions are higher, and thus the model estimates BCdep as too low)?

*Larger particles have higher deposition velocities, hence the finding that much of the particle mass containing base cations resides in the coarse mode implies that those particles should not travel far from the sources. The process is well-known from laboratory studies of particle deposition, but implies that the particles containing base cations, which originate in fugitive dust emissions and are largely in the coarse mode, will deposit faster than smaller particles. With increasing distance from fugitive dust sources, then, the neutralization effect associated with base cations contained within those particles would be expected to decrease. This has been clarified in the revised manuscript with the following sentences: "Larger particles have higher deposition velocities compared to particles with diameters of 1□m (c.f. Zhang et al., 2001), and hence these larger, "coarse mode" primary particles would be expected to rapidly deposit with increasing distance from the emissions sources. This in turn implies a reduction in $BC_{dep}$ with increasing distance from the sources, associated with this differential deposition of the larger fugitive dust particles earlier in the transportation process. "*

Pg 27 line 18-19. It shouldn't be "aircraft emissions estimates" (i.e. the emissions of the aircraft) it should be "aircraft-based emissions estimates"
*Done.*

Pg. 27-28. Would be nice to have a "take home message" in terms of how far from a large point source do you need to be before one needs to worry. Seems like from Figure 8 you need to be >100 km, which would be a useful rule of thumb to include. Are these underestimates for Canada

likely occurring elsewhere in the world as well?...Later I see that the 142 km threshold (pg. 33) is presented, which is slightly different but related and useful.

*It's problematic to conclusively state that the 142km distance we see here will hold in other parts of the world, in that the drop-off with distance from the (area, not point) sources of fugitive dust will depend on the size distribution of the emitted particles.  In that sense, to the extent that the fugitive dust size distribution of the region studied is "typical", then yes, we would expect a similar drop-off with distance from the sources of fugitive dust.  To our fourth point in the Discussion section, we have added the following:  "We also note that the 142 km drop-off distance associated with $BC_{dep}$ shown here is a function of the size distribution of the emitted fugitive dust particles – while our expectation is that the bulk of fugitive dust emissions are likely to be in the coarse mode (sizes greater than 2.5 μm diameter) as they are here, differences in the initial size distribution may lead to different decrease functions with distance from fugitive dust sources.  However, a general result from our findings is that fugitive dust base cation neutralization will be limited in spatial scope, due to the effect of particle deposition increasing with increasing size in the coarse mode.".*

Pg 29 line 1-2. Unclear why the improvement in the BC emissions using the aircraft observations also improves the Sdep and Ndep, please clarify.
*Whoops – that should have read "the use of this correction and other observation-based corrections on the original model estimates"; this has been corrected in the revised manuscript.*

Section 3.4. It's not really clear to me what value the comparison with snowpack adds except to say that the snowpack isn't a very useful comparison. If that is the case, I'd move all this to the supplement, and just have a short statement to that effect. This would shorten an already very long paper.
*The main addition (and one as a result of which we think this section is worth retaining in the revised manuscript) is one of measurement methodology, and how the measurements can/should be compared to model values.  We have identified the "open" versus "throughfall" deposition estimates as a potential confounding factor, of which modellers of atmospheric deposition may be unaware.  We have added the following sentence to our fifth discussion point: "Snowpack deposition observations should attempt to measure both "throughfall" and "open" deposition, in order to more accurately estimate total deposition to snow-covered vegetation. ".*

Figure 12 a, what is the red hotspot to the NE of the Athabasca oil sands where $BC_{dep}$ again dominates? Is that another source or evidence of long range transport?
*No – rather, it suggests potential regions of base cation accumulation in surface waters.  The given panel shows the ratio two different base cation fluxes, the first an estimate of BCdep to the region from sparse observations gathered to the south, and the second an estimate of the amount of base cations leaving the region via export in catchment waters, as was described in the text referencing this figure.  Areas where the ratio is greater than unity thus represent places where accumulation of base cations might be expected to occur, and hence where, over time, $BC_{dep}$ might be expected to have a greater role in the neutralization of  $S_{dep}$ and $N_{dep}$, at 2013 emissions levels.*

Figure 14 b-c: Unclear whether the S and N dep are precipitation adjusted, which I think they should be in one of either b or c? The description in Figure 13 is more clear and complete (i.e. "GEM-MACH S+N deposition scaled according to precipitation observations, base cations scaled using precipitation and aircraft data".
*This has been corrected in the revised caption for Figure 14.*

Not really clear what Figures 15 and 19 on the Regions adds, please clarify.

*An additional paragraph has been added to describe what these figures show (they are very policy-relevant, in that they show how exceedances could be reduced in different areas), referencing the regions of Figure 1:*
*Figure 15 added text: "Figure 15 presents possible avenues to reduce the impacts of deposition. Areas within Regions 1,2, and 3 with respect to Figure 1 may be brought below exceedance levels through a combination of reductions in $S_{dep}$ and $N_{dep}$, the relative magnitude of each depending on the location of the current $N_{dep}$,$S_{dep}$ on Figure 1, with more than one reduction strategy often possible. However, areas within region 4 may only be brought below exceedance by reductions in $S_{dep}$. Figure 15 thus may be of use to policy-makers in determining strategies to reduce deposition to levels below critical load exceedance."*
*Figure 19 added text: "Figure 19 shows that most of the exceedances for aquatic ecosystems reside within Regions 1 or 2 with respect to the regions shown in Figure 1, and thus may be brought to below exceedance conditions by different combinations of reductions in $S_{dep}$ and $N_{dep}$, depending on the location of the current $N_{dep}$,$S_{dep}$ on Figure 1."*

Discussion:

This is fantastic work for Canada, but some discussion of whether the results and lessons would hold for other parts of the world would be nice if briefly discussed (e.g. Europe and the US?).

Pg 46 line 29-30: In addition to exceedances not helping with the timeline or recovery, they also don't really comment on the magnitude of effect, also an important note.
*The line has been modified to read, "As noted earlier, exceedances to critical loads indicate the potential for ecosystem damage, but not the timeline over which damage may be expected to occur or has occurred, the time to ecosystem recovery (if acidifying deposition is reduced), or the magnitude of the ecosystem impacts of exceedance".*

Pg 47 and Figure 20: This is a really nice addition, but it seems to me that comparing exceedances with actual impact is a much bigger effort than this would suggest. If it's just included as a preview, it's fine, otherwise, maybe caveat that much more comparison with observed effects is needed (e.g. forest tree growth??).

*It has been included to indicate that some observation data suggests that there is some observational evidence that the effects of exceedances are starting to appear in lakes closest to the emissions region – but also that more direct observations and monitoring over time is needed. In that respect, the Figure is intended to show the importance of maintaining aquatic ecosystem observations in the region at risk. We have added "Our calculations of aquatic critical*

load exceedances imply that acidification will eventually occur; Figure 20 highlights the need for ongoing monitoring of aquatic ecosystems in this region."

Please add a non-spatial plot to simplify the information in Figure 20. The take home point appears to be that exceedance does not translate to effect.

*See the above response – we have added text to indicate the intent and implications of Figure 20.*

Pg 49 line 15: I would not characterize a radius of 142 km as "rapid". I'd say 10 km would be rapid, but that circle in the figures is pretty large!

*That part of the sentence has been changed to "rapidly with distance in comparison to the size of the predicted areas of aquatic critical load exceedance"*

Reviewer # 2:

General remarks:

This rather extensive paper reports on detailed deposition calculations for the two Canadian provinces as well as their use in exceedance calculations for different sets of critical loads (CLs) for terrestrial and aquatic ecosystems. As the larger part – and more of the novel material – is concerns atmospheric depositions, I suggest to change the title to "Estimates of Acidifying Deposition and Critical Load Exceedances in Alberta and Saskatchewan" (or similar), and also to restructure the paper accordingly, i.e. first depositions, then CLs and their exceedances.

With respect to depositions I suggest to move some of the material to the 'Supporting Information', since its mostly material taken from existing literature.

*We have reorganized the paper's methodology to follow the section order as described by the reviewer. We have also moved the details of the snowpack methods section to the SI, as later recommended by the reviewer, to reduce the length of the methods section. While we agree with the reviewer that the estimates of deposition are novel (as are our corrections of those estimates using observations), we feel that the use of those deposition estimates to predict exceedances of the critical loads is also novel (and the main point of the paper). The original title was not "Estimates of Critical Loads for Acidifying Deposition and their Exceedances in Alberta and Saskatchewan", which would imply we are presenting the CL values for the first time, but "Estimates of Exceedances of Critical Loads for Acidifying Deposition in Alberta and Saskatchewan", which makes the point that the estimates of exceedances are the main point of the paper, and does not imply that the critical loads themselves are a result of our work. Hence we have left the title unchanged.*

*With regards to moving the deposition description to the SI, the reviewer may have missed the existing SI and the line in the original manuscript "A detailed description of the gas-phase dry deposition module of GEM-MACH (with an emphasis on the chemical species which contribute to $S_{dep}$ and $N_{dep}$) appears in the Supplementary Information; here we provide an overview" (a later comment by the reviewer regarding references appearing in the reference list but not in the paper would seem to confirm this; most of those 'superfluous' references are quoted in the SI). In the 12 page SI for the original manuscript, the gas-phase deposition algorithm of GEM-MACH is described in much more detail, including tables for all of its input parameters. That is, what appears in the main body of the manuscript is already the "condensed" version.*

*With regards to the reviewer's suggestion that the deposition material is taken from the existing literature: a sometimes overlooked aspect of the use of air-quality models such as GEM-MACH for deposition calculations is that there are a large number of different parameterizations present in the literature for deposition, a large number of choices for input variables for those parameterizations, and the choice of parameterization and input variables can have a substantial impact on the model results. Yes, all of the components of the GEM-MACH deposition algorithms appear (separately) in the literature, but the specific combination used in GEM-MACH has not appeared in the literature. That combination is critical to the model results, and as noted in the manuscript and SI, some of the choices are known to result in a factor of two variation in gas-phase deposition levels. Given that the crux of our work is dependent on the GEM-MACH deposition estimates (albeit corrected by observations), we thought it best to include the full description of the GEM-MACH deposition algorithms, particularly since that combination had yet to be presented in the literature. Most of the detailed information has already been moved to the SI, in that respect. We do, however agree with the reviewer's later suggestion that the details of the snowpack observation methodology need not appear in the main body of the manuscript, and have moved this to the SI.*

The paper (and the reviewer) would have benefitted if the authors had carefully read the paper before submission: there are close to 30 references that are there and not cited or cited and not in the reference list (see below); also the equation number is quite faulty in some parts.

*These errors have been corrected. Note that most of the "there and not cited" papers appeared in the supplemental information description of GEM-MACH's gas-phase dry deposition module.*

Furthermore, the definition of critical load exceedances (in case of non-exceedance) requires some attention (see below).

*We have modified the manuscript descriptions following the reviewer's comments, see details below.*

Apart from this, I consider the material of paper suitable for publication, after the authors have also taken into consideration the (often minor) remarks/corrections listed below.

*Thank you – we are very grateful for your careful look through the manuscript; with the large number of co-authors and the multiple sources of information, it was very helpful to have someone with a detailed knowledge of CL calculations and procedures examine the manuscript.*

Detailed remarks:

Note: 'X →Y' means: replace 'X; by 'Y' (in the text).

Title:

See 'General remarks' above.

Abstract:

P[age]1,L[ine] 33: Suggest to change 'protocols' to 'methods' (throughout the paper!), as 'protocol' has its own meaning in the context of CLRTAP!]

*Done.*

P1, L34: Delete 'forest and': forests are terrestrial ecosystems!

*Fair enough – the idea here was to differentiate the earlier and later data for these ecosystems using "forest ecosystem" for one and "terrestrial ecosystem" for the other, but we can see how that causes confusion in the abstract.*

P2, L2: Delete 'emissions and'.

*The emissions levels used as inputs to the model are the key values, here. The sentence start has been changed to "Potential ecosystem damage using 2011/13 emissions data was predicted for regions"*

P2, L7: 'was shown to have' → 'has' (otherwise it sounds the authors have shown that in this paper).

*This has been made more specific: "Base cation deposition was shown to be sufficiently high in the region to have a neutralizing effect on acidifying deposition", which we have shown in this paper.*

P2, L11: 'primary particle dust particles' → 'primary dust particles'

*Deleted.*

Introduction:

P2, L17: Delete 'regional and'.

*Deleted.*

P2, L18: The reference to the CLRTAP Manual sh/could be simplified (throughout the whole paper!) as it's always the same source. Just call it CLRTAP (2017) – with the text in the References as is now under 'CLRTAP, 2004' – since 2017 is the (last) time you accessed it.

*Done.*

P3, L16: 'Estimates of critical loads' →'Critical loads'

*Done.*

P4, L1-2: No! If BCdep is greater than Sdep + Ndep, a large part of that BCdep could be taken up by forests and harvested (i.e. taken away) and thus not be available for neutralizing the S and N deposition; and a case could be made for the converse – If the statement were true, then the CL would be equal to BCdep!

*Very true. The sentence has been changed to: "Both terrestrial and aquatic critical loads are based on the concept of ion charge balance (cations – anions), as well as terms describing the perturbation of the charge balance through, for example, removal of specific ions or groups of ions through , leaching, harvesting of biomass, etc.".*

P4, L5: 'emissions levels' → 'emission levels'.

*Done.*

P4, L7: What is 'alkylization'? (I guess the authors mean 'alkalinisation'?)

*Should have been 'alkalinization'; fixed this.*

P4, L24: 'aquatic and terrestrials' → 'aquatic and terrestrial ecosystems'.

*Done.*

P4, L25: Insert 'deposition' after 'surface'.

*Done.*

Methodology:

P5, L15: 'sum of in equivalents of' → 'sum of'; the criteria is generally reported as 'molar Bc:Al (or Al:Bc) ratio! That's the way the factor of 3/2 appears in eq 4 to convert it to equilvalents!

*The sentence has been changed to: "The most widely used soil chemical criterion is based on the molar ratio of base cations to aluminum (Bc:Al where Bc is the molar sum of calcium ($Ca^{2+}$), magnesium*

$(Mg^{2+})$ and potassium $(K^+)$) in soil solution (the factor of 3/2 in equation (4) converts this term to equivalents).”

P5, L19: There is no 'level of protection' defined for CLs.

*The sentence has been changed to "The critical level of the leaching of acid neutralizing capacity for the ecosystem ($ANC_{le,crit}$) is defined via equation 4.".*

P5, L19-20: It's the chemical criterion (here Bc:Al ratio) that *defines* the critical ANC leaching – the user does not specify the critical ANC leaching (in the case described here), he just computes it!

*We've changed the word "specified" to "defined".*

P5, eq.2: 'CLmax(S)/(1-fde))' → CLmax(S)/(1-fde): i.e. drop the superfluous parentheses.

*We've rearranged eq 2 to have the fraction as in eq. 4, ie. $\frac{CL_{max}(S)}{1-f_{de}}$*

P5, eq. 4: Delete the superfluous parentheses (twice); only the square brackets are needed.

*A minor point, but we disagree – the additional brackets prevent any possible confusion if someone was to re-write the formula (e.g. mistakenly including the 3 into the numerator, the 2 into the denominator of the start of the next function).*

P5, L30: Bc is already explained above (line 15).

*Removed this repeat definition.*

P5, L31: Insert 'annual' after 'long-term'.

*Done.*

P6, L1: 'due to other forms of removal (e.g. harvesting)' → 'due to, e.g., harvesting'. … and all other variables in the text should be in *italics* if they are so in the equations (also further below)!

*Changed as suggested and variables have been written in italics throughout the text.*

P6, L8: 'Q' is already defined on line 1.

*Change to "Q as defined above" here.*

P6, eq6 : Y does not stand for Ca+Mg+K+Na-Cl, but Sy stands for the sum of base cations minus chloride (Ca+Mg+K+Na-Cl).

*The equation has been replaced by:*

$$S_{runoff} + N_{runoff} = Ca_{runoff} + Mg_{runoff} + K_{runoff} + Na_{runoff} - Cl_{runoff} - ANC_{limit}$$

P6, L28: '(i)' $\rightarrow$ '(subscript i)', etc.

*Corrected. We also noticed in this sentence that the subscript "u" definition had been repeated twice; the second definition was removed.*

P7, L4: What is the 'charge x mole equivalent'? moles? moles of charge? …

*We were trying for a definition of the "eq" unit here. We have converted the text to "converted to units of molar equivalent (eq) deposition of $SO_4^{2-}$ (number of moles of the ion times number of charges associated with the ion).*

P7, L22: There is a change in font size from that line onwards – Any reason?

*Probably multiple versions and we missed it (a one point difference in font size in that paragraph). We've made it uniform in the revised version.*

P8, L1: 'In some instances, S deposition (or N) must be reduced to achieve non-exceedance' What do the authors want to say? As it stands it's trivial/obvious.

*Yes – we've removed the sentence.*

P8, Figure 1: (a) Why is the slope of the Critical Load Function (CLF) shown as $45^{o}$? This is a special case only for $f_{de}=0$ (see eq.2): (b) the point $(N_0, S_0)$, computed in eqs. 12, 13 should be shown on the Figure; (c) It should be indicated in the Figure how the quantity E0 is derived, i.e. where $N_A$ and $S_A$ are located on the CLF.

> *(a) It's a sketch – it was not intended to represent any particular case. Presumably the sensitivity here relates to the earlier CLRTAP protocols referencing constant values of denitrification (which result in the $45^{o}$ slope) versus later revisions to the protocols which result in shallower slopes from the horizontal for non-constant denitrification. We've modified the diagram slightly so that it depicts the more general case.*
> *(b) Done.*
> *(c) Given the reviewer's later comment, below, SA and NA themselves are superfluous, but we've added an arrow indicating the value of $E_0$ for the point $N_{dep1}, S_{dep1}$.*

P8, L5: 'denotes ecosystem': No, it does not denote ecosystems, it denotes 'the case for which'.

*Changed to "denotes cases for which $S_{dep}$ and $N_{dep}$ to an ecosystem".*

P8, L15: $E_0$, as a negative quantity, cannot be a distance, only a positive quantity can; e.g. $|E_0| = - E0$ (thus it would be better to define E0 as a positive quantity and make it $-E_0$ in eq. 11)!

See response to P8,L15 comment, below.

P8, L15: There are no 'exceedance lines' – what you mean is the critical load function.

*"The sentence has been changed to "We define here $E_0$, a negative quantity defining the smallest decrease in deposition from the critical load function (i.e. the boundary between the exceedance and non-exceedance regions of Figure 1) to reach the $N_{dep}$, $S_{dep}$ point on Figure 1."*

P8, eq.15:  In fact, NA is the Ndep-value on the CLF for a given Sdep, and the SA the Sdep-value on the CLF for a given Ndep.  It can be easily shown that Sdep-SA is always greater than (or equal to) Ndep-NA, or, to express it in positive terms (i.e. distances):  SA-Sdep ≤ NA-Ndep.  Thus eq.15 simplifies to:

$$(15)\ E_0 = \begin{cases} S_{dep} - CL_{max}(S) & for\ N_{dep} \leq CL_{min}(N) \\ m(N_{dep} - CL_{max}(N) & for\ CL_{min}(N) < N_{dep} < CL_{max}(N) \end{cases}$$

… and eq. 16 becomes superfluous

*Almost.  We think the reviewer may have made a typo on the second line of the revised (15), above; it should be:*

$$(15)\ E_0 = \begin{cases} S_{dep} - CL_{max}(S) & for\ N_{dep} \leq CL_{min}(N) \\ S_{dep} - m(N_{dep} - CL_{max}(N) & for\ CL_{min}(N) < N_{dep} < CL_{max}(N) \end{cases}$$

*The reviewer is quite right – we had solved for the more general case, where the slope of the line joining $(CL_{min}(N),CL_{max}(S))$ and $(CL_{max}(N),0)$ could have a steeper slope than 45 degrees. However, the reviewer has a good point in that equation (2) guarantees that the value of $CL_{max}(N)-CL_{min}(N)$ will always be greater than or equal to $CL_{max}(S)$, and hence the minimum value of the slope is 45 degrees.  Consequently, the maximum of $S_{dep}$-SA,$N_{dep}$-NA becomes unnecessary, since the fastest approach to exceedance will always be via the S path.  Thanks for pointing this out!  We've dropped equation (16)  and related text, and have simplified the equation (15) as per the second version above.*

P9, L5:  insert 'critical loads' after 'estimated'.

*Done.  Thanks for catching this!*

**Note:  for computing $E_0$ is a different distance measure is used than for computing positive exceedances.  This is not really faulty, but peculiar, and should at least be mentioned (and maybe 'justified').  More generally, the authors should give reasons why they map negative exceedances, as policy makers might not be so much interested in them; generally, they are 'happy' when there is non-exceedance (however small)… But it makes 'nice' maps; and maybe there is another reason as well…**

*This is not just a case of making nice maps, as the reviewer is suggesting. Our own experience with policy makers (and the reason why we included the additional E0 term) is that they are also interested in areas which, if emissions are likely to increase in the future, are likely to be in exceedance. The issue here is that the air-quality models such as GEM-MACH provide deposition estimates which are relative to a particular meteorological year but more importantly for a specific emissions inventory year. The emissions inventories used in the models are updated as new emissions data becomes available, usually on a two to three year basis by the relevant agencies (US EPA, ECCC and Mexican government, in the case of the outer domain used for our simulations in this case. Emissions activities and the amount of emissions may change from year to year – and hence the relative sensitivity of regions which are not currently in exceedance, but could be, were emissions to increase, is valuable information for policymakers. To address this point more clearly, following equation (15), the following paragraph has been added:*

*"For deposition levels below exceedance, i.e. within the grey region of Figure 1, the value of $E_0$ describes the proximity to exceedance; the fastest path by which exceedance could occur, relative to current deposition levels. Given that equation (2) guarantees that the slope of the line joining $(CL_{min}(N), CL_{max}(S))$ and $(CL_{max}(N), 0)$ will always have an inclination of less than $45^o$, the shortest path to exceedance will always be via the $S_{dep}$ path. $E_0$ is of potential interest to policymakers, in that this term describes the proximity of regions which are not yet in exceedance of critical loads to exceedance. Small magnitude values of $E_0$ thus describe ecosystems for which small increases in $S_{dep}$ or $N_{dep}$ may result in exceedances of critical loads."*

P9, L11: '… in order to obtain data for critical load estimates': Only for that purpose?

*That was indeed the main purposes of the work in Jeffries et al 2010 – the work was carried out specifically to obtain data to carry out the first estimates of acidification sensitivity of lakes in northern Manitoba and Saskatchewan; these data were used in that paper to calculate critical loads of acidity using the SSWC model.*

P9, L15/16: '…and other related information': What else c/would that be?
*Sentence has been changed to end at "biological surveys".*

P9, L16,17: 'estimates … were conducted': Do you really conduct estimates?
*Changed "conducted" to "created".*

P9, L20: Delete 'lowest'.
*Done.*

P9, L21: 'for grid cells containing smaller number of lakes, the… most sensitive lake was used': Smaller than what? (20? …). And, by the way, percentiles can be compared for any number…

*This was poorly worded on our part. The idea in the original sentence describing the Jeffries et al (2010) reference was that (1) percentiles were calculated for the lakes critical loads within each grid cell and the 5th percentile was used but (2) for lakes less than 20, the idea of a lowest 5th percentile becomes more of an approximation, so in the latter case the CL for the most sensitive lake within the grid cell was used. The sentence has been changed to "...when data from multiple lakes within the same grid cell were available, the fifth percentile of the resulting critical load values was assigned to that grid cell (for grid cells containing less than 20 lakes, the critical load for the most sensitive lake was used).*

P9, L22/23: 45 km2 grid cell – What's that? If it's a 45 km x 45 km grid cell its size is 2025 km2, if it's area is 45 km2, what are the lengths of the sides? - It's no problem to call it a 45 km grid cell in the former case…

*The sentence has been changed to "...within the given grid cell." The dimensions were specified in a sentence just above the one mentioned, so it's unnecessary here.*

P10, L5: Only BC deposition is needed for CL  alculations.

*Disagree – the context here is the entire suite of variables (aka "inputs") required to create a critical load, which includes the $BC_w(T)$ and $Bc_w(T)$ terms, for example. In case the reviewer is disagreeing with the use of the generic word "inputs", which may be misinterpreted, we have replaced this with "; the key spatial data-sets (or base maps) or formulae required for calculating critical loads"*

P10, L6: Critical alkalinity leaching is not an input, but calculated from, e.g., a critical Bc:Al ratio.

*The sentence fragment has been modified to read, "...and a critical base cation to aluminum ratio (used to calculate critical alkalinity leaching)".*

P10, L14: '(a) a critical Bc:Al ratio of 10 and a Kgibb of 3000.0 were used': This is not a simplifying assumption!  Furthermore, provide units for this numbers!

*The sentence has been modified to "...several simplifying assumptions and/or specified functions and values were applied to terms in equations (1) through (4)..." As the reviewer has noted in an earlier comment, Bc:Al is defined as a molar ratio (we've also added text elsewhere, see responses to comments below, which make this more clear, as well as use "molar ratio" rather than "ratio" in the given sentence. The units of Kgibb have been stated as $m^6\ mol_{charge}^{-2}$.*

P10, L16: Replace 'invariant' by 'spatially uniform'.

*Done.*

P10, L16/17: 'the equivalent of the (CLmax(S)/1-fde)) term…':  Although I can infer what you want to say, it's confusing for the non-expert.  What you are doing is modelling denitrification as

a constant flux Nde, (i.e. CLmin(N) = Ni + Nu + Nde) and not with the fraction fde, as in eq.2 – but this has to be explained (better)!

*We've used the reviewer's wording in the revised sentence, which reads "…both $N_i$ and the $\left(CL_{max}(S)/(1 - f_{de})\right)$ term of equation (2) were set to 35.7 eq ha$^{-1}$ yr$^{-1}$, the latter modelling denitrification as a constant flux)."*

P10,L17: Why the somewhat 'awkward' number 35.7? Why not a 36, or 40? Tell the reader that it comes from a nice round number in a different unit and an assumed soil depth of 0.5 m (this is a simplifying assumption that should be mentioned!)

*The original text, "(c) denitrification and net N immobilization was assumed to occur, but at a spatially uniform low level appropriate for well-drained upland forest ecosystems (both $N_i$ and the $\left(CL_{max}(S)/(1 - f_{de})\right)$ term of equation (2) were set to 35.7 eq ha$^{-1}$ yr$^{-1}$, the latter modelling denitrification as a constant flux), (d) ", has been modified to read: "(c) as in section 2.3.3, net N immobilization ($N_i$) was based on a 50 cm depth rooting and assumed to be equivalent to 0.5 kg N ha$^{-1}$ yr$^{-1}$ (35.7 eq ha$^{-1}$ yr$^{-1}$) (CLRTAP, 2018) , (d) denitrification ($N_{de}$) was also set to 0.5 kg N ha$^{-1}$ yr$^{-1}$ (35.7 eq ha$^{-1}$ yr$^{-1}$) following recommendations in CLRTAP (2018) as an upper limit for well-drained soils, (e)"*

P10, L18: 'weathering … was assumed to be dependent on temperature': This is not really a simplifying assumption to me.

*The sentence has been modified from using "…several simplifying assumptions and/or specified functions and values were applied to terms in equations (1) through (4)…"*

P10,eq.17: (a) the minus side should be a plus sign! (b) Why do you insert the numbers (2 times 35.7) for Ni and Nde, but not for (Bc:Al)crit and Kgibb? (c) Some reason should be given why this is now called CL(S+N) and not CLmax(N) as in eq.2.

> (a) *Yes, we caught that typo when going through the paper post-submission as well – thanks for catching it, nevertheless. (b) All specified constant values have replaced the original variable names in the revised formula (c) See the detailed response to the comment about equation (19), below.*

P10, eq.18: The square brackets are superfluous.

*Perhaps, but they are more precise. The problem: to our experience, sometimes readers will misinterpret an exponential such as this to only apply to the numerical constant, and not to the whole temperature function, mistakenly multiplying by the latter instead of including it in the exponential. Keeping the square brackets for the entire expression for the exponential is therefore safer; less likely to be misinterpreted.*

P10, L23:  'Bc$_{we}$ = 0.75 BC$_{we}$':  this is another simplifying assumption, that should be mentioned above.

*The assignment has been removed from that line in favour of an addition to the earlier list, viz. "and (f) Bc$_w$ was assumed to be equal to 0.75 BC$_w$".*

P10, L27:  (a) I guess it's 2.5km x 2.5km (or 2.5km for short) grid cells.  (b) Delete 'lowest'.

*Changed to "2.5km x 2.5km ", and "lowest" were deleted.*

P10, L31:  Parentheses around S$_{dep}$ + N$_{dep}$ superfluous.

*We've changed the line to "simplified for deposition of sulphur and nitrogen", and modified the deposition calculation itself (see following comment and response).*

P11, eq 19:  This is a 'dangerous' equation!  What if N$_{dep}$ < N$_i$ + N$_{de}$?  The remaining N-sink can not compensate any S deposition! Maybe it does never happen in AL and SK(?), but it has to be said that the (potentially) remaining N sink is not used to compensate S$_{dep}$, as the equation does as it stands!

*We agree with the reviewer on this.  This is one of the drawbacks of the NEG-ECP protocol, and we should have made that clear in the original document. We've included the ECCC Forest ecosystem dataset, constructed for that methodology, since the methodology appears in the literature (Ouiment 2005).  The NEG-ECP methodology itself is based on an earlier approach appearing as equation V.19 in the CLRTAP manual; the separation of the CL into separate terms for S$_{dep}$ and N$_{dep}$ via equations (1) through (4) resulted from recognition in the community of the potential drawbacks, such as the one pointed out by the review, of that earlier approach.  To address this issue in the revised manuscript, while still attempting to make use of the historical data constructed for this earlier approach, we've done the following:*

*In the sentence following equation (18), we've added the following caveat, "We note that equation (18), which follows the NEG-ECP methodology (Ouiment, 2005) may lead to potential errors at very low values of N$_{dep}$, in that the nitrogen sinks could potentially compensate sulphur deposition.  To avoid that possibility, we have added the caveat to equation (16) that the rightmost term is replaced by the minimum of 71.4 eq ha$^{-1}$ yr$^{-1}$ and N$_{dep}$ (that is, the calculated nitrogen sink will not be used to compensate S$_{dep}$, in the event that N$_{dep}$ is below the sum of nitrogen immobilization and denitrification (71.4 eq ha$^{-1}$ yr$^{-1}$)). "*

*When we re-examined the code for Figure 17, we realized that we had inadvertently used 37.5 eq ha$^{-1}$ yr$^{-1}$ for the two constants, rather than 35.7 eq ha$^{-1}$ yr$^{-1}$.  When this was corrected, a small increase in the exceedance area for each of the panels of Figure 17 resulted (2$^{nd}$ decimal place for the total area exceeded in each case).  We have corrected this error in the revised figure (the conclusions with regards to this figure are unchanged).*

*Having applied that correction for (min(71.4, Ndep) in equation (16), we have seen a minimal impact on the predicted CL exceedances. The actual area in exceedance with and without the correction is the same to three figures, and the differences are very hard to spot on the resulting revised Figure 13, though we have incorporated the revised values into that Figure in the paper. For the reviewer's benefit we also add an expanded version of Figure 13 here, below, which compares the original figure panels (first column of panels) with those using the value of 71.4 for the final term of equation (16) (second column of panels), versus using min(71.4,$N_{dep}$) ($3^{rd}$ column of panels, as well as, in the $4^{th}$ column, the differences in calculated exceedances for each of the panels (with correction –without correction). The main effect of the min(71.4,$N_{dep}$) correction is to make some of the regions which*

[Figure]

*below the exceedance threshold slightly closer to exceedance. However, the total area in exceedance was unchanged, to 3 significant figures. Given that the impact of the correction is minimal, to the above text we have added the sentence, "In our application of this methodology (see section 3.6.1), this additional correction was found to bring areas which were already below exceedance slightly closer to exceedance, but had no impact on the estimate of the size of areas over exceedance, to three significant figures."*

[Figure]

**Figure R1. Left column of panels: As in Figure 13, Makar et al, 2018. Middle column: using "min(71.4,$N_{dep}$)" instead of "71.4" as the last term in equation (16). Right column: impact on calculated exceedances as a result of modifying the last term of equation (16) (revised methodology exceedances) subtracted by (original NEG-ECP methodology exceedances). While the proximity to exceedance (E0 region) is reduced in the northern part of the domain, the given increases are insufficient to bring these areas into exceedance (the total area in exceedance does not change to 3 significant figures).**

P12, L1: 'permutations' seems a strange expression in this context!

*Changed to "permutations on" to "different estimates of".*

P12, L11: 'Soils' → 'Soil'.

*Done.*

P12, L20:  How deep was the rooting zone?

*This was mentioned earlier in the same paragraph – 50 cm.  We've added it again here for clarity.*

P12, L27:  Comparing $CL_{max}(S)$ with $CL(S+N)$ does not make much sense, as the latter includes N-terms; thus $CL(S+N)$ could be compared to $CL_{max}(N)$.

*Good point.  The sentence has been changed to read, "$CL_{max}(S)$ and $CL_{max}(N)$" values.  The point we were trying to make here is that the forest critical loads for (S+N) calculated through the ECP-NEG protocol imply slightly less sensitive ecosystems than via the more detailed*

*CLRTAP protocol. This being a little difficult, given that the calculation methodology has changed as well.*

P14, L4: '(equations 1 through 16)': This is misleading; e.g. eqs. 1-4 describe the CLs for terrestrial ecosystems. Improve the citation of equation numbers in the whole paragraph! By the way: the exceedance calculations given in eqs. 11-16 are not fully correct for FAB CL functions, as also the first segment of the CLF is tilted.

*Quite right – that should have been '1 through 14, with the addition of our equation 15', and the last equation references should have been 1-10, 11-15, respectively. Corrected in the text. We've added the following sentence to the description of Figure 1: "We also note that equations 11-14 themselves are a slight simplification for the FAB model, which allows for a slight inclination of the CLF for $N_{dep} < CL_{min}(N)$.*

P14,L6: 'predictive maps': wouldn't 'interpolated maps' be more appropriate?

*This has been changed to "maps", the word "predictive" has been removed.*

P14,L6/7: 'four target variables': In the next 2 lines only 3 variables are mentioned (BC, DOC, $SO_4^{2-}$); what's the 4th one?

*The "four" has been changed to "three".*

P19, L1: 'Yao' → 'Yau'

*Corrected.*

*P21, L20 – P22, L4: This paragraph goes into very much detail... Maybe to Supp Info?*

*Yes, we agree with the reviewer – the sections dealing with the details of the snowpack sampling protocol have been moved to the SI.*

P22, L7: Isn't sulphate an anion?

*Yes, that should be removed. Apologies; should have caught that.*

P22, L12: 'Manzaono' → 'Manzano'

*Corrected.*

P22, L13 and eqs: The equation numbers are wrong: On page 11 yu had already eq.19!!

*Sorry! This was a case of 'version control'; different versions of the paper having different equation numbers, and the ordering of the sections moving around between different versions of the paper prior to submission. These last three equation numbers have been corrected.*

P22, L14: 'used in equation (18)': Which eq. 18?

*See above.*

P22, L24:  Delete 'in 2014'(?)

*The date is relevant here in that other criteria were used for site selection in other years, so we have retained the date as in the original submission.*

P25, L1:  Insert 'in' after 'result'.

*Done. Thanks for catching this.*

P25, L3: 'input emissions' → 'emissions inputs'

*Done.*

P26, L4:  Sub-header:  'oil sands' →'Oil Sands region'(?)

*Changed to 'Athabasca oil sands'.  There are several oil sands regions in that part of Canada (in addition to Athabasca, where the study data was collected, there are also oil sands in the Peace River area, and the Cold Lake area.*

P27,L28: 'simulation, of' → 'simulations by'

*Done (was on line 28).*

P31,L30/31:  merge lines.

*Done.*

P33,L13:  Delete parentheses around 'BCdep-Sdep-Ndep'

*Sorry – we think the parenthesis around the term is justified in this case; the bracket is intended to denote the explicit definition being added as a sideline to the main conversation; we've added an "i.e." within the brackets to make that more clear.*

P34,L16:  'sampling for' →'sampling to monitor'

*Done.*

P35,L9:  'critical load exceedances' → 'critical loads and their exceedances'

*Done.*

P35,L13: Sub-header:  'Exceedances with respect to' → 'Exceedances of'

*Done.*

P36,L18:  'Columbia':  NO!  Columbia is 100 times larger (about $1.14 \times 10^6$ km$^2$)!

Thanks for catching this; there was a transcription error in reading from a list of country sizes. The country name in that line has been changed to Qatar ($1.15 \times 10^4$ km$^2$).

P38, L3: 'have increased in size relative to' → 'are larger than'

*Done.*

P40, L1: Sub-header: 'Exceedances with respect to' → 'Exceedances of'

*Done.*

P40, L6: 'equation (7)': No, it's equ.(5), I presume.

*Correct – thanks for catching this; corrected.*

P40, L11: 'superimposed in' → 'superimposed om'

*Changed to 'superimposed on'.*

P40, L20: 'to be in exceedance' → 'to be exceeded'

*Changed to "in exceedance of critical loads".*

Discussion:

P46,L12: 'improve the bias and correlation' → 'reduce the bias and improve the correlation'.

*Changed as suggested.*

P46,L30: 'expected to occur' → 'expected to occur or has occurred'.

*Good point; we've added that correction.*

P47,L4: 'of Figure 17(b)': Or 18(b)? As said in the caption of Fig. 20

*The text was correct; the caption to Figure 20 was corrected (should have been 17b, there).*

Author Contribution:

P51,L7: 'Lakes and Forest' → 'lakes and forest'

*Done.*

References:

The following references are **cited** in the text, but **missing** here:

- NPRI, 2013
  *We've changed this one to Zhang et al., 2018, which describes both the emissions inventory used as well as the sources of information which were included in that inventory.*
- Aherne, 2013

*We've replaced this one with Aherne and Posch, 2013 (that being a more publicly accessible reference which includes the same material; other was an internal report).*

- Aklilu, 201x

Updated to Aklilu et al, 2018

- ECCC, 2014

*We couldn't find this one in the text of the article and are assuming that the reviewer mistook an "ECCC (2017)" for "ECCC (2014)".*

- Nasr et al., 2010
  *Added.*
- Whitfield et al, 2010

*Added.*

- Pregitzer et al, 1990
  *Added.*
- Stockwell et al, 1989

*Changed to Stockwell and Lurmann (1989) and Lurmann et al., 1986, and both were added to the reference list.*

- Gong et al, 2003a

*Added.*

- Gong et al, 2003b

*Added.*

- Makar et al, 2017
  *Added.*
- Wesely et al, 1989 [or is this the same as Wesely, 1989?]
  *The former – corrected.*
- Slinn, 1982

Added.

- Jacobson, 2003 [or should it be Jacobson, 1999, which is given, but not cited?]

*Should have been the 1999 reference – fixed this.*

- Watmough et al., 2015 [or should it be 2014?]

*Should have been 2014 – corrected this.*

- Whaley et al., 2017

*Updated to Whaley et al, 2018 (was in Discussion phase in 2017, full acceptance in 2018, and we did miss it in the reference list!*

The following **references** are **superfluous**, as they are not cited in the text:

*Actually, for most of these (exceptions Henriksen, 1984, and Sverdrup and DeVries, 1994) the references were cited – in the SI on the gas dry deposition module of GEM-MACH (maybe the reviewer missed the SI).*

- Brook et al, 1999

In the SI.

- Dasch and Cadle, 1986

In the SI.

- Ellsworth and Reich, 1993

In the SI.

- Environment and Climate Change Canada, 2017 [I guess that's ECCC, 2017, cited in the text!?]

*It's cited in the text as ECCC 2017 and appears as such in the references, too.*

- Henriksen, 1984

This one was removed.

- Hicks et al, 1987

In the SI.

- Hosker and Lindberg, 1982

In the SI.

- Jacobson, 1999 [maybe 2003? – see above]

*Should have been 1999 in the text; corrected.*

- Meyers et al., 1998

In the SI

- Sverdrup and De Vries, 1994

*Reference added to the text.*

- Voldner et al., 1986

In the SI.

- Wesely and Hicks, 2000

In the SI.

Figures:

Figure 1:  Improve as suggested above

*Done.*

Figure 2:  Caption:  'Lake ($S_{dep}$)' is a somewhat starnge notation.  Why not give the equation number used to calculate the CL.  Same for 'Forest ($S_{dep}+N_{dep}$)'.
*Changed to "(a) Critical load for acidity (eq. 5) and (b) Forest ecosystems (eq. 18) (eq ha$^{-1}$ yr$^{-1}$).".*

Figure 4:  Caption:  Replace '$S_{dep}+N_{dep}$' by '$S_{dep}$ and $N_{dep}$' [and add (FAB model)' after it].
*Done.*

Figure 5:  Caption:  'Sulphur'  →'sulphur' or 'S'.
*Done.*

Figure 6:  Caption:  'Nitrogen' → 'nitrogen' or 'N'.
*Done.*
- Plate (i):  Isn't it the **sum of** particulate nitrate (dry), gaseous organic nitrate (dry), etc.? And not **each of**?

*No it's "each of"; the point being that each of these species contributes less than the lowest colour interval of the chosen scale; i.e. less than 10%, as stated in the original figure caption.*

Figure 13:  Caption:  Add that Alberta and Saskatchewan are shown in the maps (?) [also in other Figures!]
*Labelling of provinces has been added to all Figures which did not already include that labelling.*
Figure 14:  Caption:  Add that Alberta is shown in the maps (?) [also in other Figures!]
*Labelling of provinces has been added to all Figures which did not already include that labelling.*
- Year after Wang *et al.* is missing.
- *Year has been added back in.*

Figure 15:  Caption:  Add '(see Figure 1)' to explain the regions 1,2,3,4!  The term 'region' in this context is a bit confusing – e.g. 'cases' would be clearer, to distinguish from geographical regions.

*The caption has been changed to "Predicted sub-types of terrestrial ecosystem critical load exceedance (see Figure 1), with panels arranged as in Figure 14.  Inset information shows the area within S + N exceedance sub-types 1, 2, 3, and 4 (km$^2$) and the corresponding percentage of the total area of exceedance. Circled region:  140 km radius diameter circle around the Athabasca oil sands."*

Figure 19:  Caption:  Add '(see Figure 1)' to explain regions 1,2,3,4!

*This portion of the caption has been modified to read, "Predicted sub-types of aquatic ecosystem critical load exceedance (see Figure 1), with respect to deposition of sulphur and nitrogen deposition..  Boxed numbers give the area in exceedance within each of exceedance sub-types 1, 2, 3 and 4 (km$^2$) and the corresponding percentage of the total area in exceedance."*

Figure 20:  Caption incomplete!

*Last sentence of the caption has been modified to "Note that the colour of the symbols, which are for illustration purposes only, does not correspond to numerical exceedance values on the colour scale.".*